# Single-cell RNA sequencing reveals placental response under environmental stress

Eric Van Buren[1], David Azzara [2], Javier Rangel-Moreno [3], Maria de la Luz Garcia-Hernandez [3], Shawn P. Murphy[4], Ethan D. Cohen [5], Ethan Lewis [2], Xihong Lin [1,6] & Hae-Ryung Park [2] ✉

The placenta is crucial for fetal development, yet the impact of environmental stressors such as arsenic exposure remains poorly understood. We apply single-cell RNA sequencing to analyze the response of the mouse placenta to arsenic, revealing cell-type-specific gene expression, function, and pathological changes. Notably, the Prap1 gene, which encodes proline-rich acidic protein 1 (PRAP1), is significantly upregulated in 26 placental cell types including various trophoblast cells. Our study shows a female-biased increase in PRAP1 in response to arsenic and localizes it in the placenta. In vitro and ex vivo experiments confirm PRAP1 upregulation following arsenic treatment and demonstrate that recombinant PRAP1 protein reduces arsenic-induced cytotoxicity and downregulates cell cycle pathways in human trophoblast cells. Moreover, PRAP1 knockdown differentially affects cell cycle processes, proliferation, and cell death depending on the presence of arsenic. Our findings provide insights into the placental response to environmental stress, offering potential preventative and therapeutic approaches for environment-related adverse outcomes in mothers and children.

The placenta is a vital organ for fetal growth and development, engaging in diverse functions such as attachment, invasion, vascular remodeling, cell fusion, hormone production, and nutrient transport[1]. It acts as an interface between the mother and fetus, facilitating the exchange of essential substances while protecting against potential harm[1–4]. Overall, the placenta's multifaceted roles extend beyond its traditional perception as a passive organ, highlighting its active involvement in orchestrating various processes essential for the successful progression of pregnancy and the well-being of both the mother and the developing fetus[1–4].

Accumulating evidence shows that placental dysfunction and gene dysregulation are associated with pregnancy complications and adverse health outcomes in offspring through potential interactions with environmental factors. For example, exposure to heavy metals[5–7],

perfluoroalkyl and polyfluoroalkyl substances[5,6,8], bisphenols[5,6,9–11], and alcohols[12,13], has been associated with abnormal placental growth, differential gene expression, and epigenetic changes in the placenta. These environmental factors have been linked to a spectrum of pregnancy complications, including preterm labor, preeclampsia, pregnancy loss, fetal growth restriction, and adverse neurodevelopmental outcomes in children, such as hyperactivity, impaired memory, and speech and language delays[5–15]. Similarly, in mouse models, prenatal toxicant exposure[16], viral infection[17–20], or maternal immune activation[21] have been associated with neurological disorders, including autism spectrum disorders, alongside placental abnormalities and dysregulation of genes and epigenetic processes. Despite the evidence highlighting the importance of the placenta in maternal and fetal health in the context of environmental factors, our understanding of

[1]Department of Biostatistics, Harvard T.H. Chan School of Public Health, Boston, MA, USA. [2]Department of Environmental Medicine, School of Medicine and Dentistry, University of Rochester, Rochester, NY, USA. [3]Division of Allergy, Immunology and Rheumatology, Department of Medicine, University of Rochester, Rochester, NY, USA. [4]Department of Obstetrics and Gynecology, School of Medicine and Dentistry, University of Rochester, Rochester, NY, USA. [5]Department of Pediatrics, School of Medicine and Dentistry, University of Rochester, Rochester, NY, USA. [6]Department of Statistics, Harvard University, Cambridge, MA, USA. ✉e-mail: hae-ryung_park@urmc.rochester.edu

placental biology and its response to environmental stress is still limited.

Recent advancements in next-generation sequencing (NGS) technologies, such as single-cell RNA sequencing (scRNA-seq), have revolutionized our understanding of complex biological systems. scRNA-seq enables the identification of rare cell populations, unraveling of gene regulatory networks, and tracking of cell lineages during development[22]. Furthermore, scRNA-seq allows us to explore differences in gene expression across cell types and experimental conditions. Previous studies using scRNA-seq or microwell-seq have mapped placental cell types in both mice[23–29] and humans[30–35]. For example, one study conducted scRNA-seq analysis of placental cells of fetal and maternal origin in mouse placentae at E9.5[24], while another focused on E10.5, tracing the developmental trajectories of trophoblasts and highlighting signaling pathways that mediate crosstalk between cell types[25]. In addition, a different research group profiled single-cell transcriptomes of trophoblasts from E9.5 to E14.5 throughout placental development[28]. Furthermore, scRNA-seq has been applied to characterize human placental cell types from the first trimester[30,31] to term[32,33,35]. Tosevska et al.[23] reported cell-type-specific responses in the three layers (decidua, junctional zone, and labyrinth) of the mouse placenta following exposure to particulate matter. While these studies have yielded valuable insights into placental biology, our understanding of placental transcriptomic responses to environmental toxicant exposure at the single-cell level is still in its early stages.

To address this knowledge gap, we conducted the scRNA-seq analysis of mouse placenta exposed to arsenic, a ubiquitous environmental metal toxicant[36]. An estimated 94 to 220 million people are at risk of exposure to arsenic concentrations exceeding the World Health Organization (WHO) provisional guideline value of 10 µg/L in groundwater[37]. Arsenic exposure during pregnancy has been associated with adverse pregnancy complications[38] and neurological deficits in children[39–42]. Our study identified and characterized the arsenic-regulated gene *Prap1* (proline-rich acidic protein 1) and investigated its role in regulating arsenic-mediated toxicity in the placenta. By utilizing in vivo mouse models, in vitro human placental cells, and ex vivo human placental explant cultures, we aimed to elucidate the placenta's unique but understudied role in responding to environmental exposures and its potential implications for maternal and fetal health outcomes.

## Results
### scRNA-seq of mouse placental cells
To better understand cellular responses of the placenta against the ubiquitous environmental contaminant arsenic during pregnancy, we exposed time-pregnant mice to 200 ppb arsenic via drinking water from E0-E17.5 and performed scRNA seq analysis on the transcriptome of the whole mouse placenta (Fig. 1a) including decidua. The mature mouse placenta is established around mid-gestation (~E10.5) and continues to grow in size and complexity throughout gestation[43]. To fully capture transcriptional changes on mature placentae, mice were euthanized at gestational day 17.5 for collecting placentae. Although the mouse placenta has distinctive features from the human placenta, both mouse and human placentae are hemochorial and express many of the same genes that regulate placental development and function[44]. Elevated arsenic exposure, comparable to this study, is linked to adverse pregnancy outcomes (spontaneous abortion, stillbirth, low birth weight), neonatal and infant mortality, and neurocognitive deficits in adults and children[38,45–47], as well as impaired cognitive and motor functions in rodents[48–50]. No variations in the number of placentae, placental weight (Supplementary Fig. 1), fetal malformation, rate of resorption, or preterm delivery were observed due to arsenic exposure in this study.

Based on general marker genes as well as genes reported from previous studies[24–27,29,51–57] (Supplementary Data 1), we identified 36

clusters plotted in two dimensions by transcriptome similarity using uniform manifold approximation and projection (UMAP) (Fig. 1b). In addition to known cell marker genes, we identified potential marker genes including *Car2* (glycogen trophoblasts), *Psg17, Psg25, Pappa2, Ceacam3* (spongiotrophoblasts), *Fnd3c2, Nup62cL*, and *Fdx1* (S-TGCs). These results align with and validate previous research conducted at E14.5[28], underscoring the reliability and significance of our findings (Supplementary Table 1). Figure 1c shows the composition of each cell type by treatment group in the total cell population. Supplementary Fig. 2 displays cell type proportions between male and female placentas for each cell type. Average expression and the percent of cells in each cluster expressing canonical marker genes identified for each cluster are shown in two dot plots (Supplementary Figs. 3a, 4a). The heatmaps in Supplementary Figs. 3b, 4b show the expression of marker genes (y-axis) by cell type and treatment. Each column represents a single cell along the x-axis.

Trophoblasts are the major cell types of the placenta and play an important role in many critical processes during pregnancy, including implantation, hormone production and regulation, immune protection of the fetus, and nutrient supply[2–4]. Here we identify several subgroups of trophoblasts—spongiotrophoblasts, trophoblast giant cells (TGCs), labyrinth trophoblasts, glycogen trophoblasts (GlyT), and trophoblast progenitor cells.

The mouse placenta consists of three distinct layers: decidua, junctional zone, and labyrinth (Fig. 1a). Spongiotrophoblast cells comprise the junctional zone of the placenta between the outer secondary TGCs facing the decidua and the inner labyrinth layer. Clusters 1 and 2 are defined as spongiotrophoblasts based on high expression for *Prl8a9, Flt1*, and *Slco2a1*[27] (Fig. 1b; Supplementary Fig. 3). The transcriptome of Cluster 2 (Spong_1) is highly similar to invasive spongiotrophoblast reported previously by Han et al.[26]. In addition to known marker genes, we found *Psg17, Psg25, Pappa2* and *Ceacam3* are specifically expressed in spongiotrophoblasts (Supplementary Fig. 3; Supplementary Data 1; Supplementary Table 1)[28]. GlyT are differentiated from spongiotrophoblasts and found in the junctional zone of the fully developed placenta. Subsequently, a portion of these cells infiltrate the maternal decidua to form clusters around the maternal spiral arteries[58]. Cluster 9 is defined as GlyT based on the expression of *Plac8, Pla2g4d, Igfbp7*, and *Ncam1*[27] (Fig. 1b; Supplementary Fig. 3). Highly enriched and specific expression of *Car2* in GlyT indicates *Car2* may be a marker gene for GlyT (Supplementary Data 1; Supplementary Table 1)[28]. Consistent with this finding, specific expression of CAR2 protein in GlyT cells from mouse placentae at E10 and E18 has been reported[59].

The labyrinth serves as the interface for exchange between fetal and maternal blood in the mouse placenta[27]. A thin membrane, composed of four cellular layers, separates fetal and maternal blood. These layers consist of fetal endothelial cells, syncytiotrophoblast II (SynTII), syncytiotrophoblast I (SynTI), and sinusoidal TGCs (S-TGCs)[56]. Endothelial cells (Clusters 1 and 11) were identified based on the expression of *Kdr, Pecam1*[25,26], *Plvap*, and *Cdkn1c*[51] (Fig. 1b; Supplementary Fig. 4). We identified labyrinth trophoblasts (Clusters 21, 23, and 25), presumably SynT, based on the expression of *Krt 8*[24,52], *Epcam, Krt19*[26], and *Tfrc*[53] (Fig. 1b; Supplementary Fig. 3). However, due to the multinucleated nature of SynTI and SynTII, it was challenging to differentiate between them using methods such as fluorescence-activated cell sorting or scRNA-seq through microfluidic channels for droplet-based analysis[24,27,52,54]. Based on the UMAP plot, Clusters 21 and 25 are in close proximity while Cluster 23 is more distant, implicating two subgroups of labyrinth trophoblasts. We identified S-TGCs (Clusters 5 and 27) based on the expression of *Ctsj*[25], *Ctsq*[55,56] and *Lepr*[27] (Fig. 1b; Supplementary Fig. 3). In addition, we identified potential gene markers such as *Fnd3c2, Nup62cL*, and *Fdx1* for S-TGCs (Supplementary Data 1; Supplementary Table 1)[28]. We did not identify other types of TGC such as spiral artery-associated, maternal canal-associate, and

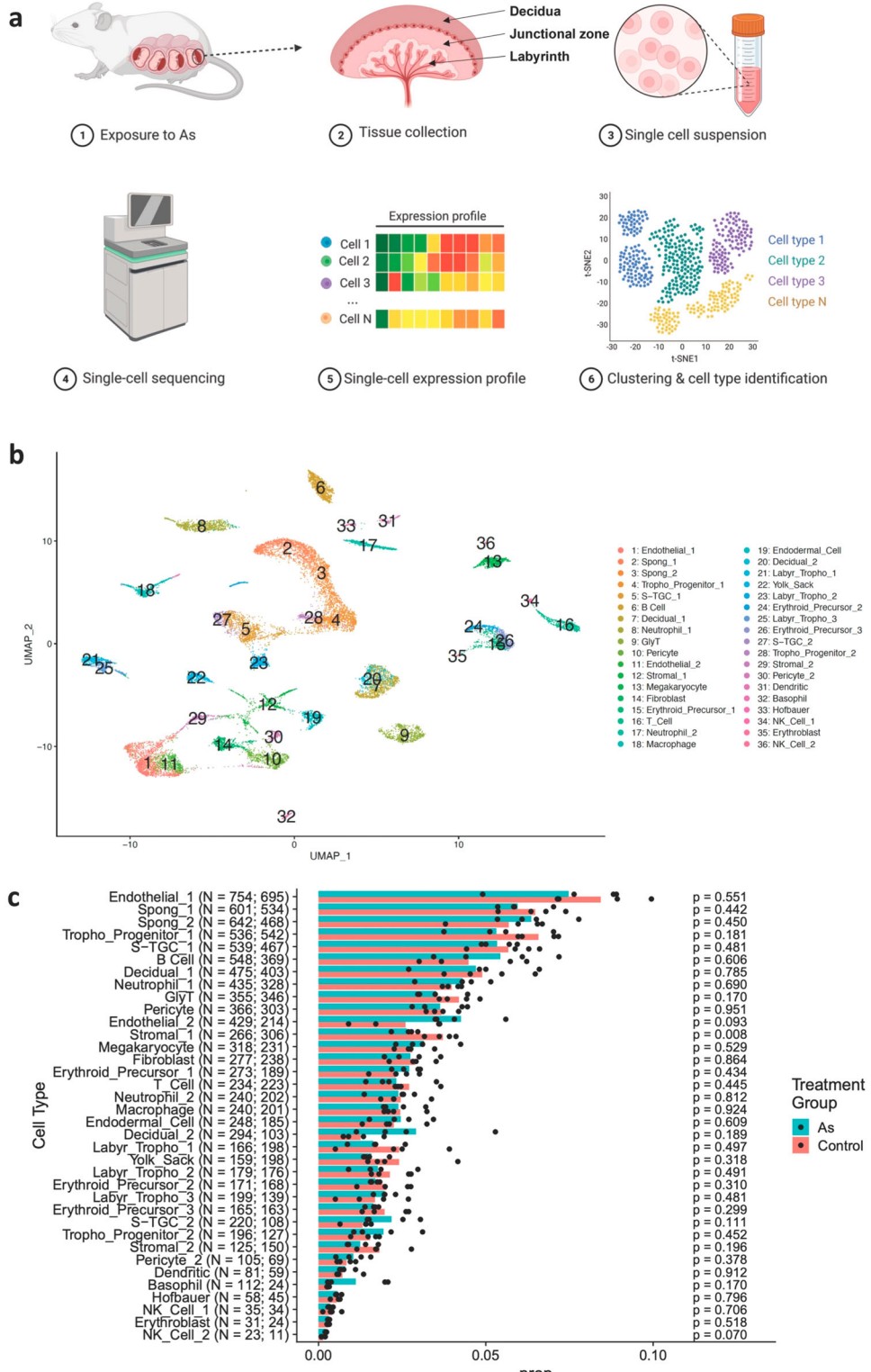

**Fig. 1 | scRNA seq analysis of mouse placenta. a** Schematic workflow of the study. Created with BioRender.com released under a Creative Commons Attribution-NonCommercial-NoDerivs 4.0 International license (https://creativecommons.org/licenses/by-nc-nd/4.0/deed.en)' **b** Visualization of the single cells plotted in two dimensions by transcriptome similarity using uniform manifold approximation and projection (UMAP) with cell clusters (numbers 1-36). Each dot represents one cell colored according to assignment by clustering analysis. **c** Proportion of each cluster by treatment group. *P*-values were calculated using the two-sided *t*-test, and *p*-values reported are not adjusted for multiple comparisons. As arsenic.

parietal TGC, potentially because of low number of cells or genes for these cells. Finally, we identified trophoblast progenitor cells (Clusters 3 and 27) based on the expression of *Foxo4, Isg20, Cited2*, and *Taf7l*[26,51] (Fig. 1b; Supplementary Fig. 3).

Decidual cells (Clusters 7 and 20) were identified based on the expression of *Prl8a2, Adm*, and *Cryab*[26,29] while stromal cells (Clusters 12 and 29) were identified based on the expression of *Col4a1*[51] (Fig. 1b; Supplementary Fig. 4). Fibroblasts (Cluster 14) were identified based

on the expression of *Col1a1, Col3a1, Col6a1, Col5a1, Col5a2,* and *Col6a3*[51]. Endodermal cells (Cluster 19) share both epithelial and mesenchymal characteristics expressing *S100g, Fabp3,* and *Aqp8*[26]. Cluster 22 is defined as Yolk Sac cells with expression of *Apoa4* and *Fxyd2*[25], but its gene signature is highly similar to endodermal cells[26]. We identified pericytes (Clusters 10 and 30) based on the expression of *Acta2, Actg2, Myl9,* and *Mylk*[25–27], which are part of the fetal mesenchyme[27] (Fig. 1b; Supplementary Fig. 4).

We identified a variety of immune cells, including B cells (Cluster 6, *Igkc* and *Cd79a*), neutrophils (Clusters 8 and 17, *S100a9,* and *S100a8*), megakaryocytes (Cluster 13, *Pf4, Nrgn, Gp9,* and *Treml1*), macrophages (Cluster 18 and 33, *Apoe, C1qb, C1qa,* and *C1qc*), dendritic cells (Cluster 31, *Cd74, Cst3, Cd209a,* and *Cd83*), basophils (Cluster 32, *Cd69, Ifitm1, Cd200r3,* and *Mcpt8*), and NK cells (Clusters 34 and 36, *Ccl5, Gzma, Nkg7, Prf1, Gzmb,* and *Cst7*) (Fig. 1b; Supplementary Fig. 2). Based on the expression of *Xist* in male placental samples (Supplementary Fig. 5; Supplementary Table 2), it appears that most immune cells consist of both maternal and fetal cells. The majority of Macrophage_2, B Cell, T_Cell, Dendritic, Basophil, and NK_Cell_1 cells were of maternal origin (Supplementary Fig. 5; Supplementary Table 2). On the other hand, Neutrophil_1, Neutrophil_2, Macrophage_1, Megakaryocyte, and NK_Cell_2 were mostly of fetal origin (Supplementary Fig. 5; Supplementary Table 2). Macrophage_1 is speculated to be Hofbauer cells[25,57], but further validation is needed.

In summary, we have defined cell clusters in the E17.5 mouse placenta and discovered potential cell markers.

## Differentially expressed genes with arsenic exposure

To understand how prenatal arsenic exposure affects the transcriptome of the mouse placenta, we identified differentially expressed (DE) genes for each cell type ($P_{adj} < 0.05$). The full list of DE genes is available in Supplementary Data 2. Supplementary Table 4 shows DE genes that are unique to a single cell type ($P_{adj} < 0.05$, FC > 1.25 or FC < 0.75). Interestingly, the percentages of cells expressing specific genes vary significantly between treatments. For instance, *Pirb* and *Alox15* (Basophil) were not expressed in control cells, whereas they were expressed in over 40% of arsenic-exposed cells. On the other hand, control cells exhibited higher expression levels of *Tnnt2* and *S100a4* (Labyr_Tropho_3) compared to arsenic-exposed cells. The heatmap for unique DE genes shows differential expression levels in each cell type (Supplementary Fig. 6). These data indicate that arsenic exposure affects both the magnitude of gene expression and the proportion of cells within a cell type which express several different genes; these patterns may not be readily detected with bulk RNA seq. The full list of unique DE genes is available in Supplementary Data 3. Additionally, we selected statistically significant genes from four human studies on placental transcriptomic changes with arsenic exposure[60–63], converted these human genes to their corresponding mouse orthologs to evaluate their significance across all cell types in our dataset. Of the 638 genes identified in the above cited studies[60–63], 15 genes showed significant differential expression at a relaxed significance threshold of $p < 0.10$ after adjusting for multiple comparisons as shown in Supplementary Table 3.

Metallothioneins (MTs) are highly conserved, small, cysteine-rich, metal-chelating proteins[64]. While the protective function of MTs in response to arsenic toxicity has been established in other tissues[65–67], their influence on placental toxicity induced by arsenic exposure remains poorly understood. The expression of *Mt1* and *Mt2*, which encode MT1 and MT2 proteins respectively, was found to be significantly downregulated in Labyr_Tropho_3 cells exposed to arsenic, as indicated in Supplementary Table 4. This suggests potential disruption of metal homeostasis by arsenic exposure in the placenta.

*CAR2* was found to be significantly upregulated in GlyT cells following arsenic exposure (Supplementary Table 4; Supplementary Fig. 7). Consistent with previous studies[58,59], CAR2 expression was mainly detected in the junctional zone and decidua of mouse placentae, where GlyT cells reside (Supplementary Fig. 7). *Car2* encodes for carbonic anhydrase 2, a protein with a well-established role in respiratory gas exchange[68,69]. Notably, dysregulation of CAR2 has been associated with placental hyperplasia[59], suggesting that arsenic exposure may impact placental growth and development.

We further ranked DE genes by log₂ Fold change and found that many of them were significant across multiple cell types. The heatmap depicted in Fig. 2 illustrates the expression levels of select DE genes categorized by cell type and treatment. Supplementary Fig. 8 shows the violin plots showing the expression level for these select DE genes, which included *Ctla2A, Guca2B, Gpx3, Apob, Apoa2,* and *Afp*. Notably, with the exception of *Gpx3*, all of these select DE genes encode secreted proteins[70–75].

Proline rich acidic protein 1 (PRAP1) has been implicated as a uterine marker for successful implantation[70,76]. However, its specific roles in the placenta remain largely unknown. In our study, we made an interesting observation that *Prap1* consistently showed upregulation in 26 different cell types, including various trophoblast cell types such as spongiotrophoblasts, labyrinth trophoblasts, S-TGCs, GlyT, and trophoblast progenitor cells, in response to arsenic exposure (Figs. 2, 3a–c). This finding suggests a potential involvement of PRAP1 in regulating placental responses to arsenic exposure.

Cytotoxic T-lymphocyte-associated protein 2-alpha (CTLA2A) is a cathepsin L-like cysteine protease inhibitor expressed in the placenta[77]. While the expression level of *Ctla2a* varies considerably across different cell types (Fig. 2, Supplementary Fig. 8a, and Supplementary Data 2), our analysis revealed that it was significantly upregulated in 22 cell types (Supplementary Data 2). Although the precise role of CTLA2A in the placenta remains poorly understood, it is worth noting that the canonical function of cathepsins is to facilitate placental invasion by degrading extracellular matrix proteins[78].

*Guca2b* encoding Guanylate Cyclase Activator 2B (GUCA2B) was found to be upregulated in 22 cell types; however, its expression was observed in only a small fraction of cells (Fig. 2 and Supplementary Fig. 8b). Consistent with this finding, immunofluorescence (IF) staining of GUCA2B protein shows its sparse expression in mouse placentae, and therefore it was challenging to determine differential expression between groups (Supplementary Fig. 9). Co-staining with Ly6G indicates that much of GUCA2B expression is from neutrophils (Supplementary Fig. 9). GUCA2B is known to regulate salt and water homeostasis in the intestine and kidneys[79]. The significance of GUCA2B's upregulation in the placenta warrants further investigation.

Apolipoproteins are crucial components required for the synthesis and secretion of triglyceride-rich lipoproteins[80]. They serve a vital role in facilitating the transfer of lipids, contributing to fetal growth and development[81]. We found that the expression of *Apob* and *Apoa2* was upregulated in response to arsenic exposure (Supplementary Figs. 8d, e), suggesting a potential disruption in lipid metabolism within the placenta due to arsenic exposure.

Alpha fetoprotein (AFP) is primarily synthesized by the yolk sac and fetal liver[82], and it can cross the placenta and be detected in the maternal circulation[83]. In our study, we observed upregulation of *Afp* with arsenic exposure in four cell types, namely Endothelial_1, Labyr_Tropho_3, Pericyte, and Spong_2 (Supplementary Fig. 8f and Supplementary Data 2). Elevated AFP levels have been associated with adverse pregnancy outcomes[84,85], suggesting a potential link between arsenic exposure and adverse pregnancy outcomes.

We conducted further investigations to explore the impact of sex on placental responses to arsenic. Supplementary Data files 4 and 5 provide information on the significant DE genes by sex in the control and arsenic-treated groups, respectively. Notably, we discovered that one of the select DE genes, *Prap1*, exhibited upregulation specifically in arsenic-treated female placentae, while no significant upregulation was observed in arsenic-treated male placentae (Fig. 3d). Additionally, the

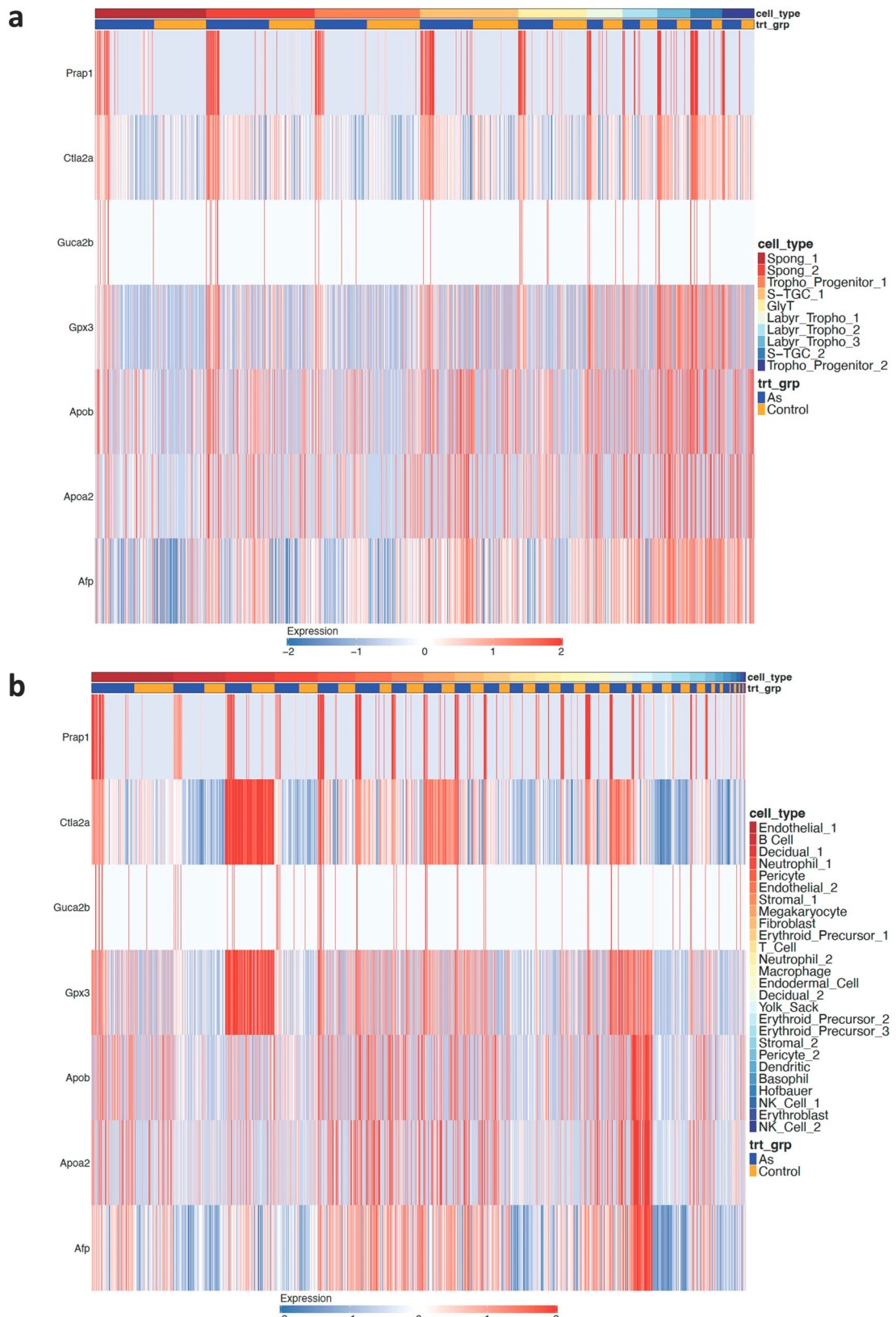

**Fig. 2 | Differentially expressed (DE) genes with arsenic exposure compared to control.** Heatmaps showing expression of DE genes between arsenic exposed and Control cells in (**a**) trophoblasts and (**b**) non-trophoblast cells. As arsenic.

percentage of female cells expressing *Prap1* was found to be considerably higher compared to the percentage of male cells expressing *Prap1* (Supplementary Table 5). In contrast, in the control group, the expression level of *Prap1* as well as the number of cells expressing *Prap1* were low in both female and male placentae (Fig. 3d). Figures 3e and 3f provide a detailed depiction of *Prap1* expression in each cell type stratified by sex. These findings suggest the potential existence of sex-dependent regulation of *Prap1* expression in response to arsenic

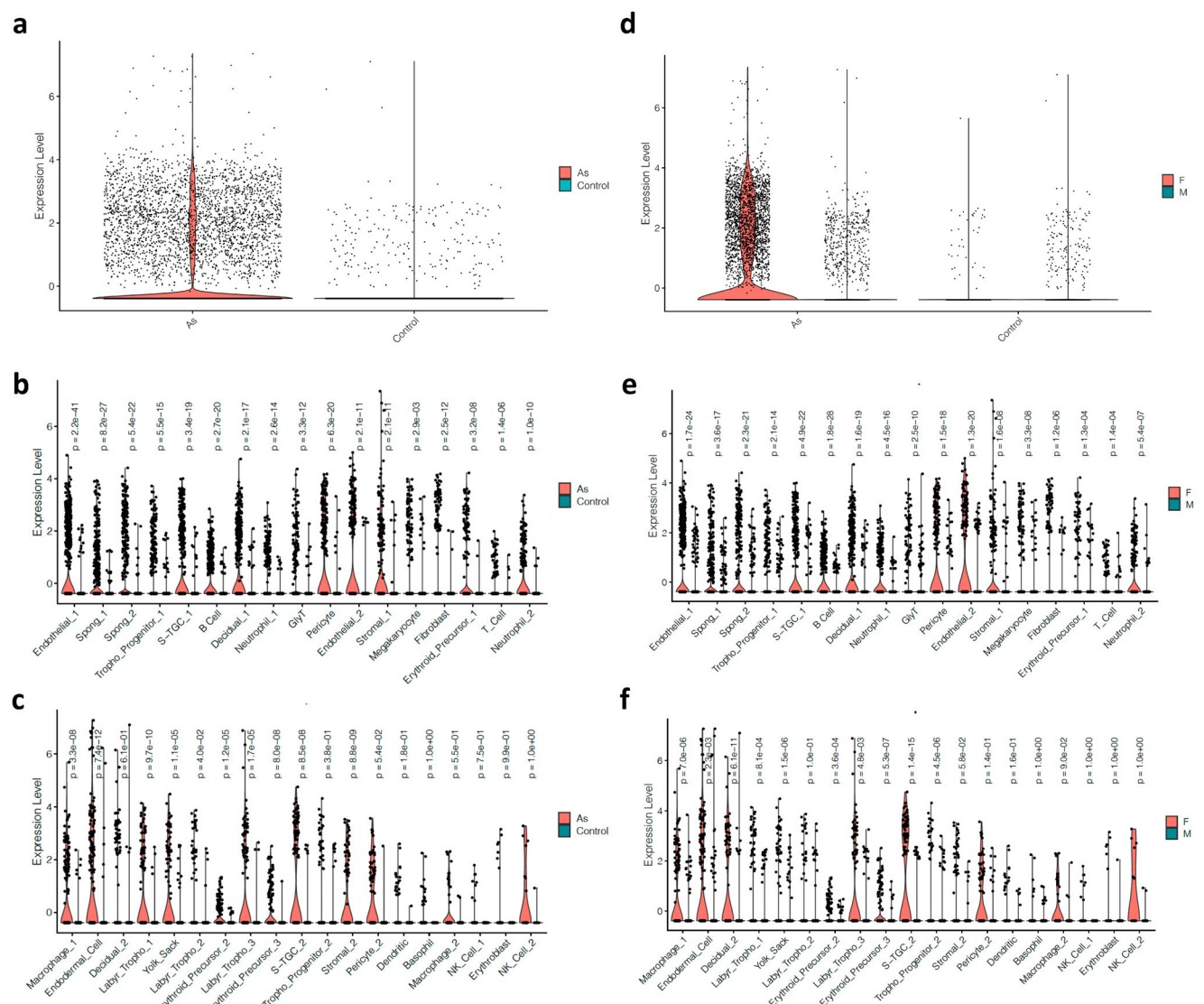

**Fig. 3 | *PRAP1* expression in mouse placenta.** *PRAP1* expression by treatment (**a**); by treatment in each cell type (**b**, **c**); by treatment and sex (**d**); by sex in each cell type from arsenic-treated mouse placenta (**e**, **f**). As arsenic, F female, M male. Differential expression testing was performed using MAST, as implemented in the Seurat package. *P*-values reported are two-sided and adjusted for multiple comparisons using the Benjamini-Hochberg method to control the False Discovery Rate.

exposure, which may explain, in part, sexually differential adverse outcomes mediated by arsenic exposure[86–88].

In summary, with the advantages of scRNA-seq, the present study identified unique DE genes in individual cell types, as well as common DE genes across multiple cell types, in response to arsenic exposure. Moreover, the study discovered genes that have not been previously associated with arsenic exposure or whose functions in the placenta are not well understood. Finally, the study observed sex-dependent differences in the response to arsenic exposure, potentially offering an explanation for the development of sex-dependent pathologies resulting from exposure to arsenic during pregnancy[86–88].

### Gene ontology analysis of each cell type

Gene ontology (GO) term analyses[89] were performed to compare cell types to one another and arsenic to Control cells within a given cell type to provide insight into the distinct functional activities of cell types and changes due to arsenic exposure. Part of the data is presented in Fig. 4 and Supplementary Figs. 10, 11. A complete list of the enriched GO terms is provided in Supplementary Figs. 12, 13.

The junctional zone serves as the primary endocrine compartment of the placenta, responsible for producing hormones, growth factors, and cytokines that are crucial for the normal progression of pregnancy[90,91]. TGCs are one of the major endocrine cells of the placenta, producing steroid and peptide hormones[90–92]. Spongiotrophoblast cells are located immediately beneath the TGC layer and synthesize and secrete peptide hormones[90–92]. Consistent with this, S-TGCs were enriched for lactation, hormone secretion and metabolism (Supplementary Data 3). In addition, they are enriched in terms related to membrane fusion and migration, as they are connected to SynTI through desmosomes, which undergo constant remodeling and have additional signaling and migration functions[27,56,93]. Despite the known role of GlyT in glycogen storage[94], their functions remain incompletely understood. GlyT cells exhibit a notable enrichment in terms associated with protein translation, ribosome biogenesis, and ribosomal RNA (rRNA) processing (Supplementary Fig. 10). This observation suggests the importance of GlyT cells in processes related to cellular growth and development during late pregnancy. Spongiotrophoblasts were enriched for lactation and female pregnancy,

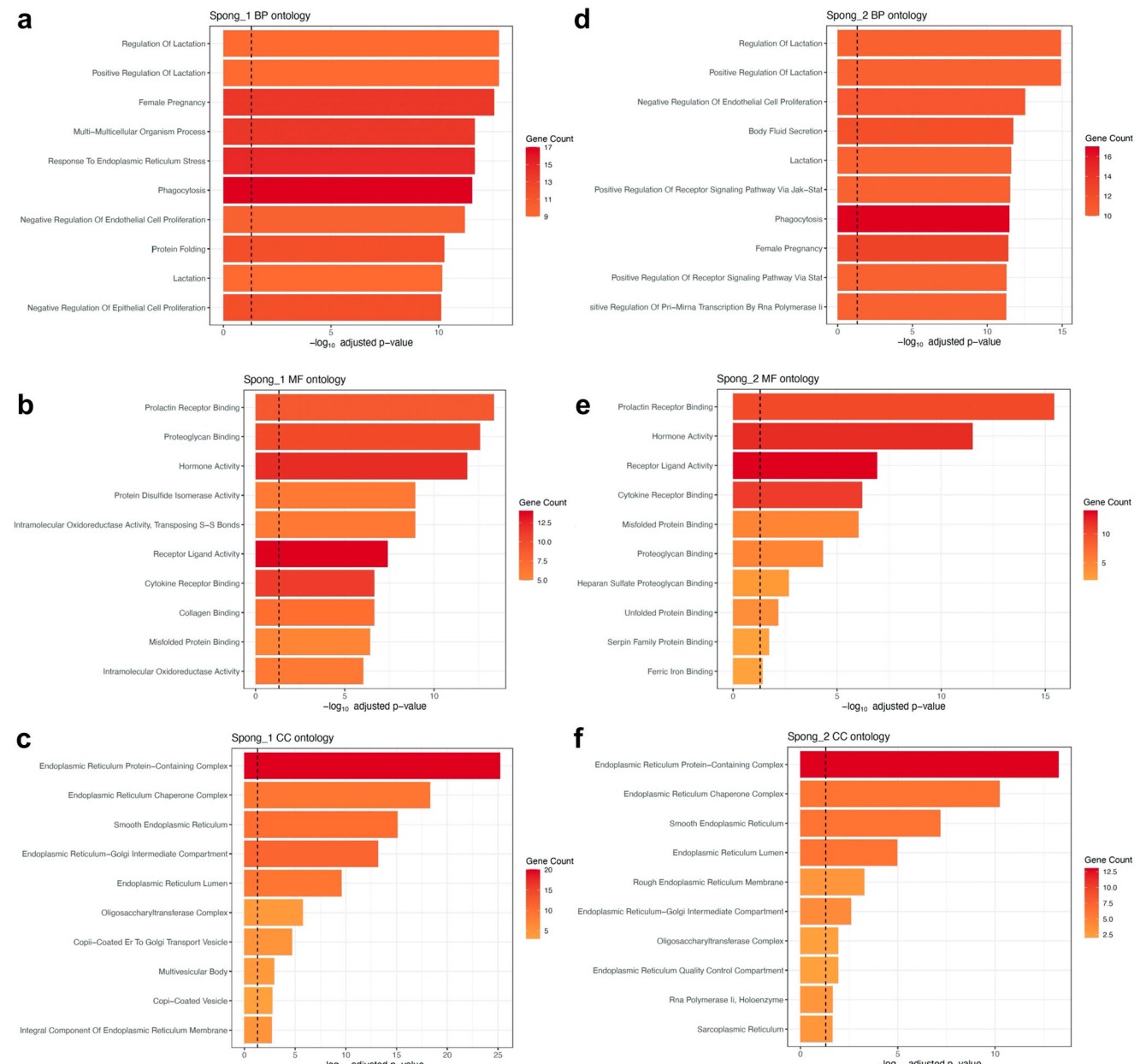

**Fig. 4 | Gene ontology (GO) term analyses comparing arsenic to control within a cell type.** Biological Processes (BP) ontology for selected trophoblast cell types are presented to show enriched terms with arsenic exposure compared to Control for a given cell type. **a** Spong_1, (**b**) Spong_2, (**c**) GlyT, (**d**) S-TGC_1, (**e**) Labyr_Tropho_2, (**f**) Labyr_Tropho_3. GO term analyses are based on the hypergeometric test, with two-sided *p*-values adjusting for multiple comparisons via the Benjamini-Hochberg method to control the False Discovery Rate. Full GO analysis results for all cell types are available in Supplementary Figs. 12, 13.

consistent with their endocrine function. Notably, they exhibit a high enrichment in terms related to the endoplasmic reticulum (ER), ER stress, protein folding, and vesicle transport (Supplementary Fig. 11). This indicates that spongiotrophoblasts are actively engaged in the synthesis and transport of secretory proteins and participate in regulating ER stress, which is crucial for maintaining proper protein folding and cellular homeostasis.

GO term analyses comparing arsenic to Control within a given cell type revealed mechanistic insights on arsenic-induced placental changes and associated adverse health outcomes in mothers and children. Spongiotrophoblasts are enriched for monocyte chemotaxis upon arsenic exposure and GlyT are enriched for phagocytosis (Fig. 4a, c), implicating increased inflammation and tissue damage in the junctional zone of the placenta. Additionally, the terms related to

ER stress (Spong_1) and regulation of protein stability (GlyT) are enriched (Fig. 4a, c), implicating increased protein misfolding and disturbance of ER homeostasis by arsenic exposure. This result is well aligned with the GO term analysis result comparing cell types (Supplementary Figs. 10, 11), indicating that Spongiotrophoblasts and GlyT play roles in regulating ER homeostasis and protein biogenesis, respectively. Notably, many trophoblasts and non-trophoblast cells are enriched for terms related to lipid metabolism, protein-lipid complex remodeling and organization, and lipid homeostasis (Fig. 4b, d, e and Supplementary Fig. 13). This is consistent with identification of DE genes such as *Apob* and *Apoa2* which encode lipid-transporting apolipoproteins (Fig. 2 and Supplementary Fig. 8).

Overall, GO term analyses illuminated specific functional roles for each cell type, with both unexpected and anticipated functions.

Furthermore, our investigation identified arsenic-mediated functional alterations within each cell type, contributing to a deeper understanding of the mechanistic underpinnings of arsenic-related pathologies.

### Immunofluorescence staining for PRAP1 in mouse placenta

Among DE genes, we decided to validate and characterize *Prap1* because its role in the placenta is not well understood, and there is evidence suggesting its potential protective effects against environmental stress[70,95]. To confirm the upregulation of *Prap1* in response to arsenic exposure, we conducted immunofluorescence staining for the PRAP1 protein in E17.5 mouse placenta. Upon examining histological

changes using H&E staining, the overall tissue structure and architecture of the placenta remained unaffected by the treatments (Fig. 5a, top panel). The major regions of the mouse placenta, the decidua, junctional zone, and labyrinth, were clearly identifiable in the H&E-stained images (Fig. 5a, top panel). Immunofluorescence staining of PRAP1 revealed a prominent expression pattern localized mainly in the junctional zone of the placenta (Fig. 5a, second panel from the top), implicating differential post-transcriptional and translation regulation by different cell types[96] as well as the unique roles of the junctional zone in the synthesis transport of secretory proteins such as PRAP1 (Supplementary Figs. 10, 11). Consistent with our scRNA-seq analysis, the female placenta treated with arsenic exhibited a wider PRAP1 signal

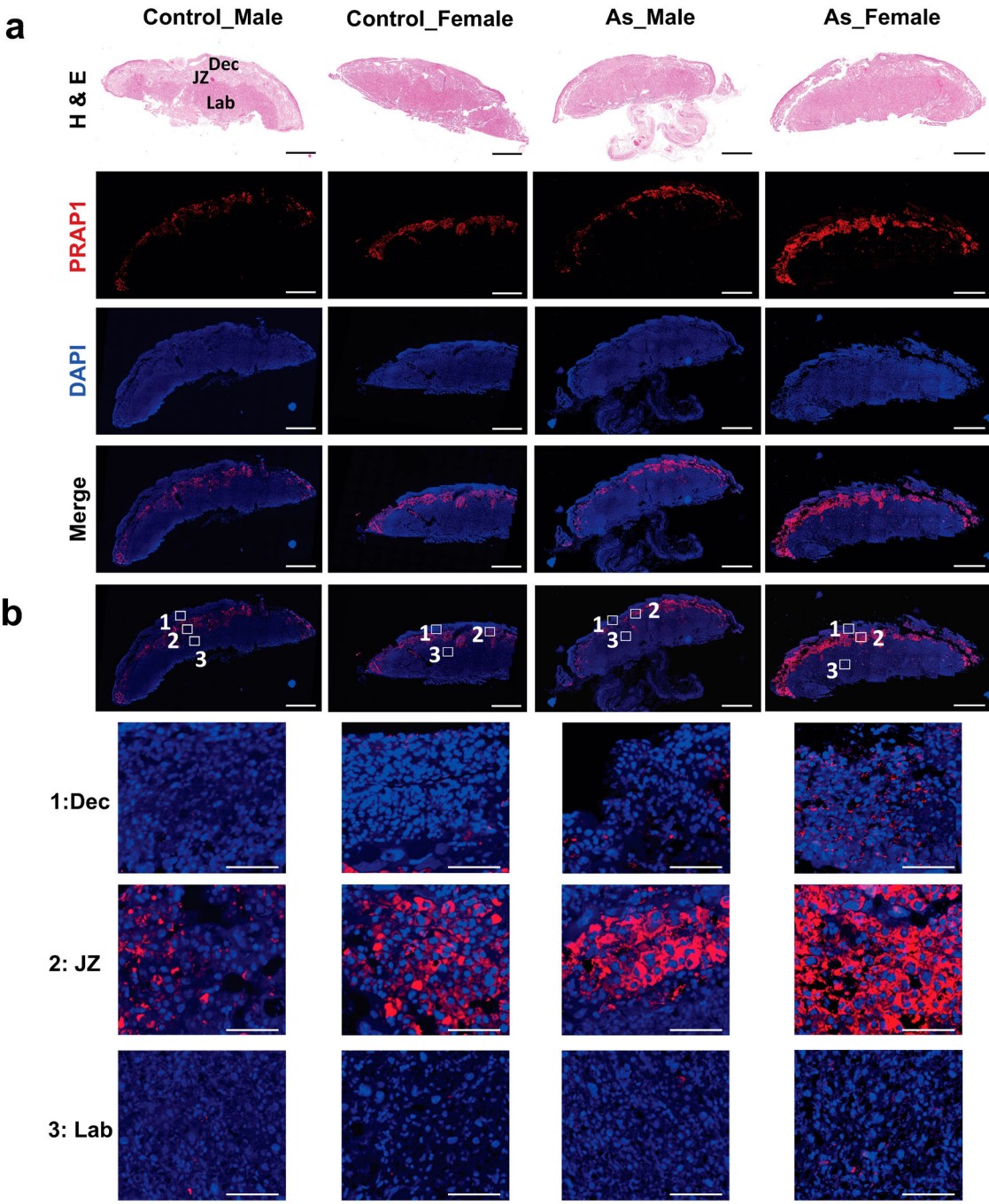

**Fig. 5 | Histology and immunofluorescence staining of PRAP1 in mouse placenta. a** Hematoxilin and Eosin (H & E) staining (**b**) Immunofluorescence staining for PRAP1. Scale bars represent 1000 μm (**c**) Magnified images at three regions (1. Dec Decidua, 2. JZ Junctional Zone, C. Lab Labyrinth). Scale bars represent 100 μm. As arsenic. (*n* = 7 placentae in Control_Male, 4 in Control_Female, 7 in As_Male, and 12 placentae in As_Female).

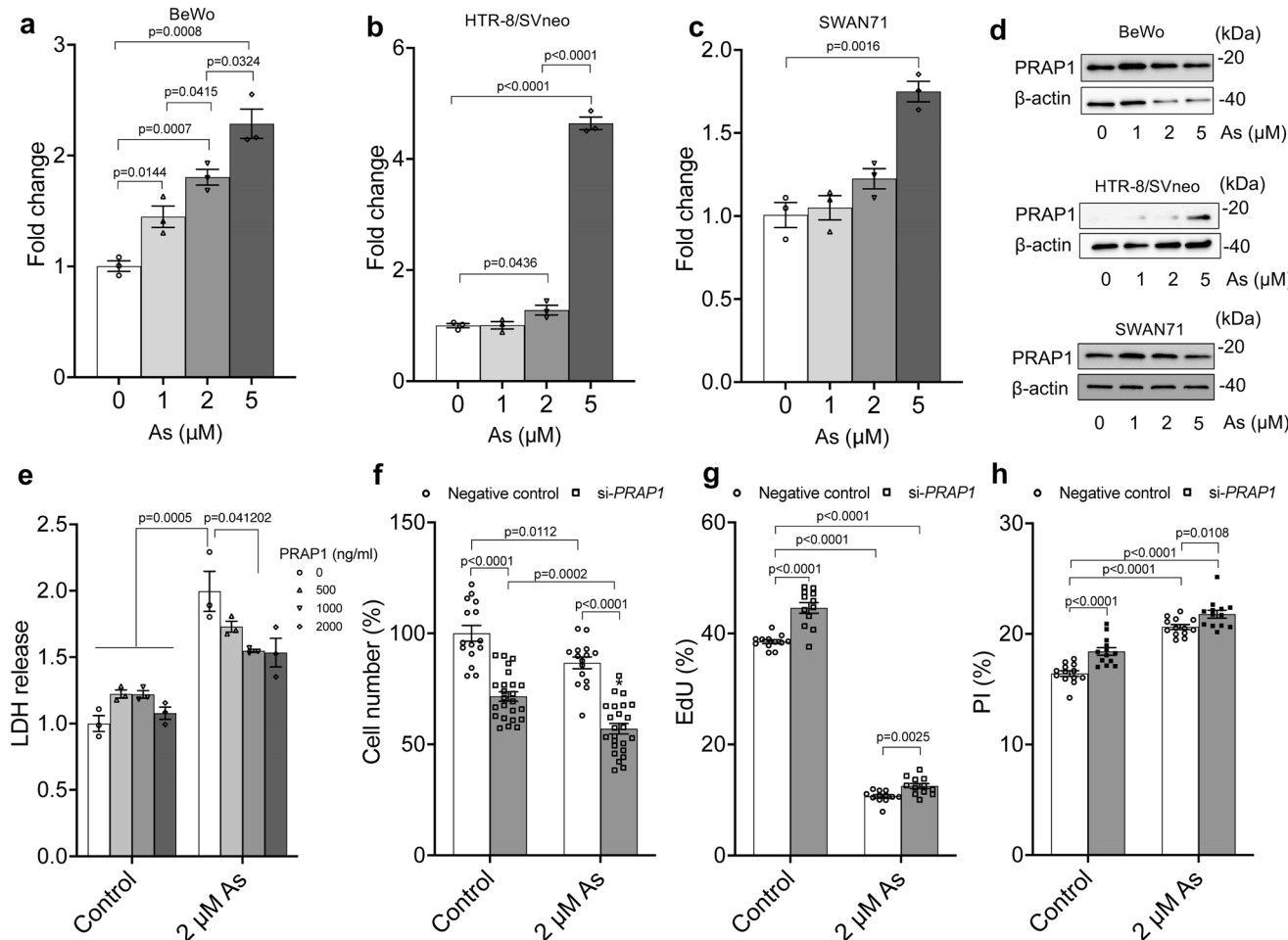

**Fig. 6 | Validation of *Prap1* expression in human placental cells.** Expression of *Prap1* in human placental cell lines (BeWo, HTR-8/SVneo, and SWAN71) exposed to arsenic measured by (**a–c**) qRT-PCR (normalized to GAPDH) or (**d**) western blotting. $n = 3$ cell culture plates in each group. Unpaired, two-tailed *t*-tests (**e**) Effect of PRAP1 on LDH release in BeWo cells with or without arsenic treatment ($n = 3$ cell culture plates in each group; Unpaired, two-tailed *t*-tests). Effect of PRAP1 knockdown on (**f**) cell number measured by Cyquant assay ($n = 15$ cell culture plates in each group; Two-way ANOVA followed by Tukey's test adjusted for multiple comparisons), (**g**) cell proliferation measured by EdU incorporation ($n = 12$ cell culture plates in each group; Unpaired, two-tailed *t*-tests), (**h**) cell death measured by PI staining in BeWo cells ($n = 13$ cell culture plates in each group; Unpaired, two-tailed *t*-tests). As arsenic. Source data are provided as a Source Data file.

in all three regions compared to the other groups (Fig. 5). Semi-quantification of PRAP1 expression from immunofluorescence images reveals significantly higher expression in Arsenic_Female compared to Arsenic_Male (Supplementary Fig. 14, Source data are provided as a Source Data file). Further experiments conducted on ex vivo mouse placental explants have corroborated a sex-specific regulatory pattern in *Prap1* expression, as detailed in Supplementary Fig. 15 (Source data are provided as a Source Data file). These findings provide the evidence implicating sexually differential expression of PRAP1 in response to arsenic exposure and highlight the regional distribution of PRAP1 within the mouse placenta.

**In vitro and ex vivo validation of PRAP1 expression**
To investigate the potential upregulation of *Prap1* in human placenta exposed to arsenic, we conducted experiments using multiple human placental cell lines. Since trophoblasts are the predominant cell type in the placenta and *Prap1* showed increased expression in all trophoblast cell types according to scRNA-seq analysis of the mouse placenta, we utilized human trophoblast cell lines, including HTR-8/SVneo, BeWo, and Swan 71.

As shown in Fig. 6a–c, treatment of human trophoblast cell lines with increasing concentrations of arsenic for 24 h resulted in a significant concentration-dependent increase in *PRAP1* mRNA expression. Figure 6d shows increased PRAP1 protein expression with arsenic treatment in the cells. These data confirm that PRAP1 is upregulated in human trophoblast cells upon exposure to arsenic. To test the effect of chronic arsenic treatment on PRAP1 expression, we treated human placental cells (HTR-8/SVneo) for 7 days, then measured PRAP1 expression. Supplementary Fig. 16a demonstrates a significant increase in PRAP1 mRNA expression in the cells. However, at 2 μM arsenic, PRAP1 protein was not detected and β-actin expression decreased, suggesting increased cytotoxicity in the cells with chronic arsenic exposure or compensatory mRNA upregulation to increase PRAP1 protein (Supplementary Fig. 16a, b).

In addition to cell lines, we examined the effect of arsenic treatment on *PRAP1* expression in placental villous explants obtained from human placentas collected from elective terminations. While the average *PRAP1* mRNA expression increased with 5 μM arsenic treatment, we observed highly variable responses among different donors (Supplementary Fig. 16c, d). Due to the limited availability of donor samples, it was inconclusive whether the effects of trimester or sex influenced *PRAP1* expression. To further explore the sex-dependent regulation of PRAP1 in human models, we utilized trophoblasts derived from both male and female human induced pluripotent stem cells

(iPSCs). As illustrated in Supplementary Fig. 16e, these differentiated trophoblasts express the trophoblast-specific markers including *HAND1* and *GATA4*. Treatment with arsenic resulted in a female-biased upregulation of *PRAP1*, consistent with our findings in mouse placentae (Supplementary Fig. 16f).

In addition to *PRAP1*, significant upregulation of additional DE genes, including *APOA2, APOB, AFP, CA2, GPX,* and *GUCA2B* (Supplementary Fig. 6–9 and Supplementary Table 4, 5) identified from scRNA-seq analysis in the mouse placenta, has been validated by qPCR in human trophoblast cells (Supplementary Fig. 17).

Previous studies have reported the potential protective roles of PRAP1 against environmental stress in the small intestinal epithelium and microvascular endothelium[70,95]. To assess the role of PRAP1 in regulating arsenic toxicity in the placenta, we treated BeWo cells to recombinant human PRAP1 protein with or without arsenic, and subsequently measured cytotoxicity using the LDH assay. As depicted in Fig. 6e, treatment with 2 μM arsenic for 24 h resulted in approximately a two-fold increase in LDH release, indicating increased cytotoxicity due to arsenic exposure. Treatment with 1000 ng/mL PRAP1 protein significantly reduced LDH release at 24 h, suggesting a potential regulatory mechanism of PRAP1 against arsenic-induced toxicity.

In summary, our findings confirm that arsenic treatment triggers PRAP1 expression in human trophoblast cell lines and also demonstrate that PRAP1 mitigates arsenic-induced cytotoxicity in these cells.

### RNA-seq on human placental cells treated with PRAP1

To gain deeper insights into the impact of PRAP1 on placental cells, we conducted global transcriptomic profiling in the human placental trophoblast cell line BeWo, treated with recombinant human PRAP1 protein (Supplementary Fig. 18a). Subsequently, we performed GO analysis to identify enriched pathways among the differentially expressed genes treated with PRAP1 (Supplementary Fig. 18a, b). Notably, the majority of significantly upregulated pathways were associated with transmembrane transport and signal transduction when cells were treated with exogenous PRAP1 protein (Supplementary Fig. 18b, top panel). In addition, the pathway related to the regulation of cyclin-dependent protein serine/threonine kinase activity, crucial in cell cycle regulation, has been significantly upregulated (Supplementary Data 6). Furthermore, pathways related to cell cycle regulation, G2/M transition, and mitosis were significantly downregulated, suggesting potential roles of PRAP1 in these processes (Supplementary Fig. 18b, bottom panel). The dysregulation of selected DE genes, including *CDKN1A, CCNE1, CENPF, CDK1, CDK7,* and *PLK1*, involved in cell cycle regulation, is summarized in Supplementary Table 6. On a different note, the significant downregulation of the estrogen signaling pathway suggests a potential interaction between PRAP1 and estrogen signaling, potentially influencing the sex-differential regulation of PRAP1 expression. The complete list of differentially expressed genes and enriched pathways is provided in Supplementary Data 6.

Building on the pathway enrichment analysis that revealed a significant downregulation of cell cycle regulation, we conducted cell cycle analysis in cells treated with non-targeting siRNA (Neg) or anti-PRAP1 siRNA (si-PRAP1) and exposed to control (NT) or 2 μM arsenic for 24 hr. Consistent with previous studies indicating that arsenic treatment blocks the G1 to S transition[97,98], the percentages of arsenic-treated cells in S and G2 were lower than in Neg/NT cells (Supplementary Fig. 18c). Conversely, the number of cells in G2 was higher in si-PRAP1/NT cells compared to Neg/NT cells (Supplementary Fig. 18c), aligning with the reduced expression of G2 to M checkpoint genes in cells treated with PRAP1 protein (Supplementary Fig. 18b; Supplementary Table 6; Supplementary Data 6). The number of cells in G2 in si-PRAP1/arsenic cells decreased compared to si-PRAP1/NT cells, confirming the impact of arsenic on G2 decrease (Supplementary Fig. 18c).

Supplementary Fig. 16d illustrates the histogram comparing the four groups.

### PRAP1's impact on human placental cell dynamics

Building on our findings from the RNA-seq and GO analysis, along with the preliminary cell cycle analysis, we delved deeper into the role of PRAP1 in regulating cell proliferation and cell death in placental trophoblast cells in vitro, both in the absence and presence of arsenic. In Fig. 6f, we observed that PRAP1 knockdown significantly reduces the cell number compared to the negative control, as measured by the Cyquant assay. Furthermore, PRAP1 knockdown exacerbates the arsenic-induced reduction in cell number (Fig. 6f). In Fig. 6g, we found that PRAP1 knockdown significantly increases cell proliferation, potentially explaining the observed increase in the G2 phase with PRAP1. Conversely, arsenic treatment leads to a significant reduction in cell proliferation (Fig. 6g), aligning with the decreased S and G2 phases observed with arsenic (Supplementary Fig. 18c, d). Intriguingly, PRAP1 knockdown increases cell death, measured by PI staining, in both the absence and presence of arsenic (Fig. 6g). This may elucidate the overall reduction in cell number (Fig. 6f), despite the observed increase in cell proliferation (Fig. 6g).

In summary, these data suggest the potential involvement of PRAP1 in cell cycle processes, cell proliferation, and cell death. These findings may contribute to an altered placental cell function, influencing placental responses to environmental stress, such as exposure to arsenic.

## Discussion

Understanding the impact of environmental exposures on placental responses and its implications for maternal and fetal health is crucial for elucidating the underlying mechanisms and developing preventive strategies. In this study, we employed scRNA-seq to profile the transcriptomic responses of the mouse placenta exposed to arsenic, an environmental metal toxicant associated with adverse pregnancy complications and developmental deficits in children[38–42]. We identified genes that were differentially expressed by cell type, treatment, and sex. Using GO term analyses, we uncovered the functional roles of each cell type as well as potential mechanisms for arsenic-mediated adverse health outcomes. We further characterized PRAP1 as a regulatory protein against arsenic toxicity in the placenta. This study presents a single-cell survey of the mouse placenta under arsenic exposure and provides insights into potential mechanisms underlying environment-related pregnancy pathologies and adverse health outcomes in children.

Previous studies have reported the dysregulation of placental gene expression in response to arsenic exposure in vitro, in vivo, and in human studies[7,60–63,99–101]. However, these studies did not fully elucidate the heterogeneous responses from various cell types within the placenta, limiting our understanding of the intricate mechanisms involved. With the advantages of scRNA-seq, the present study identified unique DE genes in individual cell types, as well as common DE genes across multiple cell types, in response to arsenic exposure. We also found the expression levels of DE genes and the proportion of cells expressing each gene varied significantly within cell types and treatment groups. The number of DE genes was modest because we used a moderate dose of arsenic designed to mimic a biologically-plausible exposure. However, the study demonstrated high sensitivity in detecting modest but significant changes in gene expression within each cell type. These changes may have been masked in bulk RNA sequencing due to low cell numbers in rare cell types or modest fold changes[96]. Moreover, the study discovered genes that have not been previously associated with arsenic exposure or whose functions in the placenta are not well understood. Examples of these genes include *Car2, Prap1, Ctla2a,* and *Guca2b*.

In comparison to four existing human studies[60–63], we identified 15 genes that were significantly regulated by arsenic in both our study and one or more of the referenced studies (Supplementary Table 3). The limited replication observed between our study and these prior works may stem from several factors: (1) Different analysis objectives and priorities, such as differential transcript usage highlighted in Deyssenroth et al.[60]. (2) Challenges in directly comparing results from bulk and single-cell studies. (3) Inherent difficulties in replicating findings within human studies, exemplified by the singular replication of the *PRDM6* gene across two of the cited studies. (4) Modest signal strengths in the original studies, complicating the achievement of sufficient statistical power for replication. Moreover, inherent biological differences between humans and mice may also contribute to these disparities.

The concomitant upregulation of *Prap1* and *Guca2b* in the placenta under various environmental stresses has been reported in vivo. For instance, *Prap1* and *Guca2b* were upregulated in the placenta from mice undergoing in vitro fertilization (IVF) using plastic ware compared to IVF using glassware or in vivo mating[102]. Furthermore, they were upregulated in the placenta from high-fat diet-fed mice[103] or from Schistosoma mansoni-infected mothers mated during the $TH_2$ phase of infection[104]. These findings suggest their potential roles as stress-responsive genes against multiple environmental stimuli such as chemicals, nutrition, and pathogens.

PRAP1 is a conserved secreted protein highly expressed in the uterine luminal epithelium[70] and has been proposed as a marker for successful embryo implantation in mice[76,105]. Our study demonstrates that PRAP1 is upregulated in response to arsenic exposure in both mouse and human placental models. Notably, we observed a female-biased expression pattern of PRAP1 in the mouse placenta and human trophoblasts derived from iPSCs. However, we did not observe sex-dependent regulation of PRAP1 in human placental tissues. This discrepancy may be attributed to various factors including limited tissue samples, inherent species differences, and differences in experimental designs—such as in vivo versus ex vivo approaches, chronic versus acute exposure, and stages of pregnancy. The complex nature of sex differences in biological and medical contexts suggests that phenomena observed in vivo do not always replicate in vitro[106–108]. Moreover, findings in animal models are not always replicated in human models[109–111]. These discrepancies highlight the challenges in translating laboratory findings to clinical practice and underscore the need for careful consideration and comprehensive research to fully understand the implications of biological sex differences on human health.

Although the mechanisms underlying the sex-dependent response to arsenic exposure are still elusive, previous research has implicated the involvement of estrogen signaling in PRAP1 regulation indicating potential link between sex hormones and PRAP1 expression[112]. Consistent with this, our RNA-seq and GO analysis on PRAP1-treated human trophoblast cells identified a significant downregulation of the estrogen signaling pathway (Supplementary Fig. 18b; Supplementary Data 6). Further mechanistic studies will be warranted to test the role of estrogen in PRAP1 regulation.

Studies have consistently shown that arsenic exposure has differential impacts on placental gene regulation and pregnancy outcomes based on sex[61–63,86–88,99,113–116]. For example, transcriptomic analysis of human placentas from the New Hampshire Birth Cohort Study has revealed that female placentas exposed to arsenic show gene expression patterns associated with lower birth weights, unlike male placentas[61]. Moreover, the expression of GLI family zinc finger 3 gene has shown a negative correlation with arsenic exposure in female placentas, yet a positive correlation with infant birth weight[62]. Additionally, the effect of arsenic on placental expression of epigenetic regulators appears to be sex-specific, and is potentially linked to varied outcomes in birth weight and gestational age among male and female infants[63]. Arsenic in maternal blood or drinking water was associated with decreased birth weight and gestational age, smaller ponderal index in male babies[113,114]. On the other hand, only girls, particularly those born to overweight or obese mothers, showed a significant decrease in birth weight with arsenic exposure[115]. This suggests a complex interplay between maternal factors, fetal sex, and arsenic exposure influencing pregnancy outcomes. Although these studies and our findings suggest that PRAP1 may play a role in the development of sex-dependent pathologies resulting from exposure to arsenic during pregnancy[86–88], in vivo studies investigating the impact of PRAP1 disruption on pregnancy and developmental outcomes, both with or without arsenic exposure, will be necessary to draw any definite conclusions.

Our study also sheds light on the understudied roles of the junctional zone in the mouse placenta. We report that PRAP1 protein is predominantly expressed in the junctional zone of the E17.5 mouse placenta, primarily consisting of spongiotrophoblasts and GlyT. This discovery is consistent with our GO term analyses, which revealed roles of the junctional zone in protein biogenesis, protein folding, ER homeostasis, and vesicle transport—essential processes for the synthesis and transport of secretory proteins like PRAP1 (Supplementary Figs. 10, 11). Additionally, arsenic exposure appeared to induce ER stress and increased protein instability in the junctional zone (Fig. 4), potentially connecting arsenic-induced ER stress[117] with PRAP1 upregulation. Our GO term analyses further revealed that arsenic exposure perturbed lipid metabolism and homeostasis in the placenta (Supplementary Figs. 10, 11), processes crucial for pregnancy outcomes, fetal growth, development, and long-term health[118–124]. Both arsenic exposure and ER stress have been associated with dysregulation of lipid metabolism[125,126]. Notably, PRAP1 has recently been identified as a lipid-binding protein involved in lipid transport and absorption[127]. Of note, our study demonstrates that treatment of human trophoblast cells with PRAP1 leads to significant dysregulation of genes and pathways involved in lipoprotein transport, lipoprotein metabolism, protein folding, and unfolded protein response (Supplementary Data 6). These findings suggest potential connections between ER stress and lipid metabolism, with PRAP1 possibly playing a crucial role in regulating arsenic toxicity in the placenta. It would be of interest to explore the time-course expression of PRAP1 in the placenta with or without arsenic exposure, as well as the role of arsenic-induced ER stress and disruption of lipid homeostasis on PRAP1 regulation.

Finally, we demonstrated that PRAP1 differentially regulate cell proliferation and cell death in human trophoblast cells in vitro, impacting genes and pathways involved in cell cycle processes (Figs. 6e–h; Supplementary Fig. 18; Supplementary Table 6; Supplementary Data 6). Consistent with our findings, previous studies have suggested potential roles of PRAP1 in regulating cell cycle progression and cellular responses against environmental stress[70,95,128,129]. For instance, *Prap1* knockout mice were more susceptible to irradiation-induced apoptosis in the small intestinal epithelium, showing a significant increase in apoptosis and expression of p21, a potent cycle-dependent kinase inhibitor regulating cell cycle progression[70]. PRAP1 was also upregulated in human microvascular endothelial cells exposed to cisplatin and was shown to regulate cisplatin-induced DNA damage repair[95]. Furthermore, PRAP1 safeguards cells from apoptosis by triggering cell-cycle arrest, implying that the induction of PRAP1 expression by p53 in response to DNA-damaging agents contributes to cancer cell survival[128]. Finally, PRAP1 was shown to suppress mitotic checkpoint signalling in hepatocellular carcinoma[129]. Abnormal placental trophoblast proliferation and apoptosis, along with placental dysfunction, have been associated with the pathogenesis of pregnancy complications such as preeclampsia[130,131] and intrauterine growth restriction[132], as well as neuropsychological outcomes in children[133]. Addressing fundamental questions about the mechanisms underlying arsenic-induced toxicity, including the potential roles of ER stress and lipid metabolism, and exploring the roles of PRAP1 in placental

function, will significantly enhance our understanding of the pathogenesis of environment-related pathologies in both mothers and children.

Although this study presents insights into the placental responses to arsenic exposure associated with adverse health outcomes in mothers and children, we acknowledge certain limitations in this research, which have implications for future studies. For example, in vivo studies testing the functional roles of PRAP1 will be warranted to confirm the direct translation of the observed cellular effects to pregnancy complications or developmental deficits in offspring. Therefore, future research should endeavor to address this gap by investigating the impact of PRAP1 disruption on pregnancy and neurobehavioral outcomes in wild type or *PRAP1* KO mice in response to arsenic exposure. In addition, the sex-dependent regulation of PRAP1 and its underlying mechanisms require further exploration in human models to substantiate its public health implications, particularly in regulating arsenic toxicity relevant to maternal and fetal health. The pioneering efforts by Okae et al.[134] in developing trophoblast stem cells from cytotrophoblasts and blastocysts provide an essential tool for understanding the molecular and functional dynamics of human trophoblasts. Recent advancements by Hori et al.[135] in creating trophoblast organoids from these stem cells underscore the potential of these models to offer deeper insights into the molecular and functional dynamics of human trophoblasts. Furthermore, exploring the roles of arsenic and PRAP1 in various cell types, including endothelial cells and immune cells, will enhance our comprehension of the placental response to environmental exposures.

In conclusion, this study provides insights into the impact of environmental exposure on placental responses. By employing scRNA-seq, we defined the transcriptomic responses of the mouse placenta under arsenic exposure at the single-cell level, elucidating cell type-specific gene expression and function that would have been overlooked by bulk RNA sequencing approaches. We identified PRAP1 as a regulatory protein in cell proliferation, cell death, and cellular responses against arsenic toxicity in placental cells in vitro, implicating its potential sex-differential regulation in response to arsenic exposure. While further research is needed to fully comprehend the implications of these findings, they contribute to our understanding of the molecular mechanisms underlying environment-related pregnancy pathologies and provide potential opportunities for the development of preventive strategies to improve maternal and fetal health.

## Methods

### Ethical compliance

All research activities described in this study were conducted in accordance with the ethical standards of the University of Rochester and have complied with all relevant national regulations concerning the use of animals in research. The study protocols were approved by the Institutional Animal Care and Use Committee (UCAR-2021-004) at the University of Rochester. Throughout the duration of the experiment, all animals were housed under controlled temperature and lighting conditions with ad libitum access to food and water. Animal welfare was monitored daily by qualified veterinary staff to ensure their health and well-being. Any signs of distress or illness were addressed promptly under the supervision of veterinary personnel. Euthanasia was performed using $CO_2$ inhalation followed by cervical dislocation. All mice procedures were overseen by certified staff to ensure it was conducted humanely and with minimal suffering. All mice experiments were performed in accordance with relevant guidelines and regulations at the University of Rochester. Both sexes were included in the experiments.

### Animals and exposure

C57BL/6 mice were purchased from the Jackson Laboratory (Bar Harbor, ME). Three months-old female C57/BL6 mice were mated in groups of 3–4 with males. Upon confirmation of a sperm plug, they were separated and exposed via their drinking water to 200 ppb sodium meta-arsenite ($NaAsO_2$) (S7400, Sigma Aldrich). The provisional guideline value by the WHO for arsenic in drinking water is 10 µg/L (10 ppb)[37,136]. Human exposure to arsenic in water supplies can vary and falls within the range of 10–1000 ppb[37,136]. In rodent studies, the administered arsenic doses range from 50 to 5000 ppb[136]. For this study, a dose of 200 ppb has been selected, which is considered low to modest and not excessively toxic based on rodent studies. Moreover, this selected dose aligns with the range of human exposure, yet it has been linked to adverse neurodevelopmental outcomes in offspring[42]. Pregnant mice were euthanized on E17.5, and whole placentae including decidua and fetal tails were collected immediately. The sex of each fetus was determined by using Extract-N-Amp Tissue PCR Kit (Sigma) for *SRY* gene. Primer sequences are listed in Supplementary Table 7.

### Single cell preparation from mouse placentae

To prepare single cell suspension for scRNA seq, fresh whole mouse placentae were washed twice with cold PBS and were enzymatically digested in Liberase TL (Roche, 1:10 in PBS) for 15 min at 37 °C with shaking. Then, placentae were manually disaggregated by pulling up and down 10 times on a 3-mL syringe carrying a 16 G needle, placed back at 37 °C with shaking for another 15 min, then manually disaggregated again with an 18 G needle. The digestion was stopped by adding 2 mL fetal bovine serum (Invitrogen) and was passed through a 70 µm cell strainer. Cells were collected by centrifugation at $300 \times g$ for 5 min and resuspended in 10 mL PBS/0.04% BSA for counting. Erythrocytes were depleted from the samples by MACS using Anti-Ter-119 MicroBeads and LS columns (Miltenyi) according to the MicroBeads' protocol. We utilized two time-pregnant mice for each treatment group (control and arsenic exposure). From each pregnant mouse, we collected 7–8 placentae corresponding to the number of pups ($n = 2$ mice in each group). Once the fetal sex was determined, and we subsequently pooled the cells of identical sex from each mouse to proceed with the scRNA seq analysis.

### Library preparation and scRNA sequencing

Cell suspensions were processed to generate single-cell RNA-Seq libraries using Chromium Next GEM Single Cell 3′ GEM, Library and Gel Bead Kit v3.1 (10x Genomics), per the manufacturer's recommendations, as summarized below. Samples were loaded on a Chromium Single-Cell Instrument (10x Genomics, Pleasanton, CA, USA) to generate single-cell GEMs (Gel Bead-in-Emulsions) (16,500 cells per sample, to target 10,000 cells for capture). GEM reverse transcription (GEM-RT) was performed to produce a barcoded, full-length cDNA from poly-adenylated mRNA. After incubation, GEMs were broken, the pooled GEM-RT reaction mixtures were recovered, and cDNA was purified with silane magnetic beads (DynaBeads MyOne Silane Beads, ThermoFisher Scientific). The purified cDNA was further amplified by PCR to generate sufficient material for library construction. Enzymatic fragmentation and size selection was used to optimize the cDNA amplicon size and indexed sequencing libraries were constructed by end repair, A-tailing, adaptor ligation, and PCR. Final libraries contain the P5 and P7 priming sites used in Illumina bridge amplification. Sequence data was generated using Illumina's NovaSeq 6000.

### Data processing and analysis

All statistical analysis was performed in R (version 4.0.2) using the Seurat package (version 4.1.1)[137]. We kept cells with <5% of reads mapping to mitochondrial DNA and which expressed at least two hundred genes. Then, we removed cells which expressed more than 2500 genes to catch and remove doublets. We also kept only genes expressed in at least 10 cells. Normalization and sample integration were achieved using the NormalizeData function and SCTransform

workflow within Seurat, normalizing expression values across cells and mitigating sequencing depth effects. Initial concatenation of cells across the eight samples revealed remnant batch effects between the different samples (Supplementary Fig. 19). To mitigate this concern, we used the *FindIntegrationAnchors* and *IntegrateData* functions in Seurat using 3000 genes and the "SCT" normalization method to perform comprehensive data integration and remove batch effects[138]. An initial round of clustering was then performed using the *Find-Neighbors* and *FindClusters* functions in Seurat with 25 PCs and a resolution of 0.8 (Supplementary Fig. 20). We subsequently removed cells from the top three clusters due to concerns that they represented maternal uterine tissue given their consistently higher expression of mitochondrial genes than other cell types. After completing these steps, we did not see remaining systematic batch effects by sample id or treatment group (Supplementary Fig. 21). We subsequently assigned cluster labels employing the marker gene list found in Supplementary Data 1, in conjunction with signature marker genes derived from previous studies[24–27,29,51–57], and the Panglao Database. Supplementary Figs. 22 and 23 show the number of genes per cell by treatment group, sex, or sample. Differential expression between the Arsenic and Control groups, and between Male and Female within the treatment groups, was performed using the "MAST" test option of the *FindMarkers* function. Genes were considered as DE if their FDR-adjusted *p*-value was less than 0.05.

### Gene ontology analysis

GO term analyses were performed to compare Arsenic to Control cells within a given cell type or to compare each cell type to another. Analysis was performed using the clusterProfiler R package (version 4.2.2)[139], which takes as input differentially expressed marker genes for each cell type and selects up to 100 genes based on the DE *p*-value. The *FindAllMarkers* function in Seurat was used to select marker genes based on a positive logFC above 0.1 when using MAST for DE testing and the "SCT" normalization option across samples. *P*-values shown are from the hypergeometric test after FDR adjustment for multiple comparisons.

### Histology and immunofluorescence staining

Freshly harvested mouse placentas were fixed in 4% PFA overnight at room temperature, paraffin-embedded at the Molecular Imaging Core in the Center for Musculoskeletal Research at the University of Rochester Medical Center. A 5 μm-thick slide for each sample was stained with Hematoxilin and Eosin. Digital images of the stained images were acquired in a VS120 Virtual Slide Microscope (Olympus, Waltham, MA, USA). Five μm paraffin placental sections were incubated at 60 °C for deparaffinization for immunofluorescence staining with antibodies specific for PRAP1. Briefly, tissue sections were transferred to xylenes and gradually hydrated by sequential transfers into 100% alcohol, 95% alcohol, 70% alcohol and water. Sections were then immersed in antigen retrieval solution (S1699, Agilent DAKO, Santa Clara, CA) and boiled for 30 min. Non-specific binding was blocked by incubating tissue sections with 5% normal donkey serum (Jackson ImmunoResearch Laboratories) at room temperature for 30 min in a humid chamber. Primary antibodies were added to the tissue sections immediately after removing the blocking solution (PRAP1-11932-1-AP, Proteintech; CAR2- 50695-RP02, Sino Biological; GUCA2B- ORB312251, Biorbyt; Ly6G-127606, Biolegend). Slides were incubated overnight with the primary antibodies at room temperature. Tissue sections were washed with PBS and secondary antibodies were incubated for 1 h at RT. Finally, tissues were washed with PBS and mounted with Vecta-shield antifade mounting media with DAPI (H-1200, Vector Laboratories, Newark, CA). Mosaic images were taken with a Zeiss Axioplan 2 microscope and collected with a Hamamatsu camera. Semi-quantification of PRAP1 was performed by measuring the area of PRAP1 expression normalized to total placental area using ImageJ.

### Cell culture and exposure of human trophoblast cells

A first trimester extravillous trophoblast HTR-8/SVneo, a trophoblast-derived choriocarcinoma cell line BeWo, and a human telomerase reverse transcriptase (hTERT)-infected first trimester cell line Swan 71 were used for in vitro experiments. HTR-8/SVneo (CRL-3271) and BeWo(CCL-98) cell lines were purchased from ATCC. Swan 71cells were kindly provided by Dr. Gil Mor (Wayne State University). HTR-8/SVneo and Swan 71 cells were cultured with RPMI medium 1640 supplemented with 10% fetal bovine serum (FBS), and 5% Pen Strep (Gibco). BeWo cells were cultured with F-12K Medium (ATCC) with 10% FBS, and 5% Pen Strep (Gibco). To measure *PRAP1* expression in human placental cells, cells were plated at a density of 50,000 cells/well in a 24-well plate and incubated overnight at 37 °C. Next day, the cells were cultured in the absence and presence of arsenic (1, 2, or 5 μM) for 24 hr. After 24-h incubation, cells were harvested and processed for qRT-PCR or western blotting. For Bulk RNA-seq analysis, BeWo cells were treated with vehicle control (PBS) or human recombinant PRAP1 protein (1000 ng/ml) (ab167965, abcam) for 24 hr, harvested, and processed for RNA extraction, library preparation and sequencing.

### Mouse placental explant culture

Mouse placental explants were prepared according to established methods[140,141] with minor modifications. Pregnant C57BL/6 mice were euthanized between embryonic days 16.5 and 17.5. Freshly collected placenta was rinsed with DPBS, sectioned into six pieces, and placed in individual wells of a 24-well plate containing DMEM/F12 supplemented with 10% FBS and 1% P/S. The explants were incubated overnight to allow for acclimation. The following day, they were exposed to arsenic concentrations of 0, 2, or 5 μM for 24 h in duplicate for each condition. The expression of the gene *Prap1* was subsequently quantified by qPCR. A total of 9 female and 17 male placentae from four pregnant mice were analyzed in this study.

### Derivation of trophoblasts from human iPSCs

Human induced pluripotent stem cells (iPSCs, SKU: 30HU-002) were procured from iXCells Biotechnologies, San Diego, CA, sourced from healthy male (Lot: 400531) and female (Lot: 400530) donors. These cells were maintained and propagated in Human iPSC Feeder-Free Growth Medium (MD-0019, iXCells Biotechnologies) on plates pre-coated with Cultrex (3434-005-02, R&D Systems), adhering strictly to the supplier's guidelines. The iPSCs underwent differentiation into trophoblasts using the same medium supplemented with 100 ng/ml BMP4 (314-BP, R & D Systems), following established protocols[142]. Subsequently, the differentiated trophoblast cells were treated with arsenic concentrations of 0, 2, or 5 μM for 4 or 24 hrs. qPCR analysis was then performed to measure the expression levels of *PRAP1* as well as trophoblast markers *HAND1* and *GATA4*[142].

### Human placental collections and culture

First trimester (6–13 weeks; *n* = 2) and 2nd trimester (14–22 weeks; *n* = 3) placentas were anonymously collected from elective pregnancy terminations. Collection of tissues was approved by the Institutional Human Subjects Review Board at the University of Rochester (IRB ID#: 6740). Informed consent was obtained by the study resident physician. Patients did not receive any compensation as part of this study. All women of childbearing age (15-50 years of age) were included in this study. Tissues from pregnancies with known complications or abnormalities, including sexually transmitted infections, diabetes, hypertension, kidney disease, genetic abnormalities and tobacco or illicit drug use were excluded from this study. All tissues collected for this study were from informed and consented women. Tissues were typically collected within 30 min of completion of the corresponding procedure. Placental villous explants ( ~10–15 mg each) were dissected and adapted to ex vivo culture for 1 h at 37 °C and 5% $CO_2$ in DMEM/F12 (1:1 Gibco, Grand Island, NY), 10% FBS (Atlanta Biologicals, Atlanta, GA)

and 100 U/ml streptomycin/penicillin (Gibco). The explants were subsequently exposed to 0, 2 or 5 μM arsenic for 24 h at 37 °C and 5% $CO_2$, washed three times with 1x PBS, flash frozen in liquid nitrogen and stored at −80 °C until processing for RNA and protein extraction. A minimum of three explants from each placenta was exposed to each experimental condition. To determine the sex of each placenta, the placental tissues were homogenized with an 18 G needle and syringe and genomic DNA from the tissues was extracted using the Blood and Cell Culture DNA Mini Kit (Qiagen). PCR for *SRY* gene was performed using the Extract-N-Amp Tissue PCR Kit (Sigma). Primer sequences are listed in Supplementary Table 7.

### Library preparation, bulk RNA-seq, and gene ontology analysis
The TruSeq Stranded mRNA Sample Preparation Kit (Illumina, San Diego, CA) was used for next generation sequencing library construction per manufacturer's protocols. Briefly, mRNA was purified from 200 ng total RNA with oligo-dT magnetic beads and fragmented. First-strand cDNA synthesis was performed with random hexamer priming followed by second-strand cDNA synthesis using dUTP incorporation for strand marking. End repair and 3′ adenylation was then performed on the double stranded cDNA. Illumina adaptors were ligated to both ends of the cDNA and amplified with PCR primers specific to the adaptor sequences to generate cDNA amplicons of approximately 200–500 bp in size. Raw reads generated from the Illumina basecalls were demultiplexed using bcl2fastq version 2.20.0. Quality filtering and adapter removal are performed using FastP version 0.23.1 with the following parameters: "--length_required 35 --cut_front_window_size 1 --cut_front_mean_quality 13 --cut_front --cut_tail_window_size 1 --cut_tail_mean_quality 13 --cut_tail -y −r". Processed/cleaned reads were then mapped to the GRCm39/gencode M31 or GRCh38/genecode38 (Mouse OR Human) reference using STAR_2.7.9a with the following parameters: "−twopass Mode Basic --runMode alignReads --outSAMtype BAM Unsorted – outSAMstrandField intronMotif --outFilterIntronMotifs RemoveNoncanonical −outReadsUnmapped Fastx". Gene level read quantification was derived using the subread-2.0.1 package (featureCounts) with a GTF annotation file (GRCm39/genecode M31 or GRCh38/gencode42) and the following parameters for stranded RNA libraries "-s 2 -t exon -g gene_name". Differential expression analysis was performed using DESeq2-1.34.0 with a *P*-value threshold of 0.05 within R version 4.0.2 (https://www.R-project.org/). Gene ontology analyses were performed using the EnrichR package.

### Cell cycle analysis
BeWo cells were treated overnight with either negative control siRNA (Catalog number:4390843, ThermoFisher), or si-*PRAP1* (Catalog number: 4392420, siRNA ID:s42213, ThermoFisher) using Lipofectamin RNAiMax transfection reagent (ThermoFisher) following the manufacturer's protocol. After transfection, the cells exposed to either the vehicle control ($H_2O$) or arsenic (2 μM) for 24 hr. Then the cells were trypsinized, fixed in 70% ethanol, stained with FxCycle™PI/RNase Staining Solution (F10797, Invitrogen), and examined using BD LSRII (BD Biosciences). Cell cycle analysis was performed using the online flow cytometry analysis program Floreada (https://floreada.io/).

### Measurement of cell number, cell proliferation, and cell death
BeWo cells were transfected overnight with either negative control siRNA (Catalog number:4390843, ThermoFisher), or si-*PRAP1* (Catalog number: 4392420, siRNA ID:s42213, ThermoFisher) using Lipofectamin RNAiMax transfection reagent (ThermoFisher) following the manufacturer's protocol. After transfection, the cells were plated in 96-well plates at a density of 10,000 cells per well and exposed to either the vehicle control ($H_2O$) or arsenic (2 μM) for 24 hr. Cell numbers were determined using the CyQUANT® Cell Proliferation Assay (C7026, Invitrogen), following the manufacturer's protocol. Cell proliferation

was assessed by EdU incorporation, a measure of DNA synthesis, using the Click-iT™ EdU Proliferation Assay (Invitrogen), following the manufacturer's instructions. Cell death was quantified by Propidium Iodide (PI) staining, which binds to DNA in dead cells. Alternatively, cells were treated with the vehicle control ($H_2O$) or arsenic (2 μM) in the absence or presence of human recombinant PRAP1 protein (500,1000, or 2000 ng/ml) for 24 h. Then LDH release was measured using CytoTox 96® Non-Radioactive Cytotoxicity Assay (G1780, Promega).

### qRT-PCR
RNA was extracted and purified from cell cultures and placental explants using the RNeasy Mini Kit (Qiagen) and reverse transcribed using the iScript cDNA Synthesis Kit (Bio-Rad). The resulting cDNA was amplified using iTaq Universal SYBR Green Supermix (Bio-Rad), 250 nM forward primer, and 250 nM reverse primer in a CFX96 Touch thermocycler (Bio-Rad). Fold changes were calculated using the ΔΔCt method with GAPDH as the housekeeping gene used for normalization. All primer sequences used in this study are listed in Supplementary Table 5.

### Western blotting
Whole-cell lysates were prepared in NP40 buffer supplemented with 1x Halt Protease Inhibitor Cocktail (Thermo) and 1 mM PMSF. Cleared lysates were denatured with 1x Laemmli Sample Buffer (Bio-Rad) and β-Mercaptoethanol as a reducing agent, and heated at 95 °C for 10 min before being run in SDS-PAGE and transferred to a 0.45-μm PVDF blotting membrane. Primary antibodies used in this study include anti-PRAP1 (Proteintech 11932-1-AP) and anti-β-actin (BioLegend 643807).

### Statistics and reproducibility
Statistical analysis for in vitro and in vivo experiments was performed with GraphPad Prism version 10 (La Jolla, CA 92037, USA). Data were analyzed by one way ANOVA, two way ANOVA or *t*-tests. A *p*-value < 0.05 was considered statistically different. Data were expressed as means ± SEM. Sample sizes were not predetermined based on statistical methods but were chosen according to the standards of the field. In vivo experiments were randomized and the investigators were blinded to outcome assessment.

### Reporting summary
Further information on research design is available in the Nature Portfolio Reporting Summary linked to this article.

## Data availability
The single-cell RNA-seqencing data generated in this study have been deposited in the ArrayExpress database under accession code E-MTAB-14214. The processed scRNA-seq data are available at ArrayExpress using the same accession code and on Zenodo at https://zenodo.org/records/10258020. The authors declare that all other data supporting the findings of this study are available within the paper and its supplementary information files. Source data are provided with this paper.

## Code availability
Code needed to reproduce all analyses and figures using the single-cell RNA-sequencing data is available at https://github.com/edvanburen/placenta_code.

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

## Acknowledgements

This study was funded by National Institute of Health/National Institute of Environmental Health Sciences R00 grant (ES029548) to H.P., P30 grant (ES001247) to H.P., R01 (AI111914) to J.R.-M., the start-up fund from University of Rochester Medical Center to H.P., Medicine Department Funds and Lupus Research Alliance (975578) to J.R.-M. National Psoriasis Foundation (NPF851079) to M.G.-H., and U01 (HG012064) to E.V.B. and X.L., T32 (ES007142) to E.V.B., and P42 (ES030990) to X.L. The authors would like to thank the Genomics Research Center and the Molecular Imaging Core in the Center for Musculoskeletal Research at the University of Rochester Medical Center.

## Author contributions

H.P. conceived and designed research; H.P., D.A., and E.L. performed in vivo and in vitro experiments; S.P.M. performed experiments with human placenta. J.R.-M and M.G.-H. performed staining and imaging of mouse placenta. E.V.B. processed and analyzed raw scRNA-seq data. H.P. and E.V.B. analyzed data and interpreted results of experiments; E.D.C. performed cell cycle analysis. H.P. and E.V.B. prepared Figures; H.P. drafted manuscript; All authors participated in editing and revising the manuscript and approved the final version for submission.

## Competing interests

The authors declare no competing interests.
