## [Peer Review File · Nature Communications]

REVIEWER COMMENTS

Reviewer #1 (Remarks to the Author):

In this study, the authors undertook single-cell RNA-sequencing evaluation of the whole placenta from mice exposed to arsenic (As) during the second half of pregnancy. The authors show cell type-specific changes in placental gene expression and pathway/process enrichment driven by As exposure, which led to identification of proline-rich acidic protein 1 (PRAP1) as a notable responder to As in both E17.5 murine placenta and human 1st/2nd trimester samples. The authors used IF staining to demonstrate PRAP1 upregulation in the murine placenta, which was also suggestive of sex-specific differences in its regulation. In vitro experiments using human trophoblast cell lines indicated PRAP1 upregulation in response to As exposure. The authors conclude that their findings offer insight into the placental response to environmental stress/pollutants.

Overall, the study provides some interesting insights into a novel research question. However, the study is mostly descriptive and lacks thorough validation of key findings. Specific comments are as follows:

1. Some of the claims made in the abstract and conclusions of the paper are somewhat overreaching. The sex-dependent upregulation of PRAP1 was not sufficiently validated to be concretely demonstrated. Moreover, a protective role of PRAP1 was not necessarily demonstrated without additional in vivo studies.
2. The abstract should be revised for clarity, paying special attention to distinguish between human and murine findings.
3. Introduction: Is there any available information regarding the potential risk and/or incidence of arsenic exposure?
4. Introduction: Some statements are missing supporting references.
5. Results: What are the complications (maternal or fetal/neonatal) resulting from this or a comparable As dose? Even if not performed in this study, a brief summary of any previously observed phenotypes/outcomes of the As exposure model would be helpful to readers. Given that As is teratogenic, did the authors (or previous investigators) note any fetal malformations? Observational studies would help address this point. Were there any secondary effects of As in the mother?
6. Results: Were fetuses viable at E17.5, or were increased rates of resorption observed? Did the authors observe preterm delivery prior to E17.5?
7. Results/Methods: Did the processed mouse placenta include the decidual tissue? Figure 1a suggests so, but it is not clear. This should be stated in the results and methods. Why did the authors not elect to separate the decidua and placenta for sequencing?

8. Results, Lines 112-117: Could the authors distinguish invading extravillous trophoblasts present in the decidual tissues? Are these included in the GlyT cluster, or would these be among the less-represented clusters mentioned in Lines 147-151?
9. Results: Were the numbers of cells obtained comparable between male and female placentas?
10. Results, Lines 118-169: I think this part of the results is mainly focused on Figure 1, but still please cite individual figure panels for each statement.
11. Results, Line 168: Please dampen this statement, as it is unlikely to be the first time the murine placental cell clusters are elucidated at E17.5.
12. Results, Figure 2 and Extended Data: Did the authors perform protein validation for any of the key DEGs found to be specific to As exposure, besides PRAP1? The authors provide a large amount of discussion/interpretation focused on these proteins in prior research.
13. Results, Lines 272-274: Please be more specific regarding the cell types involved in these processes.
14. Results, Lines 310-327: Did the authors perform any semi-quantification from the IF imaging? Or other immunoassays to detect protein levels?
15. Results, Lines 342-347: How comparable are the human elective termination samples to the late pregnancy (E17.5) murine samples in terms of developmental stage?
16. Results, Lines 348-356: How was the PRAP1 concentration chosen?
17. Methods: For single-cell analysis, please clarify the datasets that were used as reference for cell type annotation.
18. Methods: Did the authors consider using primary placental cells for the in vitro studies? Also, how comparable is the acute (24 h) As exposure used for in vitro studies to the chronic exposure performed in mice?
19. Methods: Prizm should be Prism.
20. Methods: Single-cell data should be deposited and publicly available at publication.

Reviewer #2 (Remarks to the Author):

Arsenic (As) is a world-wide health threat, with regional areas in many parts of the world experiencing very high levels of exposure due to contaminated ground and well water. The question of how As affects the placenta is thus interesting and important, as generational health effects of As exposure could be mediated via disruption of placental biology. Given the many cell types that

make up the placenta, the ability to answer this question at the individual cell-type level has the potential to be quite powerful. In addition, mouse models of As exposure have yielded many insights into As pathophysiology, making the model system used in this study an appropriate one to answer this question.

In the manuscript, the authors determine if As exposure has changed the relative proportion of any placental cell types (it did not), identify differentially expressed genes (DEGs) in placental trophoblasts and non-trophoblast cells (there were only 7), and then after describing these each individually in the text, focus then on PRAP1 for further analysis. They also perform pathway analysis for selected trophoblast cell types, and using between 4-9 genes for each cell type (not clear if these are DEGs), identify enriched pathways, although it is not clear, despite the statistical data provided, that with so few genes to consider here has been any true pathway disruption with biological consequence. Finally, they perform in vitro experiments to validate changes in PRAP1 in response to As exposure, and see from 1.5-3 fold (i.e. marginal at best) changes in PRAP1 expression in all lines except HTR/8-SVNeo cells, where the increase is 4-fold. The sex-specific differences noted in many of these analyses are interesting, but diminished by the few genes used to ascertain these differences, and marginal magnitude of differential expression in response to As.

Overall, no major (or even minor) insights into As pathophysiology are provided, largely because so few genes are differentially expressed, and the gene around which the paper revolves, PRAP1, has a rather enigmatic function. Indeed, the rather unimpressive data on As disruption of gene expression is entirely consistent with human data where no DEGs could be identified in human placenta of As exposed women after FSR correction (Winterbottom Env Health 2019).

Reviewer #3 (Remarks to the Author):

General comments:

Van Buren et al. reported the results of their experiments investigating cell type-specific effects of As exposure on mouse placenta using single-cell RNA sequencing and validation with human placental cell lines. This study is notably one of the first to utilize single-cell technologies to probe cell-specific toxic mechanisms in placenta. Although the study design and outcomes are solid, the manuscript would benefit from major revisions.

General Comments

1. I suggest mentioning specific cell types affected by As exposure in the abstract.

2. Through out the results, the authors are very repetitive with the wording they use in their concluding sentences. They use “insights” and ‘mechanistic insights” too many times. These words are very general and do not convey substantive conclusions. Same is true for the word “revealed” in the discussion.

3. Going along with my previous comment, the results section has quite a bit of interpretation. I think some interpretation is okay in the results section, however, it’s a fine line as to when it’s too much. For example, the summary paragraph in lines 252-264 has information in it that I would think is more appropriate in the discussion (ie discussing As dose).

Specific Comments

ABSTRACT

4. Line 35. What cell type(s) had differentially expressed PRAP1?

INTRODUCTION

5. Line 55. Sentence that starts with “For example, exposure to...”-I recommend inserting relevant citations into the sentence specifically where they are discussed, instead of all at the end of the sentence. The sentence further resembles multiple lists. I think the sentence could benefit from restructuring or splitting into a few sentences.

6. Line 76-80. Consider mentioning study by Tosevska et al which did single-cell analysis on mouse placenta exposed to PM.

a. Tosevska, A., Ghosh, S., Ganguly, A. et al. Integrated analysis of an in vivo model of intra-nasal exposure to instilled air pollutants reveals cell-type specific responses in the placenta. *Sci Rep* 12, 8438 (2022). <https://doi.org/10.1038/s41598-022-12340-z>

RESULTS

7. Line 102-112. The description given in this paragraph of clustering methods belongs in a figure caption or methods section. It’s not really describing results. Consider removing it.

8. Line 61-67. I recommend denoting whether cell types being reference are maternal or fetal.

9. Line 202. In this sentence, the authors define genes by name and symbol. In earlier paragraphs, the authors only use gene symbols. I would recommend keeping this consistent throughout the paper.

10. Line 206 needs a reference.

11. Line 278. Why is this finding surprising?

12. Line 339. You can't make conclusions about sex-specific responses based on cell lines because they are different cells. Please remove conclusion about sex-specific effects in cell lines.

13. Lines 342-348. I'm assuming you did bulk RNAseq on human villous explant tissue. It seems regressive to use bulk when you've used single-cell or cell lines. Plus, these results didn't confirm any of your other findings. I would consider removing this paragraph completely. It doesn't add anything to your

DISCUSSION

14. Line 418. Sentence "We for the first time..." should either have commas (We, for the first time,...) or should read (For the first time, we...) Please revise the sentence.

Manuscript ID: NCOMMS-23-38005-T

We sincerely thank the scientists who reviewed this manuscript. We have dedicated our best efforts to comprehensively address the reviewers' valuable comments, which have significantly contributed to the refinement of the article. The clarifications have been incorporated into the manuscript, both in the text and figures, as cited below.

In this revised manuscript, we have introduced new data on:

1. Semi-quantification of PRAP1 in mouse placentae from immunofluorescence (IF) staining images (**Extended Data Fig. 12**)
2. Protein validation of additional DE genes (*CAR1* and *GUCA2B*) in mouse placentae using IF staining (**Extended Data Fig. 7 and 9**)
3. Effect of As on PRAP1 protein expression in human trophoblast cells (**Fig. 6d**)
4. Effect of chronic As exposure on PRAP1 expression in human trophoblast cells (**Extended Data Fig. 13a and 13b**)
5. Validation of additional DE genes in human trophoblast cells (**Extended Data Fig. 13e and 13f**)
6. Bulk RNA-seq and Gene ontology analysis on human trophoblast cells (PRAP1 vs Control) (**Extended Data Fig. 14a and 14b; Supplementary Table S4; Supplementary Excel File 6**)
7. The impact of *PRAP1* siRNA knockdown in human trophoblast cells on
 - a. Cell number, measured by Cyquant assay (**Fig. 6f**)
 - b. Proliferation, determined via EdU assay (**Fig. 6g**)
 - c. Cell death, assessed through Propidium Iodide (PI) staining (**Fig. 6h**)
 - d. Cell cycle progress (**Extended Data Fig. 14c and 14d**)
8. Analysis on *Xist* expression in male placentae to determine maternal or fetal cells (**Extended Data Fig. 5**)

We also have provided additional information to improve clarity.

1. The number of placentae and placental weights used in scRNA-seq analysis (**Extended Data Fig. 1**)
2. Proportion of male and female cells in each cluster (**Extended Data Fig. 2**)

We feel that the manuscript is improved by these revisions and hope that it will be found acceptable for publication by *Nature Communications*.

Reviewer #1

1. Some of the claims made in the abstract and conclusions of the paper are somewhat overreaching.

Response: We appreciate the valuable insights provided by the reviewer and have made adjustments to the abstract and discussion sections to ensure a balanced interpretation of our findings, address limitations, and outline future research plans (Lines 48-49; 578-588, 596-597).

Our study indicates that As exposure upregulates PRAP1 in both *in vitro* and *in vivo* placental settings, and further highlights the differential impact of PRAP1 on placental cell processes *in vitro*. This prompts us to investigate potential *in vivo* consequences, such as pregnancy complications or developmental deficits in offspring. Future research will include essential *in vivo* studies to ascertain the direct translation of observed cellular effects to pregnancy outcomes or developmental deficits in offspring, involving the impact of PRAP1 disruption in wild-type or *PRAP1* knockout mice exposed to As.

Previous research demonstrated that 17 β -estradiol induces PRAP1 expression in the mouse uterus¹, suggesting involvement of estrogen signaling in PRAP1 regulation and a possible link between sex hormones and PRAP1 expression. Consistent with this, our new data from bulk RNA-seq and GO analysis on PRAP1-treated human trophoblast cells revealed a significant downregulation of the estrogen signaling pathway (Extended Data Fig. 14b; Supplementary Excel File 6). Therefore, investigating the roles of estrogen signaling in PRAP1 regulation becomes crucial to validate potential sex-dependent regulation of PRAP1 expression and its implications in As toxicity.

2. The abstract should be revised for clarity, paying special attention to distinguish between human and murine findings.

Response: We appreciate the reviewer's suggestion, and we have taken steps to clearly delineate human and murine findings in the abstract, ensuring a precise and unambiguous presentation of our study's outcomes (Lines 34, 41-42, 45-46).

3. Introduction: Is there any available information regarding the potential risk and/or incidence of arsenic exposure?

Response: We appreciate the suggestion. In response, we have included information on the risk and incidence of As exposure, as well as the associated adverse health outcomes in our study (Lines 93-96).

4. Introduction: Some statements are missing supporting references.

Response: Thank you for bringing this to our attention. We have now included all missing references in the introduction section. (Lines 58 and 61; Ref #1-4).

5.1. Results: What are the complications (maternal or fetal/neonatal) resulting from this or a comparable As dose? Even if not performed in this study, a brief summary of any previously observed phenotypes/outcomes of the As exposure model would be helpful to readers.

Response: Thank you for the suggestion. As outlined in the methods (Lines 607-613), the WHO provisional guideline for As in drinking water is 10 $\mu\text{g/L}$ (10 ppb)^{2,3}. Human exposure to As in water ranges from 10-1000 ppb^{2,3}, with rodent study doses ranging from 50-5000 ppb³. This study employs a dose of 200 ppb, considered low to modest based on rodent studies and aligning with the human exposure range. Despite its seemingly moderate level, this dose has been linked to adverse pregnancy outcomes and neurocognitive deficits in both adults and children⁴. For example, elevated levels of As in groundwater (≥ 50 ppb) have been linked to an increased risk of adverse pregnancy outcomes, including spontaneous abortion, stillbirth, low birth weight, and

infant mortality⁵. As concentrations above 100 ppb in urine or drinking water are associated with cognitive impairments in children such as low IQ, poor language skills, and vocabulary capacity^{6,7}. In adults, such elevated levels are correlated with an increased risk of psychiatric disorders, depression, and anxiety⁸. Furthermore, rodent studies have demonstrated that As exposure is connected to deficits in learning and memory, as well as impaired locomotion and motor functions⁹⁻¹¹. This information is also included in the text (Lines 115-119).

5.2. Given that As is teratogenic, did the authors (or previous investigators) note any fetal malformations? Observational studies would help address this point. Were there any secondary effects of As in the mother?

Response: We did not observe any variations in placental weight, fetal malformation, or resorption rate due to exposure to As. As mentioned earlier, the selected dose of 200 ppb, while not overly toxic in previous rodent studies, is still associated with adverse neurodevelopmental outcomes in offspring. Additional data on the number of collected placentae and placental weight can be found in Extended Data Fig. 1, and this information is also documented in the text (Lines 119-121).

Extended Data Fig. 1. The number of (a) placentae and (b) placental weights used in scRNA-seq analysis.

6. Results: Were fetuses viable at E17.5, or were increased rates of resorption observed? Did the authors observe preterm delivery prior to E17.5?

Response: No increased rate of resorption or preterm delivery was observed. This statement is also included in the text (Lines 119-121).

7. Results/Methods: Did the processed mouse placenta include the decidual tissue? This should be stated in the results and methods. Why did the authors not elect to separate the decidua and placenta for sequencing?

Response: We included decidual tissues in our scRNA-seq analysis, as is now stated in both the results and methods (Lines 109, 613-614). While acknowledging that placental cells are mostly of fetal origin, and recognizing that previous single-cell studies on mouse placenta often excluded decidual tissues¹²⁻¹⁴, we chose to analyze the whole placenta to comprehensively understand total placental responses and mitigate potential risks associated with the exclusion of fetal tissues when removing decidual tissues.

8. Results, Could the authors distinguish invading extravillous trophoblasts present in the decidual tissues? Are these included in the GlyT cluster, or would these be among the less-represented clusters mentioned in Lines 147-151?

Response: Thank you for your valuable insight. In the mouse placenta, TGC and GlyT cells function similarly to the extravillous trophoblasts in the human placenta¹⁵. We have identified two clusters of decidual cells (Cluster 7 and 20, Figure 1b), suggesting the possibility that one of them may be invasive trophoblasts. Alternatively, as you suggested, they could be included in the GlyT cluster or may not be identified due to low cell numbers in our analysis.

9. Results: Were the numbers of cells obtained comparable between male and female placentas?

Response: In total, we analyzed 9,535 cells from male placentas and 8,800 cells from female placentas. A bar plot displaying cell type proportions between male and female placentas for each cell type is now included (Extended Data Fig. 2). The figure illustrates that the total sample size for each cell type contains cells from both male and female placentas. Importantly, there are no major proportion differences between male and female cells, except for GlyT and Tropho_progenitor_2, which both show marginally significant p-values of 0.04 in the comparison.

Extended Data Fig. 2. Proportion of male and female cells in each cluster, along with p-values comparing male to female proportions within a cell type.

10. Results, I think this part of the results is mainly focused on Figure 1, but still please cite individual figure panels for each statement.

Response: Thank you for highlighting this concern. We recognize the importance of aiding readers' understanding and have now provided citations to figures, extended data, and supplementary files for each relevant statement (Lines 138-197).

11. Results, Please dampen this statement, as it is unlikely to be the first time the murine placental cell clusters are elucidated at E17.5.

Response: We agree with the reviewer's suggestion. While we did not find a single-cell survey of the mouse placenta at E17.5 specifically, there are available studies at a similar gestational period (e.g. E19)¹⁶. We have revised the statement to remove potential misinformation or overinterpretation (Line 198).

12. Results, Figure 2 and Extended Data: Did the authors perform protein validation for any of the key DEGs found to be specific to As exposure, besides PRAP1?

Response: We acknowledge the importance of reinforcing our scRNA-seq findings through additional protein validation of differentially expressed (DE) genes. In response to the reviewer's concern, we conducted immunofluorescence (IF) staining for CAR2 and GUCA2B (Extended Data Fig. 7 and Extended Data Fig. 9). These proteins were deliberately chosen due to their lack of prior association with As exposure and limited understanding of their functions in the placenta.

Extended Data Fig. 7. Immunofluorescence staining of CAR2 in mouse placenta. Scale bars represent 100 μ m. Dec: Decidua; JZ: Junctional Zone.

CAR2 exhibited a upregulation in GlyT cells in the mouse placenta following As exposure (Supplementary Table S2; Extended Data Fig. 7). As noted in previous studies^{17,18}, CAR2 expression was predominantly localized in the junctional zone and decidua of mouse placentae, precisely where GlyT cells reside (Extended Data Fig. 7).

Extended Data Fig. 9. Immunofluorescence staining of GUCA2B in mouse placenta. Scale bars represent 100 μm. Dec: Decidua; 2. JZ: Junctional Zone.

The gene *Guca2b*, encoding Guanylate Cyclase Activator 2B (GUCA2B), was upregulated in 22 cell types, albeit its expression was confined to a small fraction of cells (Fig. 2 and Extended Data Fig. 8b). IF staining of GUCA2B protein corroborated this, revealing sparse expression in mouse placentae, making differential expression determination challenging (Extended Data Fig. 9). Co-staining with Ly6G, a neutrophil marker, suggested that a considerable portion of GUCA2B expression originates from neutrophils (Extended Data Fig. 9).

Furthermore, the significant upregulation of additional DE genes, such as *APOA2*, *APOB*, *AFP*, *CA2*, *GPX*, and *GUCA2B* (Extended Data Fig. 6-9; Supplementary Table S2 and S3), identified through scRNA-seq analysis in mouse placenta, has been

validated by qPCR in human trophoblast cells (Extended Data Fig. 13e and 13f). The importance of investigating the upregulation of these genes in the placenta is evident and warrants further exploration.

Extended Data Fig. 13. Validation of *Prap1* expression in human placental cells and tissues Validation of additional DE genes in (e) BeWo and (f) HTR-8/SVneo cells treated with As (n=3). *p<0.05, significant compared to control (0 μM As). #p<0.05, significantly different from each other.

13. Results, Lines 272-274: Please be more specific regarding the cell types involved in these processes.

Response: Thank you for the suggestion. We have now included additional statements regarding specific cell types (Lines 303-308)

14. Results, Lines 310-327: Did the authors perform any semi-quantification from the IF imaging? Or other immunoassays to detect protein levels?

Response: Thank you for the suggestion. In response, we conducted semi-quantification of PRAP1 expression from IF images, calculating the PRAP1 area (%) normalized to the total placental area. Semi-quantification of PRAP1 expression from IF images reveals significantly higher expression in As_Female compared to As_Male (Extended Data Fig. 12).

Extended Data Fig. 12. Semi-quantification of PRAP1 expression in the mouse placenta.

15. Results,: How comparable are the human elective termination samples to the late pregnancy (E17.5) murine samples in terms of developmental stage?

Response: The human placentas collected spanned the first (6-13 weeks; n=2) and second trimesters (14-22 weeks; n=3), representing early to mid-pregnancy. In contrast, the mouse placentae used (E17.5) are fully matured and can be considered analogous to human placenta at 35 to 40 weeks. However, due to logistical constraints, selective late-pregnancy collection was not feasible. Additionally, the exposure duration differed, with mouse placentas exposed throughout pregnancy and human placenta explants treated for 24 h, limiting direct comparability. These limitations are acknowledged in the discussion (Lines 506-510).

16. Results, Lines 348-356: How was the PRAP1 concentration chosen?

Response: We were not able to find existing studies utilizing recombinant PRAP1 protein in a comparable experimental setup. Therefore, we conducted concentration-dependent response experiments in human placental cells based on studies involving other secretory proteins.

17. Methods: For single-cell analysis, please clarify the datasets that were used as reference for cell type annotation.

Response: We apologize for not providing sufficient detail for this important step. We did not rely on a single gold-standard reference for cell type annotation, but rather aggregated signature marker genes from a variety of sources. These included several manuscripts^{12-14,19-27} and the Panglao Database. This information is included in the text (Lines 122-123, 662-664).

18. Methods: Did the authors consider using primary placental cells for the in vitro studies? Also, how comparable is the acute (24 h) As exposure used for in vitro studies to the chronic exposure performed in mice?

Response: Thank you for raising this point. Recognizing the inherent variability in results from human villous explant cultures, we acknowledge that primary placental cells may provide a more homogeneous cell population. However, the limited availability of human placentae adds complexity to experimental setup challenges, particularly in isolating primary cells. Additionally, we concur that the 24-hour exposure may not accurately mirror chronic exposure during pregnancy. It's important to note that even with primary placental cells, the scope for exposure would still be confined to short-term durations.

To address these challenges, we conducted in vitro experiments utilizing human trophoblast cell lines (BeWo and HTR-8/SVneo), exposing them to As (0, 1, or 2 μ M) for 7 days (Extended Data Fig. 13a and 13 b). Regrettably, after the 7-day exposure, most BeWo cells succumbed to the conditions, precluding the provision of data from this cell line. Extended Data Fig. 12a demonstrates a significant increase in *PRAP1* mRNA expression in the cells. However, PRAP1 protein was not detected, and β -actin expression decreased at 2 μ M As, suggesting increased cytotoxicity with chronic As exposure or compensatory mRNA upregulation to augment PRAP1 protein (Extended Data Fig. 13a and 13b).

Extended Data Fig. 13. Validation of *Prap1* expression in human placental cells and tissues. (a and b) Effect of chronic As treatment on PRAP1 expression in HTR-8/SVneo cells (n=3). *p<0.05, significant compared to control (0 μM As). #p<0.05, significantly different from each other.

19. Methods: Prizm should be Prism.

Response: We appreciate your attention to this. The typographical error has been rectified (Line 812).

20. Methods: Single-cell data should be deposited and publicly available at publication.

Response: We agree that this is an important step for transparency. Single-cell RNA-sequencing data used in this study, including the final Seurat data object used in analyses and to generate figures, are available on Zenodo at <https://zenodo.org/records/10258020>. On a related note, we also deposited the code used to reproduce the final Seurat data object and make all figures used in the manuscript at https://github.com/edvanburen/placenta_code and noted this fact under the section titled “Code availability.”. (Lines 842-851)

Reviewer #2

1. The authors determine if As exposure has changed the relative proportion of any placental cell types (it did not), identify differentially expressed genes (DEGs) in placental trophoblasts and non-trophoblast cells (there were only 7).

Response: We appreciate your attention to the data, and we apologize for any confusion that may have arisen. It's important to clarify that **our study identified a more extensive list of differentially expressed (DE) genes than the seven** highlighted in Fig. 2. For a detailed understanding of how prenatal As exposure impacts the mouse placental transcriptome, we identified DE genes for each cell type (Padj<0.05), and the full list is available in Supplementary Excel File 2. Furthermore, Supplementary Table S3 highlights DE genes unique to specific cell types (Padj<0.05, FC >1.25 or FC <0.75).

The heatmap in Fig. 2 represents a subset of DE genes, including *Prap1*, *Ctla2A*, *Guca2B*, *Gpx3*, *Apob*, *Apoa2*, and *Afp*, chosen for visualization purposes. We recognize that the focus on these seven genes might have given the impression that only a limited number of DE genes were identified, and we regret any misunderstanding.

Our chosen dose of 200 ppb for As in this study aligns with both human exposure ranges (10-1000 ppb) and rodent study doses (50-5000 ppb)^{2,3}. Despite its seemingly moderate level, this dose has been associated with adverse pregnancy outcomes and neurocognitive deficits in both adults and children⁴. Although we did not observe variations in placental weight, fetal malformation, or resorption rate at this moderate As concentration (Extended Data Fig. 1), scRNA-seq revealed unique DE genes within individual cell types and common DE genes across multiple cell types in response to As exposure. The study's high sensitivity in detecting modest but significant changes in gene expression within each cell type highlights the advantages of scRNA-seq, particularly in capturing nuances that might be missed with bulk RNA-Seq due to low cell numbers or modest fold changes. Additionally, the identification of novel genes differentially regulated by As, such as *Car2*, *Prap1*, *Ctla2a*, and *Guca2b*, adds valuable insights to the understanding of placental responses to As exposure.

2. They also perform pathway analysis for selected trophoblast cell types, and using between 4-9 genes for each cell type (not clear if these are DEGs), identify enriched pathways, although it is not clear, despite the statistical data provided, that with so few genes to consider here has been any true pathway disruption with biological consequence.

Response: We apologize for being unclear in how our pathway analysis was performed and have updated the text to give more detail about our approach (Lines 672-677). Our analysis does require making the assumption that the genes used for GO term analysis are representative of the pathway as a whole. For example, a GO term with 50 genes total may only have 8 genes used in our GO term analyses. The remaining 42 genes would not present in the analysis because of either (1) the scRNA-seq data pre-processing steps which select only highly variable and expressed genes for analysis or (2) the DE filtering criteria used in the GO term analysis method of the clusterProfiler R

package. Unlike analyses of bulk sequencing data, step (1) follows common convention for single-cell RNA-seq data analysis and involves removing genes with sparse expression (from “drop-out” effects) from all downstream analyses. The overall goal of our scRNA-seq processing steps and analysis is to follow such common convention but still perform biologically meaningful analyses.

3. They perform in vitro experiments to validate changes in PRAP1 in response to As exposure, and see from 1.5-3 fold (i.e. marginal at best) changes in PRAP1 expression in all lines except HTR/8-SVNeo cells, where the increase is 4-fold. The sex-specific differences noted in many of these analyses are interesting, but diminished by the few genes used to ascertain these differences, and marginal magnitude of differential expression in response to As.

Response: We acknowledge the observed modest fold change in PRAP1 expression in human placental cell lines exposed to As. To gain deeper insights into the functional implications of PRAP1 expression, we conducted additional *in vitro* experiments.

Initially, we demonstrated that treatment with recombinant PRAP1 protein significantly reduced As-induced cytotoxicity *in vitro* (Fig. 6e). To explore the impact of PRAP1 on placental cells further, we performed global transcriptomic profiling in the human placental trophoblast cell line BeWo treated with recombinant human PRAP1 protein (Extended Data Fig. 14a). Subsequently, we conducted gene ontology (GO) analysis, revealing that the majority of significantly upregulated pathways were associated with transmembrane transport and signal transduction upon treatment with exogenous PRAP1 protein (Extended Data Fig. 14b, top panel). Intriguingly, pathways related to cell cycle regulation, G2/M transition, and mitosis were significantly downregulated (Extended Data Fig. 14b, bottom panel). The downregulation of specific cell cycle-related genes, including *CENPF*, *CDK1*, *CDK7*, and *PLK1*, further supported these findings (Supplementary Table S4; Supplementary Excel File 6).

Continuing our investigation, we performed cell cycle analysis in non-targeting siRNA (Neg) or anti-*PRAP1* siRNA (si-*PRAP1*) cells exposed to control (NT) or 2 μ M As for 24 hr. Consistent with previous studies indicating As treatment blocks the G1 to S transition, the percentages of As-treated cells in S and G2 were lower than in Neg/NT cells (Extended Data Fig. 14c). Conversely, the number of cells in G2 was higher in si-*PRAP1*/NT cells than in Neg/NT cells, aligning with reduced expression of G2 to M checkpoint genes in cells treated with PRAP1 protein (Extended Data Fig. 14b; Supplementary Table S4; Supplementary Excel File 6). The number of cells in G2 in si-*PRAP1*/As cells decreased compared to si-*PRAP1*/NT cells, confirming the impact of As on G2 decrease (Extended Data Fig. 14c). Extended Data Fig. 14d illustrates the histogram comparing the four groups.

Expanding on these findings, we delved into the role of PRAP1 in regulating cell proliferation and cell death in placental trophoblast cells *in vitro*, both in the absence and presence of As. In Fig. 6f, we observed that *PRAP1* knockdown significantly reduced the cell number compared to the negative control, measured by the Cyquant

assay. Furthermore, *PRAP1* knockdown exacerbated the As-induced reduction in cell number (Fig. 6f). Fig. 6g revealed that *PRAP1* knockdown significantly increased cell proliferation, potentially explaining the observed increase in the G2 phase with *PRAP1*. Conversely, As treatment led to a significant reduction in cell proliferation (Fig. 6g), aligning with the decreased S and G2 phases observed with As (Extended Data Fig. 13c and 13d). Intriguingly, *PRAP1* knockdown increased cell death, measured by PI staining, in both the absence and presence of As (Fig. 6h), potentially elucidating the overall reduction in cell number (Fig. 6f), despite the observed increase in cell proliferation (Fig. 6g).

In summary, these data suggest the potential involvement of *PRAP1* in cell cycle processes, cell proliferation, and cell death. These findings may contribute to altered placental cell function, influencing placental responses to environmental stress, such as exposure to As.

Fig. 6. Validation of *Prap1* expression in human placental cells and tissues. Expression of *Prap1* in human placental cell lines (BeWo, HTR-8/SVneo, and SWAN71) exposed to As measured by (a-c) qRT-PCR (normalized to GAPDH) or (d) western blotting. n=3. *p<0.05, significant compared to control (0 μM As). #p<0.05, significantly different from each other. (e) Effect of *PRAP1* on LDH release in BeWo cells with or without As

treatment (n=3). Effect of *PRAP1* knockdown on (f) cell number measured by Cyquant assay, (g) cell proliferation measured by EdU incorporation, (h) cell death measured by PI staining in BeWo cells (n=>10). *p<0.05, significant compared to control (Negative control/0 μ M). #p<0.05, significantly different from each other.

Extended Data Fig. 14. (a) Schematic workflow of RNA-seq and GO analysis on BeWo cells treated with PRAP1 protein. Created by BioRender.com. (b) Significantly enriched pathways with PRAP1 treatment. (c-d) Cell cycle analysis in control or *PRAP1* knockdown cells treated with 0 (NT) or 2 μ M As for 24 hr.

Reviewer #3

1. I suggest mentioning specific cell types affected by As exposure in the abstract.

Response: We agree with the reviewer's suggestion. We have revised the abstract accordingly and provided specific cell types (Lines 36-40).

2. Throughout the results, the authors are very repetitive with the wording they use in their concluding sentences. They use "insights" and "mechanistic insights" too many times. These words are very general and do not convey substantive conclusions. Same is true for the word "revealed" in the discussion.

Response: We appreciate the reviewer's feedback. While we made efforts to avoid overinterpretation of our findings, we recognize that this cautious approach may have affected the strength of our conclusions. In response, we have revised the text accordingly (Lines 339-342, 405-407, 475-478).

3. Going along with my previous comment, the results section has quite a bit of interpretation. For example, the summary paragraph in lines 252-264 has information in it that I would think is more appropriate in the discussion (ie discussing As dose).

Response: We agree with the reviewer's comment. In response, we have removed unnecessary interpretation and information from the results section, incorporating them appropriately into the discussion (Lines 483-494).

4. What cell type(s) had differentially expressed PRAP1?

Response: As mentioned above, we have revised the abstract accordingly and provided specific cell types (Lines 36-40).

5. Sentence that starts with "For example, exposure to..."-I recommend inserting relevant citations into the sentence specifically where they are discussed, instead of all at the end of the sentence. The sentence further resembles multiple lists. I think the sentence could benefit from re-structuring or splitting into a few sentences.

Response: We appreciate the reviewer's insightful suggestions. We have incorporated specific citations to bolster our claims and restructured sentences for improved clarity (Lines 64-74)

6. Consider mentioning study by Tosevska et al which did single-cell analysis on mouse placenta exposed to PM.

Response: Thank you for raising this point. We have incorporated the mentioned study into our introduction and adjusted our statement accordingly (Lines 85-87).

7. The description given in this paragraph of clustering methods belongs in a figure caption or methods section. It's not really describing results. Consider removing it.

Response: We agree with the reviewer. We have removed the information on R packages and clustering methods (Lines 122-125).

8. I recommend denoting whether cell types being reference are maternal or fetal.

Response: Thank you for the suggestion. To address this issue, we conducted additional analysis on *Xist* expression in male placentae based on Marsh and Blelloch (2020)¹⁴. Our findings suggest that most immune cells consist of both maternal and fetal cells. Specifically, the majority of Macrophage_2, B Cell, T_Cell, Dendritic, Basophil, and NK_Cell_1 cells were of maternal origin (Extended Data Fig. 5; Supplementary Table S1). Conversely, Neutrophil_1, Neutrophil_2, Macrophage_1, Megakaryocyte, and NK_Cell_2 were predominantly of fetal origin (Extended Data Fig. 5; Supplementary Table S1). This new analysis has prompted us to reclassify the cell cluster labeled as 'Hofbauer' in the initial submission to 'Macrophage_2' in the resubmission due to the observed *Xist* expression in this cluster. Although Macrophage_1 is speculated to be Hofbauer cells^{13,27}, further validation is needed. This information is included in the text (Lines 184-197).

Extended Data Fig. 5. Expression of *Xist* in male placentae. Male placental samples express *Xist* only in maternal cells.

Supplementary Table S1. The proportion of cells expressing *Xist* in each cell type

cell_type	prop_Xist
Macrophage_2	0.85
B_Cell	0.77
T_Cell	0.77
Dendritic	0.73
Basophil	0.69
NK_Cell_1	0.56
Neutrophil_2	0.37
Neutrophil_1	0.19
Stromal_1	0.15
Decidual_2	0.04
Erythroid_Precursor_2	0.04
Erythroblast	0.04
Decidual_1	0.03
Labyr_Tropho_1	0.02
Macrophage_1	0.02
Spong_1	0.02
Tropho_Progenitor_2	0.01
Erythroid_Precursor_1	0.01
Erythroid_Precursor_3	0.01
Yolk_Sack	0.01
Spong_2	0.01
Endothelial_1	0.01
Endothelial_2	0.01
Endodermal_Cell	0.01
Labyr_Tropho_2	0.01
S-TGC_1	0.01
Megakaryocyte	0.01
Tropho_Progenitor_1	0.01
Pericyte	0.01
GlyT	0
Fibroblast	0
Labyr_Tropho_3	0
S-TGC_2	0
Pericyte_2	0
Stromal_2	0
NK_Cell_2	0

9. In this sentence, the authors define genes by name and symbol. In earlier paragraphs, the authors only use gene symbols. I would recommend keeping this consistent throughout the paper.

Response: Thank you for pointing this out. We have revised the sentence to keep the consistency (Line 235).

10. needs a reference.

Response: Thank you for pointing this out. Now we have provided relevant references to this statement (Line 237, #66-71).

11. Why is this finding surprising?

Response: We acknowledge the need for clarification regarding this statement. In addition to its role in glucose transport, the placenta metabolizes glucose for internal use and stores it as glycogen, a multi-branched polysaccharide. The purpose of placental glycogen stores in normal pregnancy remains elusive, with the prevailing theory suggesting that these stores ensure a consistent fetal glucose supply during periods of heightened demand, particularly in late gestation^{28,29}. Although GlyT's involvement in glycogen storage is acknowledged, their precise roles remain incompletely understood. Our GO analysis shows that GlyT cells exhibit a significant enrichment in terms associated with protein translation, ribosome biogenesis, and ribosomal RNA (rRNA) processing (Extended Data Fig. 10), surpassing our initial expectations. This enrichment implies the crucial importance of GlyT cells in processes related to cellular growth and development, particularly during the late stages of pregnancy. We have revised the paragraph to improve the clarity (Lines 312-317).

12. You can't make conclusions about sex-specific responses based on cell lines because they are different cells. Please remove conclusion about sex-specific effects in cell lines.

Response: We agree with the reviewer's comment. Now we have removed the statement (Lines 378)

13. I'm assuming you did bulk RNAseq on human villous explant tissue. It seems regressive to use bulk when you've used single-cell or cell lines. Plus, these results didn't confirm any of your other findings. I would consider removing this paragraph completely.

Response: We appreciate the reviewer's suggestion. It's important to clarify that we did not conduct bulk RNA-seq on human villous explants. Instead, we treated the explant cultures with As for 24 hours and measured *PRAP1* expression using qPT-PCR. The inherent variability in results from human villous explant cultures poses challenges for drawing definitive conclusions (Lines 385-390; 506-510). The data has been relocated to Extended Data Fig. 13c and 13d, with new *in vitro* findings now included in Fig 6.

Fig. 6. Validation of *Prap1* expression in human placental cells and tissues. Expression of *Prap1* in human placental cell lines (BeWo, HTR-8/SVneo, and SWAN71) exposed to As measured by (a-c) qRT-PCR (normalized to GAPDH) or (d) western blotting. $n=3$. * $p<0.05$, significant compared to control (0 μ M As). # $p<0.05$, significantly different from each other. (e) Effect of PRAP1 on LDH release in BeWo cells with or without As treatment ($n=3$). Effect of *PRAP1* knockdown on (f) cell number measured by Cyquant assay, (g) cell proliferation measured by EdU incorporation, (h) cell death measured by PI staining in BeWo cells ($n>10$). * $p<0.05$, significant compared to control (Negative control/0 μ M). # $p<0.05$, significantly different from each other.

Extended Data Fig. 13. Validation of *Prap1* expression in human placental cells and tissues. (a and b) Effect of chronic As treatment on PRAP1 expression in HTR-8/SVneo cells (n=3). (c and d) *Prap1* expression human placental villous explants exposed to As by trimester and by sex. (n= 5 placentae). Validation of additional DE genes in (e) BeWo and (f) HTR-8/SVneo cells treated with As (n=3). *p<0.05, significant compared to control (0 μM As). #p<0.05, significantly different from each other.

14. Line 418. Sentence “We for the first time...” should either have commas (We, for the first time,...) or should read (For the first time, we...) Please revise the sentence.

Response: Thank you for raising this point. We have revised the sentence accordingly (Line 534).

- 1 Xiong, G. F. *et al.* Estradiol-regulated proline-rich acid protein 1 is repressed by class I histone deacetylase and functions in peri-implantation mouse uterus. *Mol Cell Endocrinol* **331**, 23-33 (2011). <https://doi.org:10.1016/j.mce.2010.06.003>
- 2 Podgorski, J. & Berg, M. Global threat of arsenic in groundwater. *Science* **368**, 845-850 (2020). <https://doi.org:10.1126/science.aba1510>
- 3 Murko, M., Elek, B., Styblo, M., Thomas, D. J. & Francesconi, K. A. Dose and Diet - Sources of Arsenic Intake in Mouse in Utero Exposure Scenarios. *Chem Res Toxicol* **31**, 156-164 (2018). <https://doi.org:10.1021/acs.chemrestox.7b00309>
- 4 Tyler, C. R. & Allan, A. M. The Effects of Arsenic Exposure on Neurological and Cognitive Dysfunction in Human and Rodent Studies: A Review. *Curr Environ Health Rep* **1**, 132-147 (2014). <https://doi.org:10.1007/s40572-014-0012-1>
- 5 Quansah, R. *et al.* Association of arsenic with adverse pregnancy outcomes/infant mortality: a systematic review and meta-analysis. *Environ Health Perspect* **123**, 412-421 (2015). <https://doi.org:10.1289/ehp.1307894>
- 6 Calderon, J. *et al.* Exposure to arsenic and lead and neuropsychological development in Mexican children. *Environ Res* **85**, 69-76 (2001). <https://doi.org:10.1006/enrs.2000.4106>
- 7 von Ehrenstein, O. S. *et al.* Children's intellectual function in relation to arsenic exposure. *Epidemiology* **18**, 44-51 (2007). <https://doi.org:10.1097/01.ede.0000248900.65613.a9>
- 8 Sen, D. & Sarathi Biswas, P. Arsenicosis: Is it a Protective or Predisposing Factor for Mental Illness? *Iran J Psychiatry* **7**, 180-183 (2012).
- 9 Gandhi, D. N., Panchal, G. M. & Patel, K. G. Developmental and neurobehavioural toxicity study of arsenic on rats following gestational exposure. *Indian J Exp Biol* **50**, 147-155 (2012).
- 10 Luo, J. *et al.* Maternal and early life arsenite exposure impairs neurodevelopment and increases the expression of PSA-NCAM in hippocampus of rat offspring. *Toxicology* **311**, 99-106 (2013). <https://doi.org:10.1016/j.tox.2013.06.007>
- 11 Martinez-Finley, E. J., Ali, A. M. & Allan, A. M. Learning deficits in C57BL/6J mice following perinatal arsenic exposure: consequence of lower corticosterone receptor levels? *Pharmacol Biochem Behav* **94**, 271-277 (2009). <https://doi.org:10.1016/j.pbb.2009.09.006>
- 12 Nelson, A. C., Mould, A. W., Bikoff, E. K. & Robertson, E. J. Single-cell RNA-seq reveals cell type-specific transcriptional signatures at the maternal-foetal interface during pregnancy. *Nat Commun* **7**, 11414 (2016). <https://doi.org:10.1038/ncomms11414>
- 13 Zhou, X. *et al.* Single-cell RNA-seq revealed diverse cell types in the mouse placenta at mid-gestation. *Exp Cell Res* **405**, 112715 (2021). <https://doi.org:10.1016/j.yexcr.2021.112715>
- 14 Marsh, B. & Blelloch, R. Single nuclei RNA-seq of mouse placental labyrinth development. *Elife* **9** (2020). <https://doi.org:10.7554/eLife.60266>
- 15 Soncin, F., Natale, D. & Parast, M. M. Signaling pathways in mouse and human trophoblast differentiation: a comparative review. *Cell Mol Life Sci* **72**, 1291-1302 (2015). <https://doi.org:10.1007/s00018-014-1794-x>

- 16 Tosevska, A. *et al.* Integrated analysis of an in vivo model of intra-nasal exposure to instilled air pollutants reveals cell-type specific responses in the placenta. *Sci Rep* **12**, 8438 (2022). <https://doi.org:10.1038/s41598-022-12340-z>
- 17 Adamson, S. L. *et al.* Interactions between trophoblast cells and the maternal and fetal circulation in the mouse placenta. *Dev Biol* **250**, 358-373 (2002). [https://doi.org:10.1016/s0012-1606\(02\)90773-6](https://doi.org:10.1016/s0012-1606(02)90773-6)
- 18 Singh, U. *et al.* Expression and functional analysis of genes deregulated in mouse placental overgrowth models: Car2 and Ncam1. *Dev Dyn* **234**, 1034-1045 (2005). <https://doi.org:10.1002/dvdy.20597>
- 19 Han, X. *et al.* Mapping the Mouse Cell Atlas by Microwell-Seq. *Cell* **173**, 1307 (2018). <https://doi.org:10.1016/j.cell.2018.05.012>
- 20 Franzen, O., Gan, L. M. & Bjorkegren, J. L. M. PanglaoDB: a web server for exploration of mouse and human single-cell RNA sequencing data. *Database (Oxford)* **2019** (2019). <https://doi.org:10.1093/database/baz046>
- 21 Home, P., Ghosh, A., Kumar, R. P., Ganguly, A., Bhattacharya, B. Trophoblast paracrine signaling regulates placental hematoendothelial niche. *bioRxiv* (2019).
- 22 Walentin, K., Hinze, C. & Schmidt-Ott, K. M. The basal chorionic trophoblast cell layer: An emerging coordinator of placenta development. *Bioessays* **38**, 254-265 (2016). <https://doi.org:10.1002/bies.201500087>
- 23 Liu, Y. *et al.* Single-cell RNA-seq reveals the diversity of trophoblast subtypes and patterns of differentiation in the human placenta. *Cell Res* **28**, 819-832 (2018). <https://doi.org:10.1038/s41422-018-0066-y>
- 24 Simmons, D. G., Fortier, A. L. & Cross, J. C. Diverse subtypes and developmental origins of trophoblast giant cells in the mouse placenta. *Dev Biol* **304**, 567-578 (2007). <https://doi.org:10.1016/j.ydbio.2007.01.009>
- 25 Simmons, D. G. *et al.* Early patterning of the chorion leads to the trilaminar trophoblast cell structure in the placental labyrinth. *Development* **135**, 2083-2091 (2008). <https://doi.org:10.1242/dev.020099>
- 26 He, J. P., Tian, Q., Zhu, Q. Y. & Liu, J. L. Identification of Intercellular Crosstalk between Decidual Cells and Niche Cells in Mice. *Int J Mol Sci* **22** (2021). <https://doi.org:10.3390/ijms22147696>
- 27 Karlsson, M. *et al.* A single-cell type transcriptomics map of human tissues. *Sci Adv* **7** (2021). <https://doi.org:10.1126/sciadv.abh2169>
- 28 Barash, V. & Shafrir, E. Mobilization of placental glycogen in diabetic rats. *Placenta* **11**, 515-521 (1990). [https://doi.org:10.1016/s0143-4004\(05\)80197-3](https://doi.org:10.1016/s0143-4004(05)80197-3)
- 29 Coan, P. M., Conroy, N., Burton, G. J. & Ferguson-Smith, A. C. Origin and characteristics of glycogen cells in the developing murine placenta. *Dev Dyn* **235**, 3280-3294 (2006). <https://doi.org:10.1002/dvdy.20981>

REVIEWER COMMENTS

Reviewer #1 (Remarks to the Author):

The study's noteworthy aspect lies in its pioneering exploration of the effects of arsenic in placental tissues, marking the first instance of such investigation. This work holds significance not only within its field but also in related areas. While acknowledging that other research has extensively covered late gestation placental studies (our group, so I prefer not to provide references), this research stands out due to its focus on toxic substances. The validation experiments effectively complement the transcriptomics findings, affirming the conclusions and claims made. There appear to be no flaws in data analysis, interpretation, or conclusions, as the revised paper is impressive. Methodologically, the study adheres to expected standards and offers sufficient detail for reproducibility, ensuring a robust contribution to the field.

Reviewer #3 (Remarks to the Author):

I am happy with the authors' responses and don't require more edits.

Reviewer #4 (Remarks to the Author):

Van Buren et al presents their work on the impact of As on placental single-cell transcriptomics using mice with some attempts to translate the findings to human health. The work is very limited in scope with major concerns about power and rigor and overall lack proper validation strategies. The analytical framework is also very simplistic and there are key experimental details missing. The concerns are further articulated below:

1. Single-cell analysis of the human and mouse placenta have been extensively studied, yet the authors present a complete and comprehensive paragraph about these findings. Instead, they should contrast (validate) their findings with other available data to convince the reader that their baseline data is solid for instance using maps published in PMID: 36928215 or those cited by authors.

2. To put results into a larger and more relevant disease context, the authors should report the phenotypes of the offspring of those mice exposed to As.

3. The authors need to validate their findings using human studies linking As with placental transcriptomics to ensure the findings reported here is relevant given known differences between human and mouse placenta (PMID: 35646058; PMID: 31752878; PMID: 26288817 etc.)
4. The dose of 200 ppb was selected for the main experiment (mice) without further dose-response analysis and with rather unclear justification lacking references. This is important as the effect of As on several genes were unspecific and significant across multiple cell types perhaps indicating a toxic, non-relevant dose.
5. The findings about how As impacts gene expression in placental cell population is presented very broadly listing genes without any orthogonal validations. Only PRAP1 is validated in human cell lines. More thorough validation is needed.
6. As mentioned above only PRAP1 is validated focusing on a single cell type only (trophoblast cells lines HTR-8/SVneo, BeWo, and Swan 71) despite the effect is seen across most cell types captured with the rationale that “trophoblasts are the predominant cell type in the placenta”. However, based on their figure 1b, trophoblast stem cells (assuming what these cell lines are used to mimic) appears not to be the predominant cell type thus their validation strategy isn’t valid. Additional cell types should be used for validation and for TS cells more accurate lines should be considered with potential to differentiation (PMID: 29249463).
7. The authors make a statement that the As effect seen on PRAP1 is sex-specific but yet fail to validate these findings in their human in vitro models. They completely leave out the sex of the cell lines used and refer to inconclusive results from the in situ experiments. This is unacceptable and needs to be expanded to validate their sex-specific effect. As mentioned above the TS models (PMID: 29249463) can be obtained or generated based on fetal sex.
8. It remains unclear how many placentas were analyzed in their discovery cohort of mouse placenta. It is described in later sections that 8 samples were analyzed but not clear if this is referring to biological or technical replicates. Extended figure 1 shows only N=2 per group raising questions about female vs male placenta with potentially only N=1. This low number is concerning for many reasons including lack of power for any sex-specific effects.
9. Similarly, the single-cell experiment is not described in sufficient detail including how many cells were loaded per channel and how potential duplets were handled.
10. Related to concern raised in 4) figure 1 is not showing how cell distributions (and expression) are observed in individual samples as only a single integrated dataset is shown. Similarly, feature plots how libraries are distributed is warranted to ensure no technical effects.

Manuscript ID: NCOMMS-23-38005A

We sincerely thank the scientists who reviewed this manuscript. We have dedicated our best efforts to comprehensively address the reviewers' valuable comments, which have significantly contributed to the refinement of the article. The clarifications as well as new data have been incorporated into the manuscript, both in the text and figures, as cited below. We feel that the manuscript is improved by these revisions and hope that it will be found acceptable for publication by *Nature Communications*.

<Reviewer #4>

1. Single-cell analysis of the human and mouse placenta have been extensively studied, yet the authors present a complete and comprehensive paragraph about these findings.

Response: We agree with the reviewer's suggestion. Now we provide more elaborated discussion on existing studies and the new contribution of our study to the field in the text (Lines 83-96; 204-205; 295-301)

Previous studies using scRNA-seq or microwell-seq have mapped placental cell types in both mice¹⁻⁷ and humans⁸⁻¹³. For example, one study conducted scRNA-seq analysis of placental cells of fetal and maternal origin in mouse placentae at E9.5², while another focused on E10.5, tracing the developmental trajectories of trophoblasts and highlighting signaling pathways that mediate crosstalk between cell types³. In addition, a different research group profiled single-cell transcriptomes of trophoblasts from E9.5 to E14.5 throughout placental development⁶. Furthermore, scRNA-seq has been applied to characterize human placental cell types from the first trimester^{8,9} to term^{10,11,13}. Tosevska *et al.*¹ reported cell-type-specific responses in the three layers (decidua, junctional zone, and labyrinth) of the mouse placenta following exposure to particulate matter. While these studies have yielded valuable insights into placental biology, our understanding of placental transcriptomic responses to environmental toxicant exposure at the single-cell level is still in its early stages.

In this work, we build upon previous studies by profiling the single-cell transcriptomes of late-term (E17.5) murine placental cells, both in the presence and absence of environmental arsenic (As) exposure. We utilized general marker genes and those reported from earlier studies^{2-5,7,14-20} for cell cluster analysis. Additionally, our study identified potential novel cell markers such as *Car2* (for GlyT cells) and *Fnd3c2*, *Nup62cl*, and *Fdx1* (for S-TGCs). We also detected unique differentially expressed (DE) genes in individual cell types and common DE genes across multiple cell types in response to As exposure. Moreover, our research has uncovered novel genes (*Car2*, *Prap1*, *Ctla2a*, and *Guca2b*) previously not linked to As exposure, with unclear functions in the placenta. Lastly, we observed sex-dependent differences in the response to As exposure, which may shed light on the development of sex-dependent pathologies from As exposure during pregnancy²¹⁻²³.

2. To put results into a larger and more relevant disease context, the authors should report the phenotypes of the offspring of those mice exposed to As.

Response: We appreciate the reviewer's perspective. As mentioned in the text (Lines 125-127), we did not observe any variations in fetal malformation, or the rate of resorption or preterm delivery due to exposure to As. The selected dose of 200 ppb, while not overly toxic in previous rodent studies, is still associated with adverse neurodevelopmental outcomes in offspring²⁴⁻²⁶ (Lines 121-125). Based on our findings that the functions of trophoblasts, including proliferation and cell death, are influenced by As exposure and differentially regulated with PRAP1, it raises an intriguing question of whether such alterations may result in phenotypic changes *in vivo*, such as neurocognitive or behavioral outcomes. While conducting *in vivo* studies falls beyond the scope of this manuscript, we intend to explore this avenue in future research endeavors. The discussion in this matter has been incorporated in the text (Lines 588-598)

3. The authors need to validate their findings using human studies linking As with placental transcriptomics to ensure the findings reported here is relevant given known differences between human and mouse placenta (PMID: 35646058; PMID: 31752878; PMID: 26288817 etc.)

Response: We acknowledge the importance of validating our findings against human studies. Our discussion indeed references human studies on As exposure effects on placental gene dysregulation and health outcomes (Lines 485-500; 524-542), including the works mentioned above.

Our study extends the existing literature by leveraging single-cell RNA sequencing (scRNA-seq) to uncover differential expression (DE) of genes across individual and common cell types within the placenta in response to As exposure. This approach has enabled us to detect nuanced changes in gene expression that are often obscured in bulk RNA sequencing due to the dilution effect from heterogeneous cell populations or subtle fold changes²⁷. Notably, our study has identified novel genes (*Car2*, *Prap1*, *Ctla2a*, *Guca2b*) not previously linked to As exposure and has provided insights into their potential roles in the placenta. We also emphasize the variable expression of DE genes among different cell types and treatment groups, underscoring the heightened sensitivity of our methodological approach.

Furthermore, we've highlighted the sex-specific impacts of As exposure on placental gene regulation, a critical determinant of pregnancy outcomes. Studies have consistently shown that As exposure has differential impacts on placental gene regulation and pregnancy outcomes based on sex^{21-23,28-35}. For example, transcriptomic analyses, like those from the New Hampshire Birth Cohort Study, indicate that As exposure affects female and male placentas differently, with implications for birth weight. Specifically, female placentas exhibited gene expression patterns linked to lower birth weights, a trend not seen in male placentas²⁸. Moreover, the GLI family zinc finger 3 gene inversely correlates with As exposure in females but is positively linked to birth weight³⁰. Additionally, As impacts epigenetic regulators in a sex-specific manner, influencing birth outcomes, such as weight and gestational age differently in male and female infants³¹. As in maternal blood or drinking water was associated with decreased birth weight and gestational age, smaller ponderal index in male babies^{32,33}. On the

other hand, only girls, particularly those born to overweight or obese mothers, showed a significant decrease in birth weight with As exposure³⁴. This suggests a complex interplay between maternal factors, fetal sex, and As exposure influencing pregnancy outcomes. Although these studies and our findings suggest that PRAP1 may play a role in the development of sex-dependent pathologies resulting from exposure to As during pregnancy²¹⁻²³, *in vivo* studies investigating the impact of PRAP1 disruption on pregnancy and developmental outcomes, both with or without As exposure, will be necessary to draw any definite conclusions.

In summary, our investigation harnesses the power of single-cell RNA sequencing to deepen the understanding of arsenic's influence on placental gene expression, revealing both novel genes and sex-specific effects. Our results illuminate the nuanced gene expression changes linked to As exposure, previously undetectable with bulk RNA sequencing. The differential impacts on male and female placentas provide insights into the subtleties of fetal sex on pregnancy outcomes. This study marks a significant step forward, setting the stage for future research to dissect the precise biological roles of these genes in As-related pathologies during pregnancy.

4. The dose of 200 ppb was selected for the main experiment (mice) without further dose-response analysis and with rather unclear justification lacking references.

Response: We appreciate the reviewer's concern regarding the selection of the 200 ppb dose for As in our experiments. To clarify, this dose was chosen based on several considerations that align it with both human exposure levels and established practices in rodent studies.

As detailed in our methods (Lines 619-626), the World Health Organization's provisional guideline for As in drinking water is 10 µg/L (10 ppb)^{36,37}. Exposure levels in humans can range from 10 to 1000 ppb^{36,37}, and rodent studies frequently use doses from 50 to 5000 ppb³⁷. Our selected dose of 200 ppb falls within the lower end of this spectrum, representing a level that is both environmentally relevant and has been associated with adverse health outcomes in humans and animals³⁸.

Specifically, exposure to As concentrations as low as 50 ppb has been linked to adverse pregnancy outcomes, such as spontaneous abortion and low birth weight³⁹. Concentrations above 100 ppb are associated with neurocognitive impairments in children, including reduced IQ and language skills^{40,41}, and in adults with increased risks of psychiatric disorders⁴². In rodents, 200 ppb As has been connected to learning, memory, locomotion, and motor function deficits²⁴⁻²⁶. These references, which substantiate the relevance of our chosen dose, are cited within our manuscript (Lines 121-125; 619-626).

In our study, we observed no significant differences in placental weight, fetal malformation, or rates of resorption or preterm delivery at the 200 ppb dose (**Extended Data Fig. 1**; Lines 125-127). Histological analysis with H&E staining showed that the placental structure and architecture were not compromised by this dose of As (**Fig. 5a**, top panel). Additionally, we found no notable changes in the proportion of cell types in

the placenta by treatment (**Fig. 1c**), further supporting our dose selection as not excessively toxic for placental development in our model.

However, we acknowledge the reviewer's point about the necessity of a dose-response analysis. To address this, we propose future studies that will delineate a more detailed dose-response profile of Prap1 expression in mouse placenta at various gestational stages (E10.5, 12.5, 14.5, and 17.5) and As exposure levels (0, 200, and 1000 ppb). This will enhance our understanding of the temporal dynamics and dose-dependent effects of Prap1's response to As, contributing valuable insights to the field.

Extended Data Fig. 1. The number of (a) placentae and (b) placental weights used in scRNA-seq analysis

Fig. 5. Histology and immunofluorescence staining of PRAP1 in mouse placenta. (a) Hematoxylin and Eosin (H & E) staining (1. Dec: Decidua; 2. JZ: Junctional Zone; C. Lab: Labyrinth). Scale bars represent 100 μ m. (n>=5 placentae in each group)

Fig. 1. scRNA seq analysis of mouse placenta. (c) Proportion of each cluster by treatment group.

5. The findings about how As impacts gene expression in placental cell population is presented very broadly listing genes without any orthogonal validations. Only PRAP1 is validated in human cell lines. More thorough validation is needed.

Response: We acknowledge the importance of reinforcing our scRNA-seq findings through additional protein validation of differentially expressed (DE) genes. We conducted immunofluorescence (IF) staining for CAR2 and GUCA2B (**Extended Data Fig. 7** and **Extended Data Fig. 9**). These proteins were deliberately chosen due to their lack of prior association with As exposure and limited understanding of their functions in the placenta.

CAR2 exhibited an upregulation in GlyT cells in the mouse placenta following As exposure (**Supplementary Table S2; Extended Data Fig. 7**). As noted in previous studies^{43,44}, CAR2 expression was predominantly localized in the junctional zone and decidua of mouse placentae, precisely where GlyT cells reside (**Extended Data Fig. 7**).

The gene *Guca2b*, encoding Guanylate Cyclase Activator 2B (GUCA2B), was upregulated in 22 cell types, albeit its expression was confined to a small fraction of cells (**Fig. 2** and **Extended Data Fig. 8b**). IF staining of GUCA2B protein corroborated this, revealing sparse expression in mouse placentae, making differential expression determination challenging (**Extended Data Fig. 9**). Co-staining with Ly6G, a neutrophil marker, suggested that a considerable portion of GUCA2B expression originates from neutrophils (**Extended Data Fig. 9**).

Extended Data Fig. 7. Immunofluorescence staining of CAR2 in mouse placenta. Scale bars represent 100 μm . Dec: Decidua; 2. JZ: Junctional Zone.

Extended Data Fig. 9. Immunofluorescence staining of GUCA2B in mouse placenta. Scale bars represent 100 μm . Dec: Decidua; 2. JZ: Junctional Zone.

Furthermore, the significant upregulation of additional DE genes, such as *APOA2*, *APOB*, *AFP*, *CA2*, *GPX*, and *GUCA2B* (**Extended Data Fig. 6-9; Supplementary Table S2 and S3**), identified through scRNA-seq analysis in mouse placenta, has been validated by qPCR in human trophoblast cells (**Extended Data Fig. 13e and 13f**). The importance of investigating the upregulation of these genes in the placenta is evident and warrants further exploration.

Extended Data Fig. 13. Validation of *Prap1* expression in human placental cells and tissues Validation of additional DE genes in (e) BeWo and (f) HTR-8/SVneo cells treated with As (n=3). *p<0.05, significant compared to control (0 μM As). #p<0.05, significantly different from each other.

6 . As mentioned above only *PRAP1* is validated focusing on a single cell type only (trophoblast cells lines HTR-8/SVneo, BeWo, and Swan 71) despite the effect is seen across most cell types captured with the rationale that “trophoblasts are the predominant cell type in the placenta”. However, Based on their figure 1b, trophoblast stem cells appears not to be the predominant cell type thus their validation strategy isn't valid. Additional cell types should be used for validation and for TS cells more accurate lines should be considered with potential to differentiation.

Response: Thank you for your valuable insights. Your comments have highlighted an area of potential misunderstanding that we would like to address. As indicated, the most abundant cell type identified in our scRNA-seq analysis is Endothelial_1 (Cluster 1), which comprises approximately 8% of the total cellular population as shown in **Fig. 1c**. In our dataset, we have identified various trophoblast subtypes, including spongiotrophoblasts, trophoblast giant cells (TGCs), labyrinth trophoblasts, and glycogen trophoblasts (GlyT), with trophoblast progenitor cells collectively amounting to roughly 37% of the total cells in mouse placentae. It is important to note that we observed no significant shifts in the proportions of each cell type following As treatment, as depicted in **Fig.1c**.

Moreover, *Prap1* expression was consistently upregulated across most trophoblast cell types in the presence of As, as detailed in **Fig. 3b** and **3c**. Given the critical functions that trophoblasts serve in placental biology²⁻⁴, we selected well-established human

trophoblast cell lines, including HTR-8, BeWo, and Swan71, for our experimental validations.

We wish to clarify that our research did not aim to replicate "trophoblast stem cells" in vitro using these cell lines. Our focus was on examining the response of trophoblast cell models to As exposure and the subsequent regulation of *PRAP1* expression. Although exploring the effects on endothelial cells and other cell types is undoubtedly intriguing and could further elucidate the roles of As and *PRAP1*, such explorations would extend beyond the scope of our current study objectives.

We fully acknowledge this methodological boundary and have discussed the implications and potential avenues for future research that include a broader range of cell types in the discussion section of our manuscript (Lines 598-601).

Fig. 1. scRNA seq analysis of mouse placenta. (c) Proportion of each cluster by treatment group.

Fig. 3. *PRAP1* expression in mouse placenta. *PRAP1* expression by treatment in each cell type (b and c). *FDR-adjusted $P < 0.05$ compared to control (b and c) in each cell type.

7. The authors make a statement that the As effect seen on PRAP1 is sex-specific but yet fail validate these findings in their human in vitro models. They completely leave out the sex of the cell lines used and refer to inconclusive results from the in situ experiments.

Response: We agree with the reviewer's comment. It was not our intention to overlook the sex of the cell lines; indeed, this aspect was initially discussed in our manuscript. Based on prior feedback, we had adjusted the manuscript accordingly, which may have led to the impression that we disregarded the sex of the cell lines.

In our mouse model, we noted a female-biased expression of Prap1 in the placenta. However, this sex-dependent regulation was not replicated in human cell lines or placental tissues. We hypothesize that this inconsistency could be due to interspecies variation, the limited number of human tissue samples available, or differences in experimental conditions such as the comparison of in vivo to ex vivo systems, chronic versus acute exposure, and the various stages of pregnancy examined. This is extensively discussed in the text (Lines 512-523).

Our research indicates that estrogen signaling may play a role in the sex-dependent response to As, with estrogen potentially modulating PRAP1 expression (refer to Extended Data Fig. 14b; Supplementary Excel File 6). Preliminary data from our lab suggest that 17 β -estradiol treatment can upregulate PRAP1 mRNA in human trophoblasts (**unpublished data**). Given that the placenta is a major source of estrogen after the first trimester⁴⁵, this points to a scenario where differential cellular responses to estrogen, rather than estrogen levels per se, could drive the sex-specific PRAP1 expression observed.

Given the complexity of these mechanisms, we believe that additional in vivo investigation is required to elucidate the sex-dependent regulation of PRAP1. Our ongoing studies are focused on mapping PRAP1 expression throughout gestation in both female and male mice exposed to As. These efforts aim to identify the timing and extent of sex-specific expression changes due to As, laying the groundwork for future validation in human placental cells or tissues.

8. It remains unclear how many placentas were analyzed in their discovery cohort of mouse placenta.

Response: Thank you for highlighting this aspect of our methods. We believe these clarifications will address your concerns and we are grateful for the opportunity to ensure the accuracy and transparency of our research methods. We have now included the specific numbers of animals and placentae used in our study within the text (Lines 642-645). Additionally, we have updated **Extended Data Fig. 1** to now include the sex of each analyzed placenta, providing a more comprehensive understanding of our results.

Regarding the scRNA seq, we utilized two time-pregnant mice for each treatment group (control and As exposure). Each mouse bore 7-8 pups, from which we harvested placentae. From each pregnant mouse, we collected 7-8 placentae corresponding to the number of pups (**Extended data Fig. 1**). The fetal sex was determined, and we subsequently pooled the cells of identical sex from each mouse to proceed with the scRNA seq analysis. This approach resulted in a total of eight distinct samples for in-depth analysis: Control_1_Male, Control_1_Female, Control_2_Male, Control_2_Female, Arsenic_1_Male, Arsenic_1_Female, Arsenic_2_Male, Arsenic_2_Female.

Extended Data Fig. 1. The number of (a) placentae and (b) placental weights used in scRNA-seq analysis.

For validation through IF staining, our study included two time-pregnant mice for the control group and three for the As exposure group, yielding 5-7 placentae (pups) per mouse. Although our initial goal was to have at least four mice per group, we encountered lower than anticipated pregnancy rates during the study period. Please note that each data point in **Extended Data Fig. 12** corresponds to the PRAP1 expression observed in each individual placenta.

9. Similarly, the single-cell experiment is not described in sufficient detail including how many cells were loaded per channel and how potential duplets were handled.

Response: Thank you for bringing this to our attention. The sequencing process was conducted at the UR Genomics Center, targeting 10,000 cells per sample for capture by loading 16,500 cells into each channel as per 10X Genomics guidelines. In data analysis, we used several commonly used steps as part of the Seurat analysis package to remove potentially problematic cells and help process the data. Briefly, the Seurat package was utilized for quality control by retaining cells with at least 200 genes and

excluding those with more than 2,500 genes to eliminate doublets. Normalization and sample integration were achieved using the NormalizeData function and SCTransform workflow within Seurat, normalizing expression values across cells and mitigating sequencing depth effects. The detailed methods on this matter are provided in the text (Lines 652-653; 666-686).

10. Related to concern raised in 4) figure 1 is not showing how cell distributions (and expression) are observed in individual samples as only a single integrated dataset is shown. Similarly, feature plots how libraries are distributed is warranted to ensure no technical effects.

Response: We apologize for being unclear. **Fig. 1c** shows the overall distribution of cell types by treatment group, including the p-value comparing the treatment groups. It also shows the proportion of each cell type for each of the 8 samples using black dots (we have clarified this in the legend). We additionally want to emphasize **Extended Data Fig. 17**, which shows the UMAP plot of **Fig. 1b** separated by both treatment group and individual sample. We now included new data (**Extended Data Fig.18** and **Extended Data Fig.19**), which address the question of library complexity by plotting the number of features detected per cell split by treatment group, sex, and each of the 8 samples. Collectively, we feel that our preprocessing steps described in Comment #9 combined with these figures demonstrate that our data is free of systemic technical artifacts or biases by sample, treatment group, or sex.

Fig. 1c. Proportion of each cluster by treatment group.

Extended Data Fig. 17. Final UMAP by sample ID and by treatment

Extended Data Fig. 18. Number of genes expressed per cell by treatment group (a) and by sex (b)

Extended Data Fig. 19. Number of genes expressed per cell by sample

References

- 1 Tosevska, A. *et al.* Integrated analysis of an in vivo model of intra-nasal exposure to instilled air pollutants reveals cell-type specific responses in the placenta. *Sci Rep* **12**, 8438 (2022). <https://doi.org:10.1038/s41598-022-12340-z>
- 2 Nelson, A. C., Mould, A. W., Bikoff, E. K. & Robertson, E. J. Single-cell RNA-seq reveals cell type-specific transcriptional signatures at the maternal-foetal interface during pregnancy. *Nat Commun* **7**, 11414 (2016). <https://doi.org:10.1038/ncomms11414>
- 3 Zhou, X. *et al.* Single-cell RNA-seq revealed diverse cell types in the mouse placenta at mid-gestation. *Exp Cell Res* **405**, 112715 (2021). <https://doi.org:10.1016/j.yexcr.2021.112715>
- 4 Han, X. *et al.* Mapping the Mouse Cell Atlas by Microwell-Seq. *Cell* **173**, 1307 (2018). <https://doi.org:10.1016/j.cell.2018.05.012>
- 5 Marsh, B. & Blelloch, R. Single nuclei RNA-seq of mouse placental labyrinth development. *Elife* **9** (2020). <https://doi.org:10.7554/eLife.60266>
- 6 Jiang, X. *et al.* A differentiation roadmap of murine placentation at single-cell resolution. *Cell Discov* **9**, 30 (2023). <https://doi.org:10.1038/s41421-022-00513-z>
- 7 He, J. P., Tian, Q., Zhu, Q. Y. & Liu, J. L. Identification of Intercellular Crosstalk between Decidual Cells and Niche Cells in Mice. *Int J Mol Sci* **22** (2021). <https://doi.org:10.3390/ijms22147696>
- 8 Vento-Tormo, R. *et al.* Single-cell reconstruction of the early maternal-fetal interface in humans. *Nature* **563**, 347-353 (2018). <https://doi.org:10.1038/s41586-018-0698-6>
- 9 Suryawanshi, H. *et al.* A single-cell survey of the human first-trimester placenta and decidua. *Sci Adv* **4**, eaau4788 (2018). <https://doi.org:10.1126/sciadv.aau4788>
- 10 Tsang, J. C. H. *et al.* Integrative single-cell and cell-free plasma RNA transcriptomics elucidates placental cellular dynamics. *Proc Natl Acad Sci U S A* **114**, E7786-E7795 (2017). <https://doi.org:10.1073/pnas.1710470114>
- 11 Pique-Regi, R. *et al.* Single cell transcriptional signatures of the human placenta in term and preterm parturition. *Elife* **8** (2019). <https://doi.org:10.7554/eLife.52004>
- 12 Yang, Y. *et al.* Transcriptomic Profiling of Human Placenta in Gestational Diabetes Mellitus at the Single-Cell Level. *Front Endocrinol (Lausanne)* **12**, 679582 (2021). <https://doi.org:10.3389/fendo.2021.679582>
- 13 Pavlicev, M. *et al.* Single-cell transcriptomics of the human placenta: inferring the cell communication network of the maternal-fetal interface. *Genome Res* **27**, 349-361 (2017). <https://doi.org:10.1101/gr.207597.116>
- 14 Franzen, O., Gan, L. M. & Bjorkegren, J. L. M. PanglaoDB: a web server for exploration of mouse and human single-cell RNA sequencing data. *Database (Oxford)* **2019** (2019). <https://doi.org:10.1093/database/baz046>
- 15 Home, P., Ghosh, A., Kumar, R. P., Ganguly, A., Bhattacharya, B. Trophoblast paracrine signaling regulates placental hematoendothelial niche. *bioRxiv* (2019).
- 16 Walentin, K., Hinze, C. & Schmidt-Ott, K. M. The basal chorionic trophoblast cell layer: An emerging coordinator of placenta development. *Bioessays* **38**, 254-265 (2016). <https://doi.org:10.1002/bies.201500087>

- 17 Liu, Y. *et al.* Single-cell RNA-seq reveals the diversity of trophoblast subtypes and patterns of differentiation in the human placenta. *Cell Res* **28**, 819-832 (2018). <https://doi.org:10.1038/s41422-018-0066-y>
- 18 Simmons, D. G., Fortier, A. L. & Cross, J. C. Diverse subtypes and developmental origins of trophoblast giant cells in the mouse placenta. *Dev Biol* **304**, 567-578 (2007). <https://doi.org:10.1016/j.ydbio.2007.01.009>
- 19 Simmons, D. G. *et al.* Early patterning of the chorion leads to the trilaminar trophoblast cell structure in the placental labyrinth. *Development* **135**, 2083-2091 (2008). <https://doi.org:10.1242/dev.020099>
- 20 Karlsson, M. *et al.* A single-cell type transcriptomics map of human tissues. *Sci Adv* **7** (2021). <https://doi.org:10.1126/sciadv.abh2169>
- 21 Pandey, R. *et al.* Arsenic Induces Differential Neurotoxicity in Male, Female, and E2-Deficient Females: Comparative Effects on Hippocampal Neurons and Cognition in Adult Rats. *Mol Neurobiol* **59**, 2729-2744 (2022). <https://doi.org:10.1007/s12035-022-02770-1>
- 22 Tyler, C. R. S. *et al.* Sex-Dependent Effects of the Histone Deacetylase Inhibitor, Sodium Valproate, on Reversal Learning After Developmental Arsenic Exposure. *Front Genet* **9**, 200 (2018). <https://doi.org:10.3389/fgene.2018.00200>
- 23 Sobolewski, M., Conrad, K., Marvin, E., Allen, J. L. & Cory-Slechta, D. A. Endocrine active metals, prenatal stress and enhanced neurobehavioral disruption. *Horm Behav* **101**, 36-49 (2018). <https://doi.org:10.1016/j.yhbeh.2018.01.004>
- 24 Gandhi, D. N., Panchal, G. M. & Patel, K. G. Developmental and neurobehavioural toxicity study of arsenic on rats following gestational exposure. *Indian J Exp Biol* **50**, 147-155 (2012).
- 25 Luo, J. *et al.* Maternal and early life arsenite exposure impairs neurodevelopment and increases the expression of PSA-NCAM in hippocampus of rat offspring. *Toxicology* **311**, 99-106 (2013). <https://doi.org:10.1016/j.tox.2013.06.007>
- 26 Martinez-Finley, E. J., Ali, A. M. & Allan, A. M. Learning deficits in C57BL/6J mice following perinatal arsenic exposure: consequence of lower corticosterone receptor levels? *Pharmacol Biochem Behav* **94**, 271-277 (2009). <https://doi.org:10.1016/j.pbb.2009.09.006>
- 27 Liu, Y., Beyer, A. & Aebersold, R. On the Dependency of Cellular Protein Levels on mRNA Abundance. *Cell* **165**, 535-550 (2016). <https://doi.org:10.1016/j.cell.2016.03.014>
- 28 Winterbottom, E. F. *et al.* Transcriptome-wide analysis of changes in the fetal placenta associated with prenatal arsenic exposure in the New Hampshire Birth Cohort Study. *Environ Health* **18**, 100 (2019). <https://doi.org:10.1186/s12940-019-0535-x>
- 29 Winterbottom, E. F. *et al.* The aquaglyceroporin AQP9 contributes to the sex-specific effects of in utero arsenic exposure on placental gene expression. *Environ Health* **16**, 59 (2017). <https://doi.org:10.1186/s12940-017-0267-8>
- 30 Winterbottom, E. F. *et al.* GLI3 Links Environmental Arsenic Exposure and Human Fetal Growth. *EBioMedicine* **2**, 536-543 (2015). <https://doi.org:10.1016/j.ebiom.2015.04.019>
- 31 Winterbottom, E. F. *et al.* Prenatal arsenic exposure alters the placental expression of multiple epigenetic regulators in a sex-dependent manner. *Environ Health* **18**, 18 (2019). <https://doi.org:10.1186/s12940-019-0455-9>

- 32 Bloom, M. S. *et al.* Low level arsenic contaminated water consumption and birth outcomes in Romania-An exploratory study. *Reprod Toxicol* **59**, 8-16 (2016). <https://doi.org:10.1016/j.reprotox.2015.10.012>
- 33 Xu, L. *et al.* Decrease in birth weight and gestational age by arsenic among the newborn in Shanghai, China. *Nihon Koshu Eisei Zasshi* **58**, 89-95 (2011).
- 34 Gilbert-Diamond, D., Emond, J. A., Baker, E. R., Korrick, S. A. & Karagas, M. R. Relation between in Utero Arsenic Exposure and Birth Outcomes in a Cohort of Mothers and Their Newborns from New Hampshire. *Environ Health Perspect* **124**, 1299-1307 (2016). <https://doi.org:10.1289/ehp.1510065>
- 35 Drobna, Z. *et al.* Analysis of maternal polymorphisms in arsenic (+3 oxidation state)-methyltransferase AS3MT and fetal sex in relation to arsenic metabolism and infant birth outcomes: Implications for risk analysis. *Reprod Toxicol* **61**, 28-38 (2016). <https://doi.org:10.1016/j.reprotox.2016.02.017>
- 36 Podgorski, J. & Berg, M. Global threat of arsenic in groundwater. *Science* **368**, 845-850 (2020). <https://doi.org:10.1126/science.aba1510>
- 37 Murko, M., Elek, B., Styblo, M., Thomas, D. J. & Francesconi, K. A. Dose and Diet - Sources of Arsenic Intake in Mouse in Utero Exposure Scenarios. *Chem Res Toxicol* **31**, 156-164 (2018). <https://doi.org:10.1021/acs.chemrestox.7b00309>
- 38 Tyler, C. R. & Allan, A. M. The Effects of Arsenic Exposure on Neurological and Cognitive Dysfunction in Human and Rodent Studies: A Review. *Curr Environ Health Rep* **1**, 132-147 (2014). <https://doi.org:10.1007/s40572-014-0012-1>
- 39 Quansah, R. *et al.* Association of arsenic with adverse pregnancy outcomes/infant mortality: a systematic review and meta-analysis. *Environ Health Perspect* **123**, 412-421 (2015). <https://doi.org:10.1289/ehp.1307894>
- 40 Calderon, J. *et al.* Exposure to arsenic and lead and neuropsychological development in Mexican children. *Environ Res* **85**, 69-76 (2001). <https://doi.org:10.1006/enrs.2000.4106>
- 41 von Ehrenstein, O. S. *et al.* Children's intellectual function in relation to arsenic exposure. *Epidemiology* **18**, 44-51 (2007). <https://doi.org:10.1097/01.ede.0000248900.65613.a9>
- 42 Sen, D. & Sarathi Biswas, P. Arsenicosis: Is it a Protective or Predisposing Factor for Mental Illness? *Iran J Psychiatry* **7**, 180-183 (2012).
- 43 Adamson, S. L. *et al.* Interactions between trophoblast cells and the maternal and fetal circulation in the mouse placenta. *Dev Biol* **250**, 358-373 (2002). [https://doi.org:10.1016/s0012-1606\(02\)90773-6](https://doi.org:10.1016/s0012-1606(02)90773-6)
- 44 Singh, U. *et al.* Expression and functional analysis of genes deregulated in mouse placental overgrowth models: Car2 and Ncam1. *Dev Dyn* **234**, 1034-1045 (2005). <https://doi.org:10.1002/dvdy.20597>
- 45 Albrecht, E. D. & Pepe, G. J. Estrogen regulation of placental angiogenesis and fetal ovarian development during primate pregnancy. *Int J Dev Biol* **54**, 397-408 (2010). <https://doi.org:10.1387/ijdb.082758ea>

REVIEWER COMMENTS

Reviewer #4 (Remarks to the Author):

While the authors have made some attempt to clarify some comments and added descriptive data, the major concerns raised remain unaddressed and referred to as “not the scope of the study” or “to be included in future work”. I have highlighted these concerns in response to original review below.

1. Single-cell analysis of the human and mouse placenta have been extensively studied, yet the authors present a complete and comprehensive paragraph about these findings. Instead, they should contrast (validate) their findings with other available data to convince the reader that their baseline data is solid for instance using maps published in PMID: 36928215 or those cited by authors.

a. While authors have modified the text slightly it did not respond to the request to integrate publicly available data from similar efforts (E 14.5) to highlight any novel part of the first result section. By doing so, it will allow robust validation and as placenta is fully developed by E14 the results may be comparable despite having E 17.5. Nevertheless, it will allow for additional insight extending the current atlas (PMID: 36928215) and validate the “potential novel cell marker”

2. The authors need to validate their findings using human studies linking As with placental transcriptomics to ensure the findings reported here is relevant given known differences between human and mouse placenta (PMID: 35646058; PMID: 31752878; PMID: 26288817 etc.)

a. While I agree with the authors that scRNA analysis can provide nuanced association, large effects are comparable with bulk assays and thus provide an excellent source of validation. The authors should extract genes known from human studies (PMID: 35646058; PMID: 31752878; PMID: 26288817 etc) and provide expression results from their sc-assays. Currently, it is not clear based on the response if this has been done.

3. The dose of 200 ppb was selected for the main experiment (mice) without further dose-response analysis and with rather unclear justification lacking references. This is important as the effect of As on several genes were unspecific and significant across multiple cell types perhaps indicating a toxic, non-relevant dose.

a. Given the targeted journal, I would argue that this relates to the current manuscript and not, as authors indicate, future studies.

4. As mentioned above only PRAP1 is validated focusing on a single cell type only (trophoblast cells lines HTR-8/SVneo, BeWo, and Swan 71) despite the effect is seen across most cell types captured with the rationale that “trophoblasts are the predominant cell type in the placenta”. However, based on their figure 1b, trophoblast stem cells (assuming what these cell lines are used to mimic) appears not to be the predominant cell type thus their validation strategy isn’t valid. Additional cell types should be used for validation and for TS cells more accurate lines should be considered with potential to differentiation (PMID: 29249463).

a. To clarify, the concern relates to using cancer cell lines for validation and not more current models of in vitro models of human trophoblast cells (PMID: 29249463).

5. The authors make a statement that the As effect seen on PRAP1 is sex-specific but yet fail validate these findings in their human in vitro models. They completely leave out the sex of the cell lines used and refer to inconclusive results from the in situ experiments. This is unacceptable and needs to be expanded to validate their sex-specific effect. As mentioned above the TS models (PMID: 29249463) can be obtained or generated based on fetal sex.

a. Once again, the authors acknowledge the limitations with presented results and refer to future studies to elucidate with no attempts to provide additional data. The results on PRAP1 is being sex-specific in mice but cannot be validated is a major concern especially with their concluding sentence “our findings provide novel insights into the placental response to environmental stress and suggest potential preventative and therapeutic strategies to mitigate environment-related adverse health outcomes in both mothers and children”. A discussion paragraph is not sufficient.

Manuscript ID: NCOMMS-23-38005B

We sincerely thank the reviewer for providing the insightful feedback, which has been instrumental in enhancing the quality and accuracy of our manuscript. We have diligently implemented a series of rigorous experiments and analyses in response to the valuable comments provided, thereby significantly enriching our study:

1. New experiments were conducted with mouse placental explants to investigate the dose response of PRAP1 expression in mouse placentae, directly addressing the reviewer's concerns.
2. We have extended our validation to include trophoblast cells derived from both male and female human induced pluripotent stem cells (iPSCs), reinforcing the relevance of our findings in human *in vitro* models.
3. Additional analyses were performed to validate our discovery of novel marker genes within the context of existing studies.
4. We have conducted further analyses to compare the significant genes from our scRNA-seq in mice with those reported in existing human studies.

These new data and detailed clarifications have been incorporated throughout the manuscript, enriching both the text and the figures, as specifically referenced below:

1. Comparative Analysis of Marker Gene Expression (**Supplementary Data 1; Lines 131-135**)
2. Comparative Analysis of Statistically Significant Genes (**Supplementary Data 2; Lines 229-234, 522-532**)
3. Expression of *Prap1* in mouse placental explants exposed to As. (**Extended Data Fig. 13; Lines 381-383, 787-796**)
4. Expression of *PRAP1* in trophoblasts derived from human iPSCs (**Extended Data Fig. 14; Lines 411-417, 543-556, 636-644, 798-809**)

We believe these revisions have significantly improved the manuscript and express our deepest gratitude for the guidance provided by the reviewer. We hope the manuscript now meets the esteemed standards of Nature Communications and look forward to its potential publication.

Reviewer #4

Comment 1. While authors have modified the text slightly it did not respond to the request to integrate publicly available data from similar efforts (E14.5) to highlight any novel part of the first result section. By doing so, it will allow robust validation and as placenta is fully developed by E14 the results may be comparable despite having E 17.5. Nevertheless, it will allow for additional insight extending the current atlas (PMID: 36928215) and validate the “potential novel cell marker”

Response: We thank the reviewer for further clarifying the validations that were initially mentioned. In response, we have utilized data from PMID 36928215¹, specifically

collecting marker genes from Table S2, which correspond to cell clusters predominantly identified in late-stage placentas (as detailed in Figure 1c from PMID 36928215, clusters 2, 3, 5, 7, 8, 9, 15, 16, 17, 19). We then compared the genes that we used for cell type assignment and described in the text as “potentially novel” (*Car2*, *Ceacam3*, *Fdx1*, *Fnd3c2*, *Nup62cl*, *Pappa2*, *Psg17*, and *Psg25*) and their corresponding cell types to results from PMID 36928215, as seen in the table below.

Supplementary Data 1. Comparative Analysis of Marker Gene Expression

Gene	Our candidate cell type	Marker Gene Cell Type From PMID 36928215 ¹
Car2	GlyT	SpT
Car2	GlyT	Gly-T
Ceacam3	Spong_1	SpA-TGC
Fdx1	S-TGC	S-TGC_Precursor
Fdx1	S-TGC	S-TGC
Fnd3c2	S-TGC	S-TGC_Precursor
Fnd3c2	S-TGC	S-TGC
Fnd3c2	S-TGC	SpA-TGC
Nup62cl	S-TGC	LaTP_1
Nup62cl	S-TGC	S-TGC_Precursor
Nup62cl	S-TGC	S-TGC
Pappa2	Spong_1	Gly-T
Pappa2	Spong_1	SpA-TGC
Psg17	Spong_1	SpA-TGC
Psg25	Spong_1	SpA-TGC

Our findings indicate that the use of *Car2*, *Fdx1*, *Fnd3c2*, and *Nup62cl* as marker genes for cell type assignment is well-supported by external validation from PMID 36928215¹. Moreover, the evidence provided by these comparisons justifies the application of *Ceacam3*, *Pappa2*, *Psg17*, and *Psg25* as novel markers for invasive spongiotrophoblasts.

In summary, our findings are consistent with previous studies²⁻¹³ and corroborate findings from research conducted at E14.5¹ emphasizing the reliability and significance of our observations. We extend our gratitude to the reviewer for their insightful comments which helped refine the validations initially mentioned.

Details of the specific changes made to the text are provided below:

Lines 131-135: “In addition to known cell marker genes, we identified potential novel marker genes including *Car2* (glycogen trophoblasts), *Psg17*, *Psg25*, *Pappa2*, *Ceacam3* (spongiotrophoblasts), *Fnd3c2*, *Nup62cL*, and *Fdx1* (S-TGCs). These results align with and validate previous research conducted at E14.5, underscoring the reliability and significance of our findings (Supplementary Data 1)”

Comment 2. The authors should extract genes known from human studies (PMID: 35646058; PMID: 31752878; PMID: 26288817 etc) and provide expression results from their sc-assays.

Response: We agree that exploring the validation of large signals found in bulk assays in human studies is a valuable exercise. In total, we used four different studies (PMID 35646058; PMID 31752878; PMID 26288817; and PMID 30819207)¹⁴⁻¹⁷, selecting genes that were statistically significant based on the p-values reported in each study. This approach yielded variable gene counts: 25 genes from Deysenroth *et al.* (PMID 35646058; Supplementary Tables 4 and 6)¹⁴, 606 genes from Winterbottom *et al.* (PMID 31752878; Additional File 4, comprising 0 genes from females and 606 from males)¹⁵, 6 genes from Winterbottom *et al.* (PMID 26288817; Supplementary Table S1)¹⁶, and 1 gene from Winterbottom *et al.* (PMID 30819207)¹⁷. Subsequently, we converted these human genes to their corresponding mouse orthologs to evaluate their significance across all cell types in our dataset.

Of the 638 genes identified in the referenced studies¹⁴⁻¹⁷, 15 genes exhibited significant differential expression at a relaxed significance threshold of $p < 0.10$ after adjusting for multiple comparisons, as detailed below. These findings are summarized in **Supplementary Data 2** and further discussed in the main text (**Lines 229-234**).

Supplementary Data 2. Comparative Analysis of Statistically Significant Genes

Current study			PMID 31752878 ¹⁵	
Mouse Gene	Cell Type	Adjusted p-value	Human Gene	p-value
Fzd4	B Cell	0.0019	FZD4	8.4e-4
Mgp	Spong_1	0.036	MGP	0.0037
Pdpr	Labyr_Tropho_3	0.0012	PDPN	0.012
Mef2c	Yolk_Sack	0.0076	MEF2C	0.012
Col1a2	Spong_2	0.018	COL1A2	0.014
Vim	Labyr_Tropho_3	2.7e-6	VIM	0.014
Unc5b	GlyT	0.017	UNC5B	0.019
Col3a1	Labyr_Tropho_2	0.041	COL3A1	0.023
Tagln	Spong_2	0.014	TAGLN	0.023
Cavin3	Labyr_Tropho_3	0.0032	CAVIN3	0.026
Tek	Decidual_2	1.5e-12	TEK	0.026
B3gnt9	Endothelial_2	0.003	B3GNT9	0.028
C1qtnf9	Neutrophil_1	0.02	C1QTNF9	0.037
Tmem184a	Yolk_Sack	0.018	TMEM184A	0.041
Fos	B Cell	0.0049	FOS	0.046

The limited replication observed between our study and earlier studies may arise from several factors: (1) differences in analysis objectives and priorities, such as those noted in Deysenroth *et al.* (PMID 35646058)¹⁴ which focuses on differential transcript usage,

(2) challenges in directly comparing results from bulk and single-cell studies, (3) inherent difficulties in replicating findings within human studies, as exemplified by the singular replication of the *PRDM6* gene across two of the above cited studies, and (4) modest signal strengths in the original studies, which could impede achieving sufficient statistical power for replication. Additionally, inherent biological differences between humans and mice may also contribute to these disparities. We have now included a detailed discussion about these issues in the manuscript (**Lines 522-532**).

In summary, we appreciate the opportunity to delve deeper into the comparative analysis prompted by your insightful comments. It has been an enriching exercise to cross-reference our findings with prior human studies and assess the translational relevance of our data.

Details of the specific changes made to the text are provided below:

Lines 229-234: “Additionally, we selected statistically significant genes from four human studies on placental transcriptomic changes with As exposure, converted these human genes to their corresponding mouse orthologs to evaluate their significance across all cell types in our dataset. Of the 638 genes identified in the above cited studies, 15 genes showed significant differential expression at a relaxed significance threshold of $p < 0.10$ after adjusting for multiple comparisons as shown in Supplementary Data 2.”

Lines 522-532: “In comparison to four existing human studies, we identified 15 genes that were significantly regulated by As in both our study and one or more of the referenced studies (Supplementary Data 2). The limited replication observed between our study and these prior works may stem from several factors: 1) Different analysis objectives and priorities, such as differential transcript usage highlighted in Deysenroth *et al.* 2) Challenges in directly comparing results from bulk and single-cell studies. 3) Inherent difficulties in replicating findings within human studies, exemplified by the singular replication of the *PRDM6* gene across two of the cited studies. 4) Modest signal strengths in the original studies, complicating the achievement of sufficient statistical power for replication. Moreover, inherent biological differences between humans and mice may also contribute to these disparities.”

Figure 1 from Winterbottom *et al.* (PMID 26288817)¹⁶

Gene-Level Results from Deyssenroth *et al.* (PMID 35646058)¹⁴

Gene	Gene-Level p-value	Gene	Gene-Level p-value
ACSL3	0.011	LTBP1	0.0012
CALCOCO1	5.10E-04	MARCHF5	0.026
CD151	4.20E-04	MRPL11	0.045
CLIC1	0.014	NSMAF	0.011
CLIC5	0.047	ORMDL1	0.018
COPS5	0.021	SNX6	0.045
DOCK9	0.007	SSBP1	0.014
GATAD2B	0.035	SSR2	0.011
GDA	0.011	TCAF1	0.017
GLG1	0.035	TMBIM6	0.045
IDH3G	0.045	TSPAN3	0.045
KMT2C	0.045	USP15	0.014

Comment 3. The dose of 200 ppb was selected for the main experiment (mice) without further dose-response analysis

Response: We sincerely appreciate the reviewer's comments on the need for dose-response analysis. In accordance with your suggestions, we conducted further experiments with mouse placental tissue explants harvested between E16.5 and E17.5, which are detailed in our manuscript (**Lines 787-796**)^{18,19}. To optimize tissue acclimation, these explants were cultured overnight before exposure to 0, 2, or 5 μM concentrations of As for 24 hours. We then assessed *Prap1* expression via qPCR, and fetal sex was established from the DNA of the fetal tail. The results of these experiments are summarized in **Extended Data Fig. 13.** and discussed in the manuscript text (**Lines 381-383**).

Extended Data Fig. 13. Expression of *Prap1* in mouse placental explants exposed to As. a. *Prap1* expression in female placentae (N=9 placentae). b. *Prap1* expression in male placentae (N=17 placentae). Placentae were collected from four pregnant mice at E 16.5 or 17.5. * $p < 0.05$. ** $p < 0.01$. N.S, not significant.

As presented in **Extended Data Fig. 13a**, there was a significant augmentation of *Prap1* expression in female placental tissues when treated with 2 or 5 μM As. Although we observed an increasing trend with higher As concentrations, the differences were not statistically significant, likely due to high variability and relatively small differences between the As concentrations. On the other hand, male placental tissues did not exhibit significant alterations (**Extended Data Fig. 13b**). These observations are in line with the data from our single-cell RNA sequencing analysis (**Fig.3b**), affirming a sex-

specific regulatory pattern of *Prap1* expression. While this partially addresses the reviewer's concerns, we recognize the limitations of *ex vivo* designs compared to *in vivo* settings.

We are currently establishing a colony of *Prap1* knockout (KO) mice from cryopreserved sperm. We anticipate this process may take approximately one year to yield a validated and sustainable colony. Upon completion, we will embark on comprehensive studies in both wild-type (WT) and KO mice. These studies will delineate the dose-response effects of *Prap1* expression at varied gestational stages (E10.5, E12.5, E14.5, and E17.5) and As exposure levels (0, 200, and 1000 ppb), in addition to examining the subsequent pregnancy and developmental outcomes in the offspring.

Given the constraints of available resources and our commitment to uphold ethical standards in animal research, we humbly suggest reserving the *in vivo* dose-response analysis for the time when our *Prap1* KO mouse cohort is fully established. This course of action not only adheres to the principles of ethical animal use but also ensures that the research conducted will be of the highest scientific validity. Once again, we are grateful for the opportunity to enhance our work through your constructive feedback.

Details of the specific changes made to the text are provided below:

Lines 381-383: “Further experiments conducted on *ex vivo* mouse placental explants have corroborated a sex-specific regulatory pattern in *Prap1* expression, as detailed in Extended Data Fig. 13.”

Lines 787-796: “Mouse placental explants were prepared according to established methods with minor modifications. Pregnant C57BL/6 mice were euthanized between embryonic days 16.5 and 17.5. Freshly collected placenta was rinsed with DPBS, sectioned into six pieces, and placed in individual wells of a 24-well plate containing DMEM/F12 supplemented with 10% FBS and 1% P/S. The explants were incubated overnight to allow for acclimation. The following day, they were exposed to As concentrations of 0, 2, or 5 μ M for 24 hours in duplicate for each condition. The expression of the gene *Prap1* was subsequently quantified by qPCR. A total of 9 female and 17 male placentae from four pregnant mice were analyzed in this study”

Comments 4 & 5. To clarify, the concern relates to using cancer cell lines for validation and not more current models of *in vitro* models of human trophoblast cells (PMID: 29249463) / The authors make a statement that the As effect seen on PRAP1 is sex-specific but yet fail validate these findings in their human *in vitro* models. This is unacceptable and needs to be expanded to validate their sex-specific effect.

Response: We have consolidated our responses to Comments 4 and 5, as both are pertinent to the further validation of our findings using human *in vitro* models.

We understand the reviewer's concern regarding the use of immortalized and transformed cell lines such as HTR-8/SVneo, BeWo, SWAN71, JEG3, and JAR. These cell lines, while extensively utilized in numerous studies (HTR-8/SVneo²⁰⁻²⁷, BeWo^{20,28-33}, SWAN71³⁴⁻³⁶, JEG3^{20-23,37}, and JAR²⁰), may not fully recapitulate the properties of primary human trophoblast cells³⁸⁻⁴¹.

Given the challenges associated with obtaining human placental tissues without pregnancy complications, our research has faced significant barriers. The rarity of uncomplicated placental samples has made it impossible for us to isolate primary trophoblast cells directly from human placentas over recent months. Furthermore, the lack of available vendors who can supply primary trophoblast cells from both male and female placentas has further complicated our efforts.

In response to the reviewer's suggestion, we have reached out to the RIKEN Cell Bank to secure human trophoblast stem cells as referenced by Okae *et al.*³⁸ (PMID: 29249463). We are currently in the process of acquiring both male and female trophoblast stem cells. However, logistical challenges, including material transfer agreement and international shipping, are expected to extend beyond two months. These delays make it impractical to incorporate additional experimental results into the revised manuscript within the stipulated timeframe.

To navigate these constraints, we have explored alternative methods and decided to employ trophoblasts derived from human induced pluripotent stem cells (iPSCs) obtained from both male and female donors. We acknowledge that while human iPSC-derived trophoblasts are an approximation, they do not entirely emulate the functional and molecular characteristics of primary trophoblast cells. Nonetheless, under the current limitations, this approach represents our best option to address the concerns raised and progress the study without undue delay.

As outlined in the Methods section (**Lines 798-809**), human iPSCs from both male and female donors were cultured and differentiated into trophoblasts according to established protocols⁴². We confirmed successful differentiation by measuring the expression of trophoblast cell markers, including *HAND1* and *GATA4*⁴². Subsequently, the differentiated trophoblast cells were exposed to As for either 4 or 24 hours, and the expression of *PRAP1* was assessed. As depicted in **Extended Data Fig. 14e**, expression of *HAND1* and *GATA4* was dramatically increased in the derived

trophoblasts compared to undifferentiated iPSCs. Furthermore, treatment with As resulted in a significant upregulation of PRAP1 after 4 and 24 hours in female trophoblasts, but not in male trophoblasts, clearly demonstrating the sex-dependent responses of PRAP1 to As exposure (**Extended Data Fig. 14f**). These additional experiments substantiate our findings from mouse models in human *in vitro* settings.

Extended Data Fig. 14. Validation of PRAP1 expression in human placental cells and tissues. (e) Expression of trophoblast markers in trophoblasts derived from male and female human iPSCs (n=3). *** $p < 0.001$ and **** $p < 0.0001$, significantly different from each other. (f) Expression of *PRAP1* in trophoblasts from male and female human iPSCs exposed to As for 4 hr (n=3). * $p < 0.05$, ** $p < 0.01$, *** $p < 0.001$, and **** $p < 0.0001$, significant compared to control (0 μM As). # $p < 0.05$, significantly different from each other.

Although our study demonstrates that PRAP1 is upregulated in response to As exposure in both mouse and human placental models, we did not observe sex-dependent regulation of PRAP1 in human placental tissues. This discrepancy may be attributed to various factors including limited tissue samples, inherent species differences, and differences in experimental designs. Research across various biological and medical fields has highlighted the complexities of sex differences, revealing that observations made *in vivo* do not always align with those *in vitro*. For example, distinct patterns of mitochondrial damage were observed in male and female cells exposed to medications like Diclofenac and Acetaminophen; such differences were evident *in vitro* but did not consistently replicate in animal studies with rats and mice, where no clear sex-specific patterns emerged across different species and tissue types⁴³. Similarly, in cardiovascular research, while certain sex-dependent gene expression changes post-myocardial infarction and during caloric restriction have been documented in animal models, these changes are not always mirrored in human cardiac cells⁴⁴. Additionally, studies using liver cancer xenograft models to assess the sex-dependent efficacy and adverse reactions to anticancer drugs have produced findings that do not always correspond with clinical data observed in humans⁴⁵. Moreover, findings in animal models are not always replicated in human models⁴⁶⁻⁴⁸. These discrepancies highlight the need for careful consideration and thorough research to understand the implications of biological sex differences fully. The intricate interplay of hormones, genetic factors, and environmental influences complicates these efforts and underscores the challenges of translating laboratory findings into clinical practice.

(Lines 543-556)

We are immensely grateful for the reviewer's reference to established human trophoblast stem cells as a robust *in vitro* model³⁸. The pioneering work by Okae et al.³⁸ in developing trophoblast stem cells from cytotrophoblasts and blastocysts provides a foundational tool for understanding the molecular and functional dynamics of human trophoblasts. The recent advances reported by Hori et al.³⁹ in creating trophoblast organoids from these stem cells highlight the potential of these models **(Lines 636-644)**. These studies represent a significant breakthrough in studying the complex biological processes of human trophoblasts, offering a more physiologically relevant and ethically sustainable platform compared to traditional *in vitro* systems. These models provide crucial insights into trophoblast behavior and function and open opportunities for advanced research and collaboration. We are eager to engage with these research teams to leverage their insights and methodologies to enhance our understanding of placental biology, particularly its interactions with environmental stressors such as As. Leveraging these innovative approaches will improve our ability to simulate and study the intricate mechanisms governing placental adaptation and pathology, potentially leading to better maternal and fetal health outcomes. Our ongoing efforts to integrate these models into our research reflect our commitment to achieving the highest standards of scientific inquiry and clinical relevance.

Details of the specific changes made to the text are provided below:

Lines 411-417: “To further explore the sex-dependent regulation of PRAP1 in human models, we utilized trophoblasts derived from both male and female human induced pluripotent stem cells (iPSCs). As illustrated in Extended Data Fig. 14e, these differentiated trophoblasts express the trophoblast-specific markers including *HAND1* and *GATA4*. Treatment with As resulted in a female-biased upregulation of *PRAP1*, consistent with our findings in mouse placentae (Extended Data Fig. 14f).”

Lines 543-556: “Our study demonstrates that PRAP1 is upregulated in response to As exposure in both mouse and human placental models. Notably, we observed a female-biased expression pattern of PRAP1 in the mouse placenta and human trophoblasts derived from iPSCs. However, we did not observe sex-dependent regulation of PRAP1 in human placental tissues. This discrepancy may be attributed to various factors including limited tissue samples, inherent species differences, and differences in experimental designs—such as *in vivo* versus *ex vivo* approaches, chronic versus acute exposure, and stages of pregnancy. The complex nature of sex differences in biological and medical contexts suggests that phenomena observed *in vivo* do not always replicate *in vitro*. Moreover, findings in animal models are not always replicated in human models. These discrepancies highlight the challenges in translating laboratory findings to clinical practice and underscore the need for careful consideration and comprehensive research to fully understand the implications of biological sex differences on human health.”

Lines 636-644: “In addition, the sex-dependent regulation of PRAP1 and its underlying mechanisms require further exploration in human models to substantiate its public health implications, particularly in regulating As toxicity relevant to maternal and fetal health. The pioneering efforts by Okae *et al.* in developing trophoblast stem cells from cytotrophoblasts and blastocysts provide an essential tool for understanding the molecular and functional dynamics of human trophoblasts. Recent advancements by Hori *et al.* in creating trophoblast organoids from these stem cells underscore the potential of these models to offer deeper insights into the molecular and functional dynamics of human trophoblasts”

Lines 798-809: “Human induced pluripotent stem cells (iPSCs, SKU: 30HU-002) were procured from iXCells Biotechnologies, San Diego, CA, sourced from healthy male (Lot: 400531) and female (Lot: 400530) donors. These cells were maintained and propagated in Human iPSC Feeder-Free Growth Medium (MD-0019, iXCells Biotechnologies) on plates precoated with Cultrex (3434-005-02, R&D Systems), adhering strictly to the supplier’s guidelines. The iPSCs underwent differentiation into trophoblasts using the same medium supplemented with 100 ng/ml BMP4 (314-BP, R & D Systems), following established protocols. Subsequently, the differentiated trophoblast cells were treated with As concentrations of 0, 2, or 5 μ M for 4 or 24 hrs. qPCR analysis was then performed to measure the expression levels of *PRAP1* as well as trophoblast markers *HAND1* and *GATA4*”

1. Jiang X, Wang Y, Xiao Z, et al. A differentiation roadmap of murine placentation at single-cell resolution. *Cell Discov* 2023;9(1):30. DOI: 10.1038/s41421-022-00513-z.
2. Han X, Wang R, Zhou Y, et al. Mapping the Mouse Cell Atlas by Microwell-Seq. *Cell* 2018;173(5):1307. DOI: 10.1016/j.cell.2018.05.012.
3. Marsh B, Blleloch R. Single nuclei RNA-seq of mouse placental labyrinth development. *Elife* 2020;9. DOI: 10.7554/eLife.60266.
4. Zhou X, Xu Y, Ren S, et al. Single-cell RNA-seq revealed diverse cell types in the mouse placenta at mid-gestation. *Exp Cell Res* 2021;405(2):112715. DOI: 10.1016/j.yexcr.2021.112715.
5. Franzen O, Gan LM, Bjorkegren JLM. PanglaoDB: a web server for exploration of mouse and human single-cell RNA sequencing data. *Database (Oxford)* 2019;2019. DOI: 10.1093/database/baz046.
6. Nelson AC, Mould AW, Bikoff EK, Robertson EJ. Single-cell RNA-seq reveals cell type-specific transcriptional signatures at the maternal-foetal interface during pregnancy. *Nat Commun* 2016;7:11414. DOI: 10.1038/ncomms11414.
7. Home P, Ghosh, A., Kumar, R. P., Ganguly, A., Bhattacharya, B. Trophoblast paracrine signaling regulates placental hematoendothelial niche. *bioRxiv* 2019.
8. Walentin K, Hinze C, Schmidt-Ott KM. The basal chorionic trophoblast cell layer: An emerging coordinator of placenta development. *Bioessays* 2016;38(3):254-65. DOI: 10.1002/bies.201500087.
9. Liu Y, Fan X, Wang R, et al. Single-cell RNA-seq reveals the diversity of trophoblast subtypes and patterns of differentiation in the human placenta. *Cell Res* 2018;28(8):819-832. DOI: 10.1038/s41422-018-0066-y.
10. Simmons DG, Fortier AL, Cross JC. Diverse subtypes and developmental origins of trophoblast giant cells in the mouse placenta. *Dev Biol* 2007;304(2):567-78. DOI: 10.1016/j.ydbio.2007.01.009.
11. Simmons DG, Natale DR, Begay V, Hughes M, Leutz A, Cross JC. Early patterning of the chorion leads to the trilaminar trophoblast cell structure in the placental labyrinth. *Development* 2008;135(12):2083-91. DOI: 10.1242/dev.020099.
12. He JP, Tian Q, Zhu QY, Liu JL. Identification of Intercellular Crosstalk between Decidual Cells and Niche Cells in Mice. *Int J Mol Sci* 2021;22(14). DOI: 10.3390/ijms22147696.
13. Karlsson M, Zhang C, Mear L, et al. A single-cell type transcriptomics map of human tissues. *Sci Adv* 2021;7(31). DOI: 10.1126/sciadv.abh2169.
14. Deysenroth MA, Peng S, Hao K, Marsit CJ, Chen J. Placental Gene Transcript Proportions are Altered in the Presence of In Utero Arsenic and Cadmium Exposures, Genetic Variants, and Birth Weight Differences. *Front Genet* 2022;13:865449. DOI: 10.3389/fgene.2022.865449.
15. Winterbottom EF, Ban Y, Sun X, et al. Transcriptome-wide analysis of changes in the fetal placenta associated with prenatal arsenic exposure in the New Hampshire Birth Cohort Study. *Environ Health* 2019;18(1):100. DOI: 10.1186/s12940-019-0535-x.
16. Winterbottom EF, Fei DL, Koestler DC, et al. GLI3 Links Environmental Arsenic Exposure and Human Fetal Growth. *EBioMedicine* 2015;2(6):536-43. DOI: 10.1016/j.ebiom.2015.04.019.

17. Winterbottom EF, Moroishi Y, Halchenko Y, et al. Prenatal arsenic exposure alters the placental expression of multiple epigenetic regulators in a sex-dependent manner. *Environ Health* 2019;18(1):18. DOI: 10.1186/s12940-019-0455-9.
18. Sato BL, Ward MA, Astern JM, Kendal-Wright CE, Collier AC. Validation of murine and human placental explant cultures for use in sex steroid and phase II conjugation toxicology studies. *Toxicol In Vitro* 2015;29(1):103-12. DOI: 10.1016/j.tiv.2014.09.008.
19. Yung HW, Burton GJ, Charnock-Jones DS. Protocol for culturing the endocrine junctional zone of the mouse placenta in serum-free medium. *STAR Protoc* 2023;4(3):102384. DOI: 10.1016/j.xpro.2023.102384.
20. Wu D, Xu Y, Zou Y, et al. Long Noncoding RNA 00473 Is Involved in Preeclampsia by LSD1 Binding-Regulated TFPI2 Transcription in Trophoblast Cells. *Mol Ther Nucleic Acids* 2018;12:381-392. DOI: 10.1016/j.omtn.2018.05.020.
21. Wang XH, Xu S, Zhou XY, et al. Low chorionic villous succinate accumulation associates with recurrent spontaneous abortion risk. *Nat Commun* 2021;12(1):3428. DOI: 10.1038/s41467-021-23827-0.
22. Revankar CM, Cimino DF, Sklar LA, Arterburn JB, Prossnitz ER. A transmembrane intracellular estrogen receptor mediates rapid cell signaling. *Science* 2005;307(5715):1625-30. DOI: 10.1126/science.1106943.
23. Miner JJ, Cao B, Govero J, et al. Zika Virus Infection during Pregnancy in Mice Causes Placental Damage and Fetal Demise. *Cell* 2016;165(5):1081-1091. DOI: 10.1016/j.cell.2016.05.008.
24. Zhao Y, Zhao G, Li W. MicroRNA-495 suppresses pre-eclampsia via activation of p53/PUMA axis. *Cell Death Discov* 2022;8(1):132. DOI: 10.1038/s41420-022-00874-0.
25. Zheng Q, Gan H, Yang F, et al. Cytoplasmic m(1)A reader YTHDF3 inhibits trophoblast invasion by downregulation of m(1)A-methylated IGF1R. *Cell Discov* 2020;6:12. DOI: 10.1038/s41421-020-0144-4.
26. Zhao HJ, Klausen C, Li Y, Zhu H, Wang YL, Leung PCK. Bone morphogenetic protein 2 promotes human trophoblast cell invasion by upregulating N-cadherin via non-canonical SMAD2/3 signaling. *Cell Death Dis* 2018;9(2):174. DOI: 10.1038/s41419-017-0230-1.
27. Liu J, Hao S, Chen X, et al. Human placental trophoblast cells contribute to maternal-fetal tolerance through expressing IL-35 and mediating iT(R)35 conversion. *Nat Commun* 2019;10(1):4601. DOI: 10.1038/s41467-019-12484-z.
28. Lee JG, Yon JM, Kim G, et al. PIBF1 regulates trophoblast syncytialization and promotes cardiovascular development. *Nat Commun* 2024;15(1):1487. DOI: 10.1038/s41467-024-45647-8.
29. Miura S, Sato K, Kato-Negishi M, Teshima T, Takeuchi S. Fluid shear triggers microvilli formation via mechanosensitive activation of TRPV6. *Nat Commun* 2015;6:8871. DOI: 10.1038/ncomms9871.
30. Hawkins SJ, Crompton LA, Sood A, et al. Nanoparticle-induced neuronal toxicity across placental barriers is mediated by autophagy and dependent on astrocytes. *Nat Nanotechnol* 2018;13(5):427-433. DOI: 10.1038/s41565-018-0085-3.
31. Zhang Y, Le T, Grabau R, et al. TMEM16F phospholipid scramblase mediates trophoblast fusion and placental development. *Sci Adv* 2020;6(19):eaba0310. DOI: 10.1126/sciadv.aba0310.

32. Slonchak A, Wang X, Aguado J, et al. Zika virus noncoding RNA cooperates with the viral protein NS5 to inhibit STAT1 phosphorylation and facilitate viral pathogenesis. *Sci Adv* 2022;8(48):eadd8095. DOI: 10.1126/sciadv.add8095.
33. Buchrieser J, Degrelle SA, Couderc T, et al. IFITM proteins inhibit placental syncytiotrophoblast formation and promote fetal demise. *Science* 2019;365(6449):176-180. DOI: 10.1126/science.aaw7733.
34. Alexandrova M, Manchorova D, You Y, Mor G, Dimitrova V, Dimova T. Functional HLA-C expressing trophoblast spheroids as a model to study placental-maternal immune interactions during human implantation. *Sci Rep* 2022;12(1):10224. DOI: 10.1038/s41598-022-12870-6.
35. Kwon JY, Aldo P, You Y, et al. Relevance of placental type I interferon beta regulation for pregnancy success. *Cell Mol Immunol* 2018;15(12):1010-1026. DOI: 10.1038/s41423-018-0050-y.
36. Suzuki T, Iizuka T, Kagami K, et al. Laeverin/aminopeptidase Q induces indoleamine 2,3-dioxygenase-1 in human monocytes. *iScience* 2023;26(9):107692. DOI: 10.1016/j.isci.2023.107692.
37. McConkey CA, Delorme-Axford E, Nickerson CA, et al. A three-dimensional culture system recapitulates placental syncytiotrophoblast development and microbial resistance. *Sci Adv* 2016;2(3):e1501462. DOI: 10.1126/sciadv.1501462.
38. Okae H, Toh H, Sato T, et al. Derivation of Human Trophoblast Stem Cells. *Cell Stem Cell* 2018;22(1):50-63 e6. DOI: 10.1016/j.stem.2017.11.004.
39. Hori T, Okae H, Shibata S, et al. Trophoblast stem cell-based organoid models of the human placental barrier. *Nat Commun* 2024;15(1):962. DOI: 10.1038/s41467-024-45279-y.
40. Bilban M, Tauber S, Haslinger P, et al. Trophoblast invasion: assessment of cellular models using gene expression signatures. *Placenta* 2010;31(11):989-96. DOI: 10.1016/j.placenta.2010.08.011.
41. Lee CQ, Gardner L, Turco M, et al. What Is Trophoblast? A Combination of Criteria Define Human First-Trimester Trophoblast. *Stem Cell Reports* 2016;6(2):257-72. DOI: 10.1016/j.stemcr.2016.01.006.
42. Chen Y, Wang K, Chandramouli GV, Knott JG, Leach R. Trophoblast lineage cells derived from human induced pluripotent stem cells. *Biochem Biophys Res Commun* 2013;436(4):677-84. DOI: 10.1016/j.bbrc.2013.06.016.
43. Mennecozzi M, Landesmann B, Palosaari T, Harris G, Whelan M. Sex differences in liver toxicity-do female and male human primary hepatocytes react differently to toxicants in vitro? *PLoS One* 2015;10(4):e0122786. DOI: 10.1371/journal.pone.0122786.
44. Murphy E, Steenbergen C. Estrogen regulation of protein expression and signaling pathways in the heart. *Biol Sex Differ* 2014;5(1):6. DOI: 10.1186/2042-6410-5-6.
45. Oh S, Jung J. Sex-dependent liver cancer xenograft models for predicting clinical data in the evaluation of anticancer drugs. *Lab Anim Res* 2021;37(1):10. DOI: 10.1186/s42826-021-00087-z.
46. Suda T, Takahashi F, Takahashi N. Bone effects of vitamin D - Discrepancies between in vivo and in vitro studies. *Arch Biochem Biophys* 2012;523(1):22-9. DOI: 10.1016/j.abb.2011.11.011.

47. Perrin S. Preclinical research: Make mouse studies work. *Nature* 2014;507:423–425.
48. Vatner SF. Why So Few New Cardiovascular Drugs Translate to the Clinics. *Circ Res* 2016;119(6):714-7. DOI: 10.1161/CIRCRESAHA.116.309512.

REVIEWERS' COMMENTS

Reviewer #4 (Remarks to the Author):

I have no further comments, authors have addressed my previous concerns.